# Fast joint parity measurement via collective interactions induced by stimulated emission

**Sainan Huai**[1,2], **Kunliang Bu**[1,2], **Xiu Gu** [1,2] ✉, **Zhenxing Zhang**[1], **Shuoming An** [1], **Xiaopei Yang**[1], **Yuan Li**[1], **Tianqi Cai** [1] ✉ **& Yicong Zheng** [1]

Parity detection is essential in quantum error correction. Error syndromes coded in parity are detected routinely by sequential CNOT gates. Here, different from the standard CNOT-gate based scheme, we propose a reliable joint parity measurement (JPM) scheme inspired by stimulated emission. By controlling the collective behavior between data qubits and syndrome qubit, we realize the parity detection and experimentally implement the weight-2 and weight-4 JPM scheme in a tunable coupling superconducting circuit, which shows comparable performance to the CNOT scheme. Moreover, with the aid of the coupling tunability in quantum system, this scheme can be further utilized for specific joint entangling state preparation (JEP) with high fidelity, such as multiqubit entangled state preparation for non-adjacent qubits. This strategy, combined with the superconducting qubit system with tunable couplers, reveals tremendous potential and applications in the surface code architecture without adding extra circuit elements. Besides, the method we develop here can readily be applied in large-scale quantum computation and quantum simulation.

Parity measurement is an essential step in both classical communication and quantum error correction[1]. The parity of a bit string is computed as the modular 2 sum of all bits, in other words, the parity is 1 (0) if the total number of 1 in the bit string is odd (even). Classically, modular 2 sum is computed by XOR gates, while in quantum error correction, it is coded in the CNOT gates[2-5]. For the widely used surface code architecture[6-8], to detect the parity of $n$ qubits, an extra syndrome qubit is initialized at $|0\rangle$, and then a CNOT gate is applied $n$ times between each data qubit and syndrome qubit. The measurement outcome of syndrome qubit 0 (1) indicates even (odd) parity of data qubits[2,6]. This standard CNOT-gate-based parity measurement scheme usually utilizes a two-qubit entangling control-Z (CZ) gate in a superconducting qubit system and has been demonstrated in various experiments[2-4,9]. Other parity measurement protocols, such as mapping the parity information of multiple data qubits to the readout cavities where qubits and cavities are directly coupled[10-13], have also been put forward and demonstrated. However, these schemes usually require readout cavities to couple with multiple qubits dispersively, which is weaker than resonant coupling and are usually used in combination with high-efficiency single photon detectors and high-gain parametric amplifiers, thus resulting in hardware overhead for the surface code architecture.

Stimulated emission, a well-known phenomenon in quantum optics, reveals that the number of total bosonic excitations is encoded in their collective interaction strength with atoms. The probability amplitude of adding (removing) a bosonic excitation to $N$ present ones is proportional to $\sqrt{N+1}$ ($\sqrt{N}$). Inspired by this excitation-related collective behavior, we find that the parity detection of two data qubits can be achieved by making them simultaneously coupled to the syndrome qubit via resonant Rabi oscillation. Similar to the stimulated emission, the strengths of exciting the syndrome qubit $g_+$ (which removes one excitation from the data qubits) and that of de-exciting the syndrome qubit $g_-$ (which adds one excitation to the data qubits) will be different, depending on the total excitation in the data qubits. In practice, when we put the syndrome qubit in the superposition states with a Hadamard gate, different interference patterns occur between

[1]Quantum Laboratory, Tencent, 518057 Shenzhen, Guangdong, China. [2]These authors contributed equally: Sainan Huai, Kunliang Bu, Xiu Gu.
✉e-mail: guxiu1@gmail.com; tianqicai@tencent.com

the exciting and de-exciting paths, bringing the syndrome qubit to parity-dependent states.

In this work, we propose and demonstrate the joint parity measurement (JPM) scheme based on the surface code architecture with tunable qubit-qubit coupling strength[14,15]. By controlling the collective behavior of the two data qubits and utilizing the syndrome qubit as an intermediate state[16,17], a parity-sensitive interaction is generated between data qubits and syndrome qubits. Meanwhile, this controllable collective behavior, which can be applied to both the iSWAP and CZ operations architecture, promotes $\sqrt{2}$ times sped up over the corresponding CNOT scheme based on the homogeneous gate operation type. Besides, we demonstrate the weight-2 and weight-4 JPM schemes with an average of 95.2% and 90.0% parity fidelity, which are comparable to that of the CNOT scheme in this experiment. Our joint parity measurement scheme, being compatible with the existing superconducting quantum processor architecture with tunable couplers[14,15,18], requires no additional device modification or circuit complexity of parity measurement. The reduced circuit time and the improved parity detection fidelity reveal its potential applications in quantum error correction and fault-tolerant quantum computation[19–21]. Moreover, we develop this protocol to further experimentally realize the specific joint entangling states preparation (JEP) with high fidelity in the same scheme with varied coupling strength.

## Results

### Theoretical model

We consider a routine setup where a syndrome qubit ($Q_0$) couples to two data qubits ($Q_1$ and $Q_2$) with coupling strengths $g_1$ and $g_2$, respectively. Different from the standard CNOT parity measurement scheme, where the coupling $g_i$ is turned on/off alternatively, here, both $g_i$ are activated simultaneously and adjusted to be equal, i.e., $g_1 = g_2 = g$, such that collective gate operation[16] can be realized. Then the Hamiltonian system in the interaction picture with the two-level approximation can be expressed as

$$H_s = g\sigma_0^{n,n+1}(\sigma_1^{10} + \sigma_2^{10}) + \text{H.c}, \qquad (1)$$

where $\sigma_i^{n,m} = |n\rangle_i\langle m|$ is the $|m\rangle \to |n\rangle$ transition matrix element for qubit $i$. For the data qubits $Q_1$ and $Q_2$, if we tune their frequencies into resonance with the syndrome qubit $Q_0$, i.e., $\omega_1 = \omega_2 = \omega_0$, then $\sigma_0^{n,n+1} = |0\rangle_0\langle 1|$ and Eq. (1) describes the collective iSWAP operation. Similarly, if we set $\omega_1 = \omega_2 = \omega_0 + \alpha_0$, where $\alpha_0$ is the anharmonicity of $Q_0$, then $\sigma_0^{n,n+1} = |1\rangle_0\langle 2|$ and Eq. (1) realizes collective CZ operation. Therefore, our scheme can be adapted to the hardware where sequential collective iSWAP and collective CZ operations are feasible. In the following, we focus on the collective iSWAP case, and the method can be easily extended to the collective CZ case.

To treat the two data qubits as a whole, we introduce the two-qubit Dicke basis, $\left|D^0\right\rangle = |00\rangle$, $\left|D^1\right\rangle = (|01\rangle + |10\rangle)/\sqrt{2}$, $\left|D^2\right\rangle = |11\rangle$, $\left|d^1\right\rangle = (|01\rangle - |10\rangle)/\sqrt{2}$. Here the superscript denotes the number of $|1\rangle$ among data qubits. Thus, it being odd (even) determines parity being 0 (1). Then Eq. (1) can be simplified by introducing the collective raising operator $J_+ = \sigma_1^{10} + \sigma_2^{10}$. Expressed in the first three symmetric bases, the matrix element of the operator $J_+ = J_-^\dagger$ are

$$g_+^k = g_-^{k+1} = \langle D^{k+1}|J_+|D^k\rangle = \sqrt{(2-k)(k+1)}, \qquad (2)$$

where $g_+^k$ ($g_-^k$) denotes the amplitude of $\left|D^k\right\rangle$ climbing up (down) the Dicke states ladder. Notice that for the dark state $\left|d^1\right\rangle$, $J_+\left|d^1\right\rangle = 0$.

With the help of Eq. (2), the first term in Eq. (1) can be interpreted as de-exciting $Q_0$ from the state $|1\rangle$ ($|n+1\rangle$) to $|0\rangle$ ($|n\rangle$) while raising the

data qubits from the collective Dicke state $\left|D^k\right\rangle$ to $\left|D^{k+1}\right\rangle$. In this way, Eq. (1) takes the same form as the Jaynes–Cummings model, which conserves the total excitation. For each $k+1$ total excitation subspace, the dynamics of $H_s$ in Eq. (1) follow the well-known two-level dynamics

$$U_s(t) = \cos\left(gg_+^k t\right) - i\sin\left(gg_+^k t\right)\tilde{\sigma}_x, \qquad (3)$$

where $\tilde{\sigma}_x$ is defined on the basis of $\left|1, D^k\right\rangle$ and $\left|0, D^{k+1}\right\rangle$, with the states described in the order of $|Q_0Q_1Q_2\rangle$ and $|1\rangle$ ($|0\rangle$) denoting the state of $Q_0$. We can note that the oscillation frequency $gg_+^k$ encodes parity information $k$, i.e., the common hybridized modes of the system undergo oscillations based on the total parity.

To map the even (odd) $k$ onto the state $|1\rangle$ ($|0\rangle$) of $Q_0$, we design the circuit as shown in Fig. 1a. Without loss of generality, we take $\left|0, \psi^k\right\rangle$ as an initial state, where $\left|\psi^k\right\rangle$ denotes the two-qubit Dicke basis. (i) A Hadamard gate is first applied to $Q_0$, creating an equal superposition of $|0\rangle\left|\psi^k\right\rangle$ and $|1\rangle\left|\psi^k\right\rangle$. (ii) A JPM unitary gate $U_s = U_s(\pi/(\sqrt{2}g))$ is turned on. Similar to the stimulated emission, the presence of one extra excitation from $Q_0$ makes these two states undergo different Rabi oscillations, as depicted in Fig. 1b. The

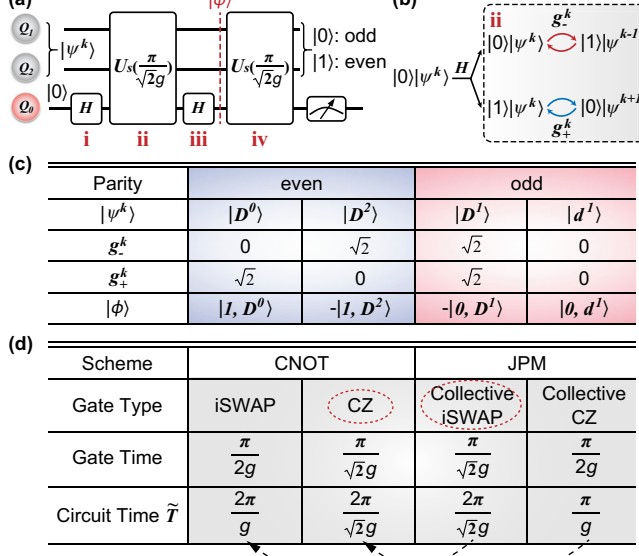

**Fig. 1 | Theoretical model of the joint parity measurement scheme. a** Joint parity measurement protocol for two qubits ($Q_1$, $Q_2$), where the syndrome qubit ($Q_0$) is initialized in state $|0\rangle$, and the data qubits are assumed to be $\left|\psi^k\right\rangle$ without loss of generality. Here, $\left|\psi^k\right\rangle$ denotes the collective two-qubit Dicke basis for the data qubits, with $k$ being the number of $|1\rangle$ among data qubits. A three-qubit JPM gate $U_s$ with time duration $\pi/(\sqrt{2}g)$ is turned on between two Hadamard gates on $Q_0$. Then a second JPM gate is applied to undo the possible extra phase. Even (odd) parity is mapped onto the measurement outcome 1 (0) of $Q_0$. **b** When the JPM gate is turned on, depending on the syndrome qubit being $|0\rangle$ ($|1\rangle$), $\left|\psi^k\right\rangle$ undergoes two different oscillations, that is $|0\rangle\left|\psi^k\right\rangle \leftrightarrow |1\rangle\left|\psi^{k-1}\right\rangle$ and $|1\rangle\left|\psi^k\right\rangle \leftrightarrow |0\rangle\left|\psi^{k+1}\right\rangle$ with corresponding frequency $gg_-^k$ and $gg_+^k$. **c** The table lists the oscillation frequency (third and fourth line) and the intermediate state (last line) after stage iii in (a) for different data qubits input state $\left|\psi^k\right\rangle$ (the first line). The blue (red) background denotes the even (odd) parity of data qubits. **d** Circuit time comparison between the 2-qubit JPM scheme and CNOT scheme (the 4-qubit case has a similar result, except that all the circuit time $\tilde{T}$ need to be doubled). Here, the circuit time calculated in the table mainly involves the entangling gate time for intuition while ignoring single-qubit gate time. The JPM scheme promotes $\sqrt{2}$ times sped up over the CNOT scheme based on the homogeneous gate type, indicated as the black dotted arrow. The gate types circled by the red dashed line are utilized in the experiment.

transition of $|0\rangle|\psi^k\rangle \longrightarrow |1\rangle|\psi^{k-1}\rangle$ ($|1\rangle|\psi^k\rangle \longrightarrow |0\rangle|\psi^{k+1}\rangle$) belongs to the total excitation $k$ ($k+1$) subspace and oscillates at frequency $gg_+^k$ ($gg_-^k$). We choose the evolution time $t$ of $U_s$ such that $|\psi^k\rangle$ will not be transferred to $|\psi^{k-1}\rangle$ or $|\psi^{k+1}\rangle$. As shown in Eq. (3), this amounts to $\sin(gg_\pm^k t) = 0$. (iii) A second Hadamard gate is applied on $Q_0$ and we obtain

$$|\phi\rangle = (|1\rangle|\Psi\rangle_+ + |0\rangle|\Psi\rangle_-)/2, \qquad (4)$$

with $|\Psi\rangle_\pm = (\cos(gg_-^k t) \mp \cos(gg_+^k t))|\psi^k\rangle$. At this stage, we can observe that the parity information $k$, encoded in $g_\pm^k$, is now correlated with the state $|0\rangle$ ($|1\rangle$) of $Q_0$. It is clear from Eq. (4) that $|\Psi\rangle_+$ ($|\Psi\rangle_-$) should contain only even (odd) $k$ so as to map the even (odd) parity of the data qubits onto the state $|1\rangle$ ($|0\rangle$) of $Q_0$. Based on the coupling matrix elements $g_\pm^k$ listed in the third and fourth rows of Fig. 1(c), we can observe that parity detection can be satisfied by setting the evolution time $t = \pi/(\sqrt{2}g)$. At this moment, different Rabi oscillations are aligned, i.e. $|\cos(gg_\pm^k t)| = 1$. We calculate $|\phi\rangle$ for each possible $|\psi^k\rangle$ and list the result in the last line of Fig. 1c. However, it can be found that there are two input states accumulating a $\pi$ phase in the intermediate state $|\phi\rangle$: input state $|0, D^2\rangle$ becomes $-|1, D^2\rangle$, while $|0, D^1\rangle$ transforms to $-|0, D^1\rangle$. Although these extra phases preserve parity, they introduce additional influence to the computational space. To undo the extra phase, we apply another $U_s(\pi/(\sqrt{2}g))$ in the circuit, and the overall protocol for parity detection is then described as

$$
\begin{aligned}
U &= U_s(\pi/(\sqrt{2}g))H_0 U_s(\pi/(\sqrt{2}g))H_0 \\
&= i\sigma_0^y(|00\rangle\langle00| + |11\rangle\langle11|) + I_0\left(|D^1\rangle\langle D^1| + |d^1\rangle\langle d^1|\right),
\end{aligned} \qquad (5)
$$

where $\sigma_0^y$ is the Pauli $Y$ on $Q_0$. The result clearly reveals that the state of syndrome qubit ($Q_0$) is flipped as the input state of data qubits ($Q_1$, $Q_2$) is in an even number of excitations while remaining unchanged if the input state is in an odd excitation number.

Now we turn to a generalization of the JPM scheme to include more data qubits, especially the four data qubits situation motivated by the surface code. In the surface code, every four data qubits are connected with one syndrome qubit, and thus 4-qubit $ZZZZ-(XXXX-)$ parity measurements are fundamental operations. An intuitive way to conduct the weight-4 parity measurement is generalizing previous analysis to four data qubits cases, i.e., collectively coupled four data qubits with the syndrome qubit. However, unlike the 2-qubit case, we cannot find a time $t$ to align all the even (odd) parity oscillations. Nevertheless, the 4-qubit JPM scheme can still be achieved by sequentially implementing 2-qubit JPM operations, as shown in Fig. 2c. More details about the derivation can be found in Supplementary Materials.

Finally, we step further to make a comparison between the JPM scheme and the standard CNOT scheme. On the one hand, the JPM scheme based on the collective iSWAP or CZ operations consumes the same circuit depth; on the other hand, owing to the collective coupling effect shown in Eq. (1), the effective entangling gates in the JPM scheme are usually $\sqrt{2}$ times faster than that of the corresponding CNOT scheme with the homogeneous gate operation type, indicated as the black dotted arrow in Fig. 1d, thus releasing the burden of iSWAP-based CNOT scheme with complex circuit composition. Besides, the iSWAP-based JPM scheme utilized in the following experimental analysis [marked as the red dotted circle in Fig. 1d] does not introduce any additional energy levels, which indicates its consistency with the surface code architecture. These properties may be effectively used to increase the depth of quantum circuits, and thus improving the complexity of compilable algorithms on quantum chips and the number of

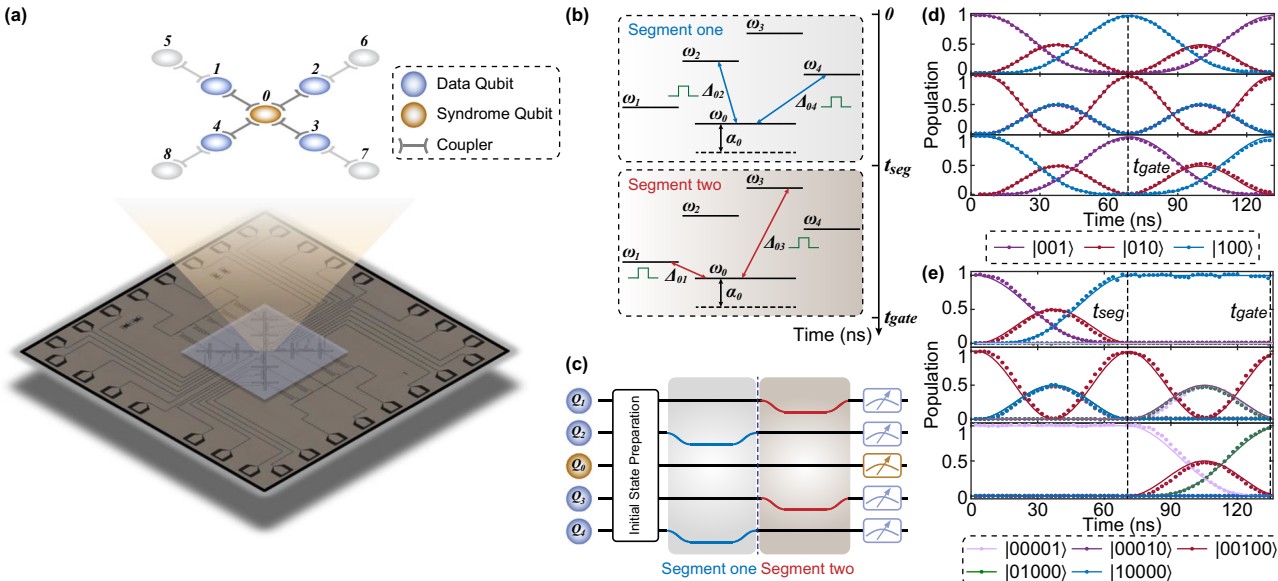

**Fig. 2 | Experimental demonstration of the superconducting quantum processor. a** Optical micrograph of the superconducting quantum processor with nine transmon qubits ($Q_0-Q_8$) and eight tunable couplers. The topological diagram reveals the connectivity and order definition of the qubits. In the experiment, the center five qubits are used as data qubits $Q_1-Q_4$ and syndrome qubit $Q_0$ to verify the JPM scheme. **b** Energy levels in the 4-qubit JPM scheme with sequential JPM unitary gates. Clearly, the gate procedure can be divided into two sequential segments ($t = 0-t_{\text{gate}}$) with the segmentation time $t_{\text{seg}}$ and each segment can be seen as a 2-qubit JPM unitary gate procedure. **c** Pulse sequence for calibrating and implementing the 4-qubit JPM unitary gate (2-qubit JPM unitary gate ignoring one of the segments). The Gaussian-shaped flux pulse is adopted to suppress the leakage error. **d** and **e** Typical experimental calibration results for **d** 2-qubit and **e** 4-qubit iSWAP-based JPM unitary gate with the pulse sequence shown in (**c**). The quantum state is initialized to $|001\rangle$ (top panel), $|010\rangle$ (middle panel), $|100\rangle$ (bottom panel) in (d) and $|00010\rangle$ (top panel), $|00100\rangle$ (middle panel), $|00001\rangle$ (bottom panel) in **e** with the definition as $|Q_2 Q_0 Q_4\rangle$ in (**d**) and $|Q_2 Q_1 Q_0 Q_4 Q_3\rangle$ in (**e**). Through the delicate calibration and the joint calibration method, the population oscillations during the time evolution match the simulation results.

parity checks that can be implemented within the limited qubit coherence time.

## Experimental demonstration

We implement the 2-qubit and 4-qubit JPM scheme on a super-conducting quantum processor, which consists of nine computational frequency-tunable transmon qubits $Q_i$ ($i = 0$–$8$), forming a cross-shaped architecture, with each pair of neighboring qubits mediated by the tunable couplers, as depicted in Fig. 2a[18]. The coupler is a kind of transmon qubit utilized to adjust the coupling strength between adjacent qubits[14,15,22,23]. The inner five qubits on the chip are involved in our experiments, where the center one functions as syndrome qubits $Q_0$, and the surrounding four qubits are data qubits $Q_1$–$Q_4$ (see the "Methods" section).

As mentioned in the section "Theoretical model", we should first calibrate the JPM unitary gate $U_s$, which is critical for constituting a parity measurement circuit. At the initial moment, all the qubits are positioned at the idle frequencies with coupler frequencies adjusted to minimize $ZZ$ crosstalk between adjacent qubits[24–27] (see Supplementary Materials for device parameters). Next, to achieve the collective coupling and realize a reliable and robust JPM unitary gate with high fidelity, two basic requirements are needed: resonant frequencies of all the qubits; and consistent effective coupling strength between each data qubit and syndrome qubit. Since both qubits and couplers are subject to the dispersive shift induced by neighboring qubits or couplers[28], careful treatments should be taken in the frequency calibration to avoid large deviations.

**Calibration of 2-qubit JPM unitary gate.** We select $Q_2$ and $Q_4$ as data qubits and $Q_0$ as a syndrome qubit in our quantum processor to implement a 2-qubit JPM scheme. Figure 2b is the corresponding schematic diagram of the energy levels of the JPM scheme. At the beginning, we confirm the working frequencies of the couplers: Let $Q_2$ and $Q_4$ execute the procedure of basic iSWAP gate with $Q_0$, respectively, and then we decide the coupler frequencies in consideration of the equivalent coupling strength. Next, we finely maintain the coupler flux modulation throughout the subsequent calibration to seek the resonant frequencies of the three qubits simultaneously. Here, we sustain the frequency of $Q_0$ and utilize the Gaussian-shaped flux pulse to shift the frequencies of two data qubits closer to $Q_0$ for suppressing the potential leakage error during the frequency tuning procedure, as depicted in Fig. 2c. Figure 2d clearly reveals the population oscillations with three different initial states after the calibration and shows good consistency with the simulation results. Moreover, the additional single-qubit phase generated during the JPM unitary gate should also be corrected accurately. By initializing the data qubits and syndrome qubit $|Q_2 Q_0 Q_4\rangle$ to the state $|+++\rangle$, we are able to extract the corresponding phase accumulation during the gate procedure. Quantum state tomography (QST) is then implemented to verify the phase calibration with the Pauli measurement and density matrix results in Fig. 3a and b. To finally figure out the fidelity of the JPM unitary gate, we carry out the quantum process tomography (QPT) to extract the experimental $\chi_{exp}$ for verification[24], reaching an average of 95.0% raw gate fidelity in 79 ns. We further optimize the results by eliminating the influence of the state preparation error and the measurement error (SPAM error)[29,30] and acquire an optimized QPT fidelity of 98.5%, as shown in Fig. 3c.

**Extension to 4-qubit JPM scheme with sequential JPM unitary gates.** We now turn to the weight-4 parity measurement which can be typically implemented in the surface code. Here, data qubits are extended to four qubits $Q_1$–$Q_4$. Following the theoretical model and similar calibration procedure, the 4-qubit JPM unitary gate can be viewed as the sequential execution of two 2-qubit JPM unitary gates. For the first stage, we need to ensure the three qubits, for instance, $Q_0$,

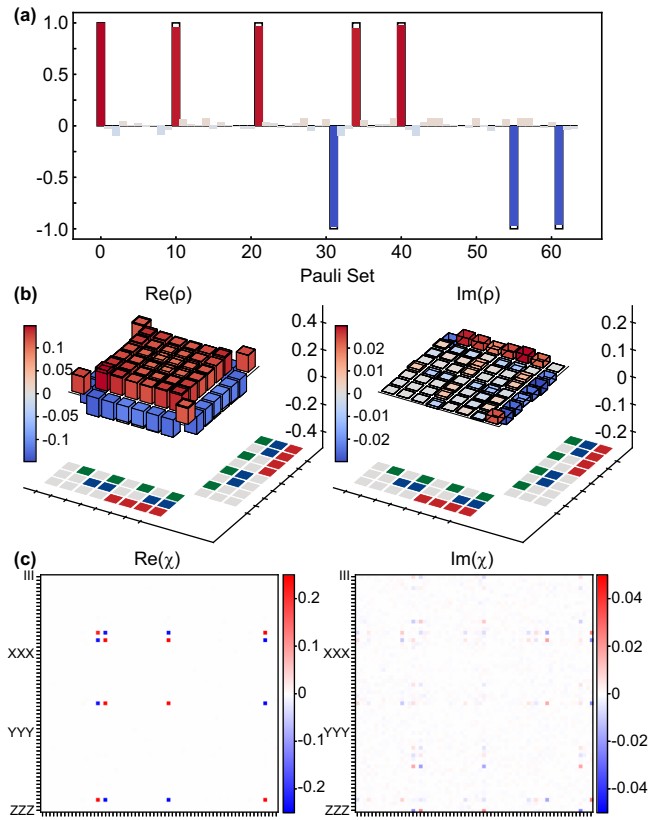

**Fig. 3 | Calibration of the 2-qubit iSWAP-based JPM scheme. a** The expectation of the Pauli set measured in a 2-qubit iSWAP-based JPM scheme with the initial states $|+++\rangle$. Here, the numbers on the horizontal axis represent the order of the Pauli set in terms of $I, X, Y, Z$ for each qubit. For instance, the first Pauli operator is $III$, and the last one is $ZZZ$. **b** Phase accumulation during the JPM unitary gate is extracted via the quantum state tomography (QST) with the initial states $|+++\rangle$. **c** Quantum process tomography (QPT) of the 2-qubit JPM unitary gate characterizing the experimental $\chi_{exp}$, reaching an average 95.0% raw gate fidelity in 79 ns. We further extract the optimized QPT data with 98.5% gate fidelity after eliminating the SPAM error.

$Q_2$ and $Q_4$, achieve full resonance and equivalent coupling strength ($g_2 = g_4 = g_{seg_1}$) as well to conduct the 2-qubit JPM unitary gate. After the first segment at $t_{seg}$, as shown in Fig. 2c, we quickly return $Q_2$ and $Q_4$ to the idle position and adjust $Q_1$ and $Q_3$ at the same time to go through a similar process with the coupling strength $g_1 = g_3 = g_{seg_2}$. After the end of the second segment at $t_{gate}$, $Q_0$, $Q_1$ and $Q_3$ are quickly adjusted to return to the idle position to finish the whole gate procedure. Figure 2e is the typical population oscillations after calibrating 4-qubit JPM unitary gate with three different initial states and the pulse sequence shown in Fig. 2c. In practice, here $t_{seg}$ is not required to be half of the gate time $t_{gate}$ since the coupling strength $g_{seg_i}$ ($i = 1, 2$) in both segments do not need to be seriously the same. Similarly, we execute QST to extract the single-qubit phase accumulation of all the qubits and evaluate the state fidelity as a whole with the initial state $|+++++\rangle$ (see Supplementary Materials).

**Experimental comparison with the CNOT scheme.** The experimental realizations of the weight-2 and weight-4 parity measurements are performed under the circuits shown in Fig. 4a. Figure 4b and c are the corresponding parity detection results in both the measure-Z and measure-X procedures. We find that compared to the 92.9% (87.9%) for the CZ-based CNOT scheme with the entire circuit time of 220 ns (365 ns) in the measure-Z and measure-X procedures, the average parity detection fidelity in weight-2 (weight-4) parity measurement of

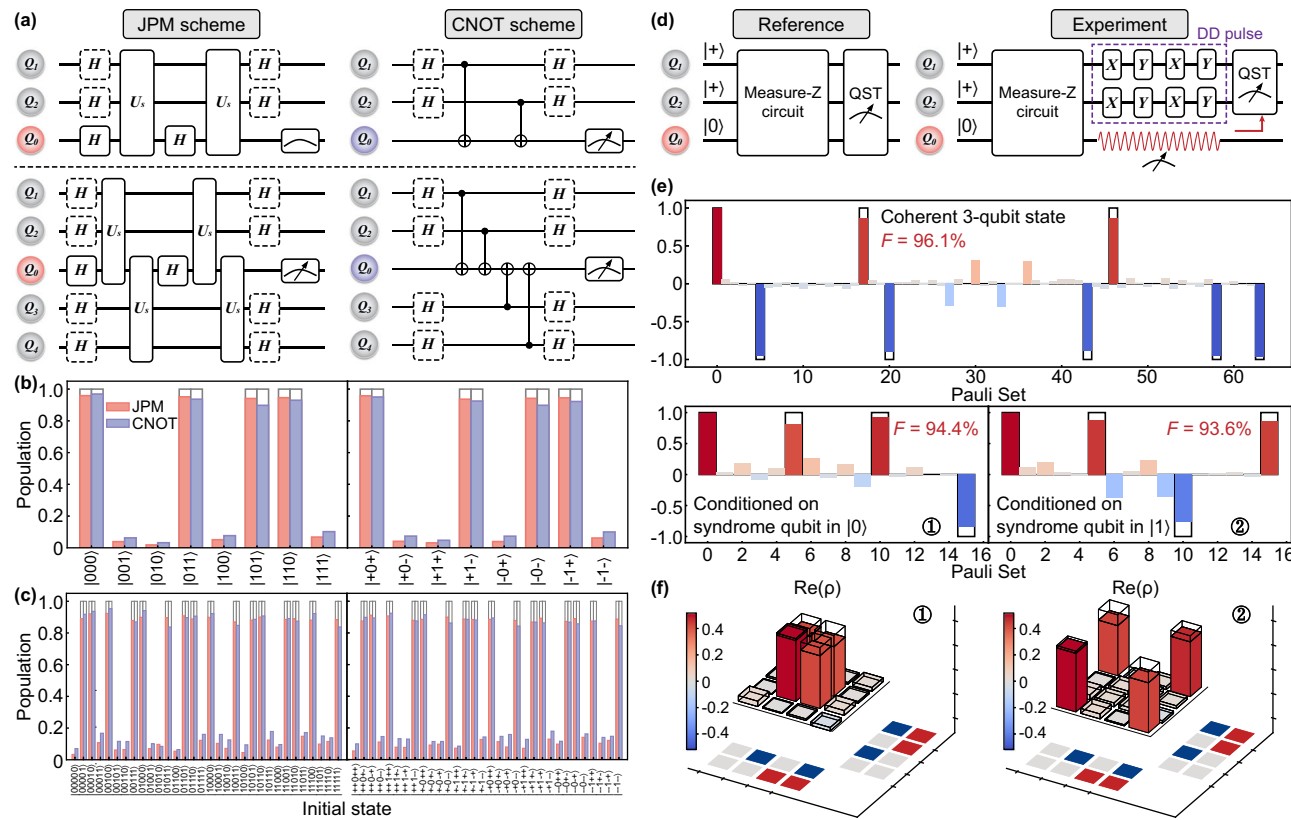

**Fig. 4 | Parity measurement and disturbance effects. a** Parity measurement circuit for the JPM scheme (left) and the CNOT scheme (right) in weight-2 (upper panel) and weight-4 (bottom panel) parity measurement. Here, the circuit, including the solid-box framed Hadamard gates, represents the measure-Z procedure, and the dotted-box framed Hadamard gates will be involved in the measure-X procedure. In particular, the circuit of measure-X procedure in the CNOT scheme can be further simplified (see Supplementary Materials). **b** Weight-2 and **c** weight-4 parity measurement in the iSWAP-based JPM scheme and the CZ-based CNOT scheme. Left panel: measure-Z procedure. Right panel: measure-X procedure. The average parity detection fidelity of the JPM scheme [**b** measure-Z: 95.3%; measure-X: 95.1%; total: 95.2% and **c** measure-Z: 90.4%; measure-X: 89.5%; total: 90.0%] is on par with the CNOT scheme [**b** measure-Z: 93.3%; measure-X: 92.5%; total: 92.9% and **c** measure-Z: 88.6%; measure-X: 87.2%; total: 87.9%] in the experiment. Notice that

here the CZ gate time is set to be close to the gate time of the 2-qubit JPM unitary gate for accurate comparison. **d** Circuit diagrams for verifying disturbance effects on data-qubit subspace based on the JPM scheme with the reference group (experimental group) shown in left (right) panel. The Dynamical Decoupling (DD) pulses are utilized with optimized pulse intervals to mitigate dephasing effect during the measurement of syndrome qubit. **e** Measured expectation values of multiqubit Pauli operators for the reference group (top panel) and the experimental group conditioned on the syndrome qubit in $|0\rangle$ state and $|1\rangle$ state (bottom panel). Here, the numbers on the horizontal axis represent the order of the Pauli set in terms of $I, X, Y, Z$ for each qubit. (f) The corresponding density matrix of Bell states $|\Phi_+\rangle = (|01\rangle + |10\rangle)/\sqrt{2}$ and $|\Psi_+\rangle = (|00\rangle + |11\rangle)/\sqrt{2}$ measured in the experimental group conditioned on the syndrome qubit in $|0\rangle$ state and $|1\rangle$ state.

16 (64) standard initial states can be achieved comparable at around 95.2% (90.0%) for the iSWAP-based JPM scheme with the entire circuit time of 238 ns (396 ns) in the measure-Z procedure and 278 ns (436 ns) in the measure-X procedure.

**Disturbance effects on the data-qubit subspace.** Furthermore, we investigate the impact of parity measurement on the data-qubit subspace in the JPM scheme by experimentally analyzing circuit diagrams shown in Fig. 4d, which are divided into a reference group (left panel) and an experimental group (right panel). In the reference group, both data qubits and syndrome qubits are measured simultaneously after passing through the measure-Z circuit. Instead, in the experimental group, syndrome qubit is measured first [indicated as a red waveform in Fig. 4d], followed by QST measurement of data qubits. To mitigate the dephasing influence on data qubits during syndrome qubit measurement, Dynamical Decoupling (DD) pulses with pulse optimization obtained by varying pulse intervals are employed[31,32]. The only difference between these two groups lies in their order of measuring data qubits and syndrome qubits, allowing us to confirm the disturbance effects of the JPM scheme on the data-qubit subspace through fidelity comparison. The results obtained from QST measurements are presented in Fig. 4e, where the top panel illustrates the coherent state of

the three qubits in the reference group, and the bottom panel characterizes the state of two data qubits conditioned on syndrome qubit measurement outcomes. By projecting the data qubits into the corresponding Bell states $|\Phi_+\rangle = (|01\rangle + |10\rangle)/\sqrt{2}$ and $|\Psi_+\rangle = (|00\rangle + |11\rangle)/\sqrt{2}$, respectively, for both syndrome qubit measurement outcomes $|0\rangle$ and $|1\rangle$ in Fig. 4f, we achieve fidelities (94.4% and 93.6%) that closely match those obtained by projecting the reconstructed three-qubit state onto its two-qubit subspace (95.8% and 96.4%). This level of agreement aligns well with the readout fidelity (~97.0%) of the syndrome qubit, indicating that parity measurement based on the proposed JPM scheme should have minimal impact on data qubits[31].

## Discussions

In this section, we extend our JPM scheme to discuss more feasible applications: potential parity measurement procedure in the surface code architecture and the generation of specific initial states, especially multiqubit entangled states for non-adjacent qubits.

Figure 5 shows a potential schematic representation of the JPM scheme applied in the surface code. It can be found that our scheme may match the current popular coupler-based surface code architecture without adding additional devices to the chip. When parity

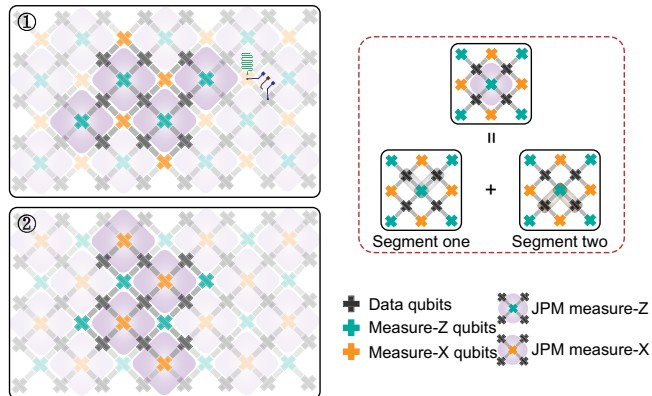

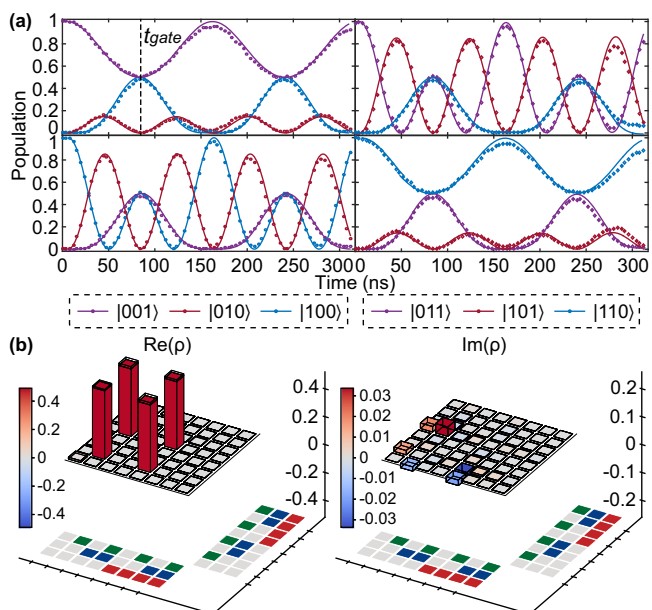

**Fig. 5 | Potential schematic application of the JPM scheme in the surface code.** The cross-shaped pattern represents the qubits (constructed by the superconducting qubits like transmon[18], c-shunt flux qubit[45–47], fluxonium[48,49] and so on) with the data qubits shown in black, measure-Z syndrome qubits shown in cyan and measure-X syndrome qubits shown in orange. The gray rectangular between adjacent qubits represent the tunable couplers, and the green curves are the readout cavity for qubits' states to be dispersively measured. $Z$ line for the coupler flux modulation is displayed as the brown curve, and the dark blue one is on behalf of the $XYZ$ line for the qubit manipulation and flux modulation.

**(a)**

**(b)**    Re($\rho$)                    Im($\rho$)

**Fig. 6 | Experimental realization of the JEP scheme. a** The experimental population results in the 2-qubit JEP scheme with initial states $|Q_2Q_0Q_4\rangle$ prepared to be $|001\rangle$ (upper left), $|011\rangle$ (upper right), $|100\rangle$ (bottom left) and $|110\rangle$ (bottom right). At a specific gate time (86 ns), the distant Bell state is generated. **b** QST measurement of the distant Bell state of $Q_2$ and $Q_4$ at the gate time with $Q_0$ initialized to be in the ground state.

measurement (measure-Z or measure-X) is required, syndrome qubits and surrounding data qubits start to shift their frequencies into resonance. At the same time, the adjacent couplers are adjusted to maintain the coupling strength. When error detection is performed, all measure-Z (measure-X) syndrome qubits can be executed at the same time based on the sequential JPM unitary gate. Therefore, it is not necessary to wait for the unitary gate to be finished to interact with the neighboring syndrome qubits, which may be beneficial for reducing the circuit time. Meanwhile, we should also mention that more research, such as leakage and error propagation in the JPM scheme

deserves to be explored to support the experimental feasibility of the surface code architecture.

A slight modification of the JPM scheme can be further used for specific joint entangling state preparation (JEP scheme). Keeping the qubit frequencies in resonance while adjusting the coupling strength $g_1/g_2$ from $\tan(\frac{\pi}{4})$ (JPM scheme) to $\tan(\frac{\pi}{8})$, we can get the population oscillation, as shown in Fig. 6a, with different initial states under the 2-qubit JEP scheme. Clearly, at specific gate time $t_{JEP} = \pi/\sqrt{(g_1^2 + g_2^2)}$ (faster than the standard method by iSWAP gates or CZ gates[30,33,34]), distant multiqubit entangled Bell state $\psi = \frac{1}{\sqrt{2}}(|00\rangle + |11\rangle)$ for $Q_2$ and $Q_4$ are directly generated as depicted in Fig. 6b, with the extracted average 98.6% state fidelity by performing QST measurement. We also expect further exploration with the pulse combination to open up more possibilities in quantum computation and quantum simulation, e.g., the extension to a 4-qubit case with the modification $g_1/g_2 = g_3/g_4 = \tan(\frac{\pi}{8})$ may be utilized to generate multiqubit states (such as Dicke states).

In summary, we theoretically analyze and experimentally realize a fast joint parity measurement scheme inspired by the stimulated emission, which utilizes the collective behavior between data qubits and syndrome qubits, and, thus, is $\sqrt{2}$ times sped up over the commonly-used CNOT scheme for the homogeneous gate operation. We verify the strategy based on our superconducting quantum processor with tunable couplers. Taking advantage of the frequency-tunable qubits and couplers, the JPM scheme can be easily attained with delicate calibration and careful optimization. Through comparison with the CNOT scheme, we find that the JPM scheme is well performed with an average 95.2% (90.0%) parity detection fidelity for weight-2 (weight-4) parity measurement. Meanwhile, a slight modification of the JPM scheme further supports the JEP scheme with high fidelity. Our results, together with the extended applications in the surface code architecture and entangling state preparation, manifest the robustness and potential of the JPM scheme in the quantum simulation and quantum error correction.

## Methods
### Sample design and device fabrication
Our quantum processor consists of nine transmon qubits with eight tunable couplers. The computational qubits form a cross-shaped architecture where $Q_0$ is positioned in the middle, and $Q_1$–$Q_4$ ($Q_5$–$Q_8$) are distributed in the inner (outer) ring. Each computational qubit has an individual $XY$ line for qubit manipulation and a $Z$ line for frequency adjustment. Meanwhile, each couplers are equipped with the $Z$ line to tune frequencies for qubit-qubit coupling strength manipulation. The distinguishment of the qubit states is realized with the individual readout cavities based on the dispersive measurement[35].

The device was fabricated on a 430 μm-thick sapphire substrate with 100 nm aluminum. The base circuit was patterned with a developed high-temperature etching method, followed by an optical lithography with the DWL 66⁺ Heidelberg instrument. A Manhattan-style Josephson junction[36] was patterned by the electron beam lithography. After that, the chip was sent to a double-angle evaporation instrument for deposition. A static oxidation was used to form the barrier layer of the Josephson junction, and finally, the Josephson junctions were lifted off in a Remover PG solution. To suppress the crosstalk on the chip and acquire a better ground connection, the airbridge was designed on the lines and the readout cavities, fabricated with a photoresist scaffold[37]. After all steps were done, the chip was further cleaned with a UV Ozone for 3 min to remove the potential organic residual.

### Measurement setup
The superconducting quantum processor is mounted in an aluminum sample holder at a base temperature of -10 mK in the dilution refrigerator. To protect the qubits from the flux noise and quasiparticle, we use one aluminum shielding, one magnetic shielding and one infrared shielding. The fundamental experimental setup is depicted in Fig. S5 in

Supplementary Materials. The generation of the *XY* signal is realized with the IQ mixing process, while the *Z* modulation is implemented directly through the fast analog signal generated by the self-developed arbitrary waveform generators (AWGs)[38]. In addition, the readout signal is amplified with the Josephson junction parametric amplifier (JPA)[39], combined with a high-electron-mobility transistor amplifier at 4 K and room-temperature amplifiers, allowing for all the qubits to achieve simultaneous single-shot readout.

## Simulations

The JPM scheme demonstrates the importance of the tunable coupler based on the surface code architecture. On the one hand, static ZZ coupling can be effectively suppressed which is beneficial to both single-qubit gate and multiqubit gate operations[14]. On the other hand, the tunability of the coupling makes the implementation of the JPM scheme feasible. In fact, we can further take advantage of the couplers to realize the rapid preparation of specific initial states like multiqubit entangling states, as mentioned in the "Discussions" section. Multiqubit entangling states, such as GHZ state, W state, and Dicke state, are essential in quantum computation, quantum error correction, and quantum communication[17,40,41]. For example. the Dicke state can be widely used in quantum teleportation and some parity detection protocols[42,43]. Moreover, the preparation of entangling states has been studied for several years in different quantum systems. Realization of high-fidelity, fast preparation of entangling states has always been a key task.

Our simulations take advantage of the Qutip, a Python-based simulation package[44], to completely acquire the time evolution of the JPM scheme. Typically, two potential simulation models are adopted. The first one is the ideal model based on the Hamiltonian Eq. (1), which directly reveals the expected results and helps us clearly understand the theoretical proposal of the JPM scheme. When it comes to the more complicated real experimental process, a more comprehensive Hamiltonian model containing both the qubits and couplers is adopted

$$
\begin{aligned}
H = & \sum_{i=0-4}\left(\omega_i a_i^\dagger a_i + \frac{\alpha_i}{2} a_i^\dagger a_i^\dagger a_i a_i\right) \\
& + \sum_{i=1-4}\left(\omega_{c,i} a_{c,i}^\dagger a_{c,i} + \frac{\alpha_{c,i}}{2} a_{c,i}^\dagger a_{c,i}^\dagger a_{c,i} a_{c,i}\right) \\
& + \sum_{i=1-4} g_{iL}\left(a_0 + a_0^\dagger\right)\left(a_{c,i} + a_{c,i}^\dagger\right) \\
& + \sum_{i=1-4} g_{iR}\left(a_i + a_i^\dagger\right)\left(a_{c,i} + a_{c,i}^\dagger\right) + g_{id},
\end{aligned}
\tag{6}
$$

where $\omega_{c,i}$ represents the bare frequency of the coupler $C_{0i}$ ($i = 1-4$); $a_{c,i}, a_{c,i}^\dagger$ ($i = 1-4$) are the annihilation and creation operators for couplers; $g_{iL}$, $g_{iR}$, $g_{id}$ ($i = 1-4$) are coupling strength for $Q_0$ and $C_{0i}$ ($i = 1-4$), $Q_i$ and $C_{0i}$ ($i = 1-4$), $Q_0$ and $Q_i$ ($i = 1-4$). Besides, in order to replicate the experimental conditions as closely as possible, we implement the pulse and calibration procedures that are essentially similar to the experiment to simulate the expected experimental results.

## Data availability

The data that support the findings of this study are available from the corresponding authors X.G. and T.Q.C. upon request.

## Code availability

The code that supports the simulations of this study is available from the corresponding authors X.G. and T.Q.C. upon request.

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

## Acknowledgements

We thank Mengyu Zhang, Fuming Liu, Guanglei Xi, Qiaonian Yu, and Hualiang Zhang for supporting room-temperature electronics. We thank Jingjing Hu and Shuoming An for providing Josephson parametric amplifiers. We thank Yuwei Ma for the valuable discussions.

## Author contributions

X.G. developed the JPM theory and conceived the experiment. T.Q.C. conducted the JEP analysis. S.N.H., T.Q.C., Z.X.Z. established the measurement setup. K.L.B. fabricated the devices. S.N.H., K.L.B. performed experimental measurements with assistance of T.Q.C., Z.X.Z., S.M.A., X.P.Y., Y.L. and Y.C.Z. T.Q.C., S.N.H., K.L.B., X.G. wrote the manuscript with feedback from all authors. T.Q.C., Y.C.Z. supervised the project. All authors contributed to the discussion of the results and the development of the manuscript.

## Competing interests

The authors declare no competing interests.
