## [Peer Review File · Nature Communications]

Fast joint parity measurement via collective interactions induced by stimulated emissionReviewer #1 (Remarks to the Author):

The authors have demonstrated a novel scheme for realizing the joint parity measurement (JPM) that is compatible with the surface code error correction scheme. With the tunability allowed in the coupler-equipped superconducting circuit platform, they show in experiment that by tuning the frequency of the two data qubits into resonance with the syndrome qubit while keeping the coupling strength same, the joint parity can be extracted easily. This scheme can be extended to 4-qubit JPM by sequentially implementing 2-qubit JPM.

This new scheme has a short gate time. From the theory analysis, they show that their scheme is usually $\sqrt{2}$ times faster than the C-NOT based scheme. Benefiting from the shorter gate time, their scheme also achieves higher fidelity compared to C-NOT based scheme. This exploration of the collective interaction in quantum information techniques is interesting, and the experimental techniques shown in this work for tuning up to nine qubits to meet the stringent coupling condition are quite impressive.

Overall, I can recommend the publication after the following issues are properly addressed.

1. The current title may not cover the content clearly; I suggest the authors to consider the replacement of "stimulated emission" by "collective interactions".
2. In the last line of the caption in Fig1, the expression for $|\psi\rangle$ is inconsistent with the one in line 155.
3. I find Fig. 1 in the main text is quite confusing at the first glance without understanding the theory described in the text. I suggest the authors to make it easier to understand.
4. Line 160: "map the even (odd) parity of the data qubits onto the state 0 (1)", from the context, I think it should read "map the even (odd) parity of the data qubits onto the state 1 (0)".
5. Since the authors aim to apply the detection scheme into the quantum error correction codes, it is worth to have discussions on whether syndrome qubit errors will propagate to data qubits, which is crucial for the fault-tolerance of error syndrome detections.
6. In lines 188-189, the authors claim that they can't find a time t to align all the even (odd) parity oscillation for the 4-qubit scheme. I am interested to see if you can find the time t for n -qubit scheme? Or does this only work for the 2-qubit scheme?
7. In line 201 and line 403, the author mentioned the $\sqrt{2}$ speed up over the traditional CNOT measurement scheme. I am confused with such a claim since no detailed condition is provided. Usually a CZ based CNOT just requires one CZ and two Hadamard gates, where the CZ operation time is on par with the JPM unitary mentioned here, I cannot see the speed up other than the constant time reduction in single-qubit operations.
8. Can authors elaborate more specifically for the near 2 percent fidelity improvement in the iSWAP based parity measurement than the CZ based CNOT scheme? Is it because the CZ fidelity is not high enough compared with the JPM unitary?
9. The term "maximally entangled state" mentioned in the manuscript is closely relevant with the entanglement measures and is not commonly acknowledged for general multiqubit states. I suggest the authors to replace the expression to "multi-qubit entangled state" for rigorousness.
10. The authors claim the preparation of a four-qubit entangled state, it would be beneficial to state explicitly what the target entangled state is if an entanglement measure is unavailable.

Reviewer #2 (Remarks to the Author):

The authors present an interesting alternative scheme to measure joint parity of states in a setting very actively used in the superconducting circuit community, namely surface-code quantum error correction.

I found the concept very neat, using joint evolution (and the difference in rates of interaction coming from the "+1" in the stimulated emission of the bosonic operators) to distinguish odd from even parity

states. In a way, this (and especially the population plots of Fig7(a)) reminded me of trapped ion MS gates, maybe there is a parallel to be made?

My main concerns are two-fold:

1.

The authors make a strong point in comparing their method to that of a CNOT-based scheme (decomposed into CZ gates). On the one hand, while the comparison is interesting, I don't see the emphasis on better performance as necessary for the message of the paper. Indeed, this new scheme's working principle is interesting no matter what, as long as fidelities are comparable, which seems to be the case here. On the other hand, I cannot follow the arguments being made. The CZ-based CNOT scheme has multiple possible implementations (especially when given a tuneable coupler), and its standard interaction time is also $\pi/\sqrt{2}g$ from working in the second excitation manifold (as here, line 163). Where is the gain? Sure, one needs to apply two (four) CZ for a weight-2(4) parity check, but isn't that the same for the unitary U described here (line 171 and fig4)?

The duration of the total protocol in both cases is not mentioned explicitly (only for the JPM scheme, lines 314-316), but the CZ was made comparable in time to the unitary gate (fig4, caption). Where was the claimed speed gain lost? Are the authors able to stand behind the statements in line 84-87?

2.

I found the clarity in the presentation of the manuscript lacking:

- the theoretical model section is very heavy, and doesn't provide much insights into the physics happening (which I believe should be relatively easy to describe, as common hybridized modes of the system which undergo oscillations based on the total parity?)
- Many figures have texts that are too small to read (1a, 1b, 2b, 8, ...).
- It is unclear what one should look at in Table Fig.1c (is Ψ given at the interaction time? what do colors depict? ...).
- Fig 1d, 1e are lacking initial state descriptions, the color labeling is not obvious.
- Fig 7a, 7c don't have axis labels.
- Fig 8 needs proper stages temperature, instead of a vague gradient.
- is Fig2c really a pulse sequence for the JPM scheme? Or only for the JPM unitary, without undoing the additional minus phase?
- line 295-296: "Figure 2(e) is the typical population oscillations in 4-qubit JPM scheme". See above point: Is it really? Or some population evolution given specific calibration input states for the JPM unitary gate.
- I really hope the authors don't "optimize the results" of the experimental chi matrix to quote a high fidelity, but rather compute the chi matrix that represents the process happening on the device, taking SPAM errors into account. (lines 272-274 and Fig3 caption).
- lines 389-390: clearly populations plots can't convince that a Bell state is "generated no matter what states Q0 is in" (coherences are missing). While I can follow the expectation, the experimental evidence is lacking.
- Fig 5 is barely referenced, and unless a proper discussion of the infidelity is made, it should just be removed, or placed in the supplementary material files.

Reviewer #3 (Remarks to the Author):

The manuscript by Huai, Bu, Gu et al. focuses on the implementation of joint parity measurements (JPM), operations designed to map the parity information of multiple data qubits to the state of a syndrome qubit. The primary result is the experimental implementation of a fast, joint parity measurement scheme based on stimulated emission that requires a superconducting quantum processor architecture with tuneable couplers. This interaction is achieved by controlling the collective behavior of multiple data qubits, ensuring resonance and consistent effective coupling strengths between all pairs involved, and benchmarked using quantum state tomography (QST) and quantum

process tomography (QPT). Here, it is worth highlighting the strong emphasis placed on the use of this scheme for surface-code based quantum error correction applications, something that serves as motivation to the authors throughout the manuscript and generates multiple comparisons, despite the simultaneous demonstration of a joint entangling state preparation (JEP) scheme.

The main advantage of the scheme is stated to be a $\sqrt{2}$ speed up over traditional CNOT-gate based parity measurement schemes, which commonly rely on two-qubit control-Z (CZ) gates to build comparable parity extraction circuits, despite a significant increase in the complexity of calibration and quantum control required for the proposed implementation. The paper is fairly well written and makes the topic accessible. I find the work to be scientifically sound. However, the limited breath of benchmarking presented for the scheme, particularly in assessing the effect of correlated error sources, such as leakage, in a scheme where error propagation would be a source of particular concern, prevent my immediate recommendation for publication in Nature Communications. Nevertheless, I would like to give the authors the opportunity to further clarify various statements made in the text, and further motivate the novelty and usefulness of the proposed scheme. As such, I would greatly appreciate the authors address the following points:

1. In the manuscript, it is stated that the proposed scheme “requires no additional device modifications or circuit complexity”. I believe this statement greatly oversimplifies the cost of such a scheme, given its requirement on tunable coupling elements, and particularly because of the additional complexity in calibration. Considering the stringent requirements on calibration, including the need to ensure resonance between all qubits involved (when all qubits are subject to dispersive shifts), and the additional complexity in quantum control sequences, required to maintain consistent effective coupling strengths in adjacent couplers, do the authors believe their original statement should stand? Moreover, could the authors comment on the benefits of the theoretically achievable $\sqrt{2}$ speed up over traditional CNOT-based measurement schemes, particularly considering the additional complexity introduced?
2. In Figure 2 (d) and (e), the lack of a legend makes it hard to interpret the results. Could the various curves be more clearly labelled, perhaps in the style of Figure 7 (a) and (c)?
3. When assessing parity measurement schemes, it is crucial to quantify the effect of backaction on the data-qubit subspace, since this would be exacerbated in any real-world use case, given the cyclic nature of parity checks in stabilizer-based error correction codes. However, it could be hard to quantify such effects based purely on the results of QST and/or QPT presented. As such, can the authors comment on the repeatability and disturbance effects of the JPM scheme proposed?
4. When comparing JPM to CNOT-based parity measurement schemes, the authors understandably place a lot of emphasis on average parity detection fidelity. However, given the outsized influence of correlated errors in the overall performance of a stabilizer-based error correction code, I would argue that quantifying the effect of correlated errors, particularly of leakage, is just as important, since the overall performance might otherwise be limited despite the higher parity detection fidelity. Could the authors comment on the effects of JPM on correlated error sources such as leakage?
5. Moreover, when comparing JPM to CNOT-based parity measurement schemes, the authors state that “the CZ gate time is set to be close to the gate time of the 2-qubit JPM unitary gate, for accurate comparison”. However, based on the description of the JPM unitary gate and the device parameters presented, it appears that a 2-qubit non-adiabatic CZ gate, fully exploiting leakage interference to approach the speed limit, could potentially perform faster than the JPM scheme (owing to the need to undo the minus phase in the Dicke states for the latter). Could the authors comment on the durations used in this manuscript for 2-qubit CZ gates and their respective speed limits?
6. Considering that the proposed JPM scheme effectively corresponds to a three-qubit entangling gate, it should be expected that error propagation, particularly within the parity-measurement cycle, be

more problematic than in the CNOT-based scheme. Could the authors comment on the effect of their proposed scheme on error propagation? Can parity check measurements implemented using this scheme be made fully fault tolerant?

7. In motivating the merit of the proposed scheme, a comparison with a CNOT-based parity measurement scheme features prominently. In the manuscript, the authors attribute the main source of error in the JPM scheme to qubit decoherence but, at the same time, state that the CZ gate time is set to be close to the gate time of the 2-qubit JPM unitary gate. As such, decoherence alone does not seem to capture the difference in performance between the two gates. Given the important role of this comparison in motivating the JPM scheme, it would be important for the authors to more thoroughly study the error channels currently limiting the performance of their CNOT scheme. Do the authors anticipate that further optimization could bring the performance of the CNOT-based scheme closer to that of the JPM scheme? Could the authors comment on the reason why error channels limiting the performance of the CNOT-based scheme do not equally affect the performance of the JPM scheme?

Authors' Response to the Review Comments

Journal: *Nature Communications*

Manuscript ID: *NCOMMS-23-25497*

Title of Paper: *Fast joint parity measurement via collective interactions induced by stimulated emission*

Date Sent: *February, 2024*

We appreciate the time and efforts by the editor and referees in reviewing this manuscript, and thank the referees for providing insightful comments. We address the referee's comments as follows: (here we first quote the referees' comments in black color and then provide our response after each point in blue)

Summary of major changes

All the changes are marked in red in the revised manuscript and here we list major changes:

1. The title of the revised manuscript has been changed to "*Fast joint parity measurement via collective interactions induced by stimulated emission*".
2. We reconstruct THEORETICAL MODEL section and Fig. 1 to make them more concise and easy to grasp the key points of our joint parity measurement (JPM) scheme. Besides, time consumption of both JPM scheme and CNOT scheme is added in Fig. 1(d) for explicit comparison.
3. We add subsection "*Disturbance effects on the data-qubit subspace*" in EXPERIMENTAL DEMONSTRATION and Fig. 4(d)-(f) to further investigate the impact of parity measurement on the data-qubit space in the JPM scheme via experiments.
4. We remove ERROR ANALYSES section and Fig. 5 in the original main text. We add ERROR ANALYSES section with subsections "*Error sources in the transversal parity measurement circuit*", "*Error sources in the longitudinal parity measurement circuit*" and Fig. S6, Fig. S7, Fig. S8 in the Supplementary Materials. These additions allow us to fully explore the error sources in the JPM scheme.

Response to Reviewer #1

Overall comments

The authors have demonstrated a novel scheme for realizing the joint parity measurement (JPM) that is compatible with the surface code error correction scheme. With the tunability allowed in the coupler-equipped superconducting circuit platform, they show in experiment that by tuning the frequency of the two data qubits into resonance with the syndrome qubit while keeping the coupling strength same, the joint parity can be extracted easily. This scheme can be extended to 4-qubit JPM by sequentially implementing 2-qubit JPM.

This new scheme has a short gate time. From the theory analysis, they show that their scheme is usually $\sqrt{2}$ times faster than the C-NOT based scheme. Benefiting from the shorter gate time, their scheme also achieves higher fidelity compared to C-NOT based scheme. This exploration of the collective interaction in quantum information techniques is interesting, and the experimental techniques shown in this work for tuning up to nine qubits to meet the stringent coupling condition are quite impressive.

Overall, I can recommend the publication after the following issues are properly addressed.

Response:

We thank the referee for his/her time and efforts in reviewing our manuscript. We are pleased to know that the referee recognizes this work as "*This exploration of the collective interaction in quantum information techniques is interesting, and the experimental techniques shown in this work for tuning up to nine qubits to meet the stringent coupling condition are quite impressive.*" In particular, we appreciate that the referee points out "*Overall, I can recommend the publication after the following issues are properly addressed.*" We follow the referee's insightful suggestions and make major revisions in the revised manuscript to address the raised issues.

Comment 1

The current title may not cover the content clearly; I suggest the authors to consider the replacement of "stimulated emission" by "collective interactions".

Response:

We thank the referee for his/her valuable suggestion. We change the title to "Fast joint parity measurement via collective interactions induced by stimulated emission" considering the suggestion of the referee and the main content of the paper. As the referee mentioned, the implementation method of the JPM scheme relies on the collective interaction between the syndrome and data qubits. However, it is important to note that relying solely on collective interaction may not fully capture the specific physical mechanisms, such as dispersive or resonant coupling. Our purpose is to present a parity measurement scheme

that utilizes the resonant coupling method in a way similar to stimulated emission. Specifically, the strengths of exciting the syndrome qubit g_+ (which removes one excitation from the data qubits) and that of de-exciting the syndrome qubit g_- (which adds one excitation to the data qubits) will be different, depending on the total excitation in the data qubits. By identifying a suitable time to align the different oscillations, we can establish a correlation between the even (odd) number of excitations and the state of the syndrome qubit, which is either 1 or 0. Additionally, as stimulated emission is a widely recognized phenomenon in the field of quantum optics, we have revised the title of the article to encompass both aspects and better reflect the content of the paper, making it more accessible to readers from different fields.

Comment 2

In the last line of the caption in Fig.1, the expression for $|\psi\rangle$ is inconsistent with the one in line 155.

Response:

We thank the referee for his/her careful check. Following the suggestion of *Reviewer #2* in *Comment 2-1*, we have revised THEORETICAL MODEL section and Fig. 1 to better clarify our JPM scheme, therefore the expression for $|\psi\rangle$ in the caption of Fig. 1 in the original manuscript has been deleted, while now appears in line 172 in the revised manuscript.

Comment 3

I find Fig. 1 in the main text is quite confusing at the first glance without understanding the theory described in the text. I suggest the authors to make it easier to understand.

Response:

We thank the referee for his/her valuable suggestions and we notice that the *Reviewer #2* also mentioned the lack of clarity and intuitiveness of Fig. 1(c) in *Comment 2-3*. Therefore, in our revised manuscript, we redraw Fig. 1 and rewrite the section of THEORETICAL MODEL to make the Fig. 1 (depicted in Fig. R1-1 here for convenience) and theory of JPM scheme easier to understand.

Firstly, we rearrange the order of Fig. R1-1(a) and (b). In Fig. R1-1(a), we present the whole circuit diagram of the JPM scheme for parity measurement to give readers an intuitive understanding, and then provide detailed explanations for some intermediate steps in (b) and (c). Specifically, we label each stage in the JPM circuit diagram of Fig. R1-1(a) with i, ii, iii, iv. Without loss of generality, we take $|0, \psi^k\rangle$ (in the order of $|Q_0, Q_1, Q_2\rangle$) as the initial state, and label the intermediate state $|\phi\rangle$ after state iii. Then, similar to the stimulated emission, in Fig. R1-1(b), we illustrate the evolution of the qubits at stage ii. Depending on the state of syndrome qubit ($|0\rangle$ or $|1\rangle$), the data qubits ($|\psi^k\rangle$) experience different oscillations, denoted by $|0\rangle|\psi^k\rangle \leftrightarrow |1\rangle|\psi^{k-1}\rangle$ and $|1\rangle|\psi^k\rangle \leftrightarrow |0\rangle|\psi^{k+1}\rangle$ with different transition rates g_-^k and g_+^k , respectively. In the table of Fig. R1-1(c), we detail the transition rates for different Dicke states $|\psi^k\rangle$ of the data qubits. For the situation where the excitation number of the data

qubits is in even (odd) parity, we fill it with blue (red) background. At the bottom row of the table, we present the intermediate state $|\phi\rangle$ of the qubit after the third stage iii. And to eliminate the accumulated π phase appeared in $|\phi\rangle$ for the input state $|0, D_1\rangle$ and $|0, D_2\rangle$, we introduce another U_s in state iv. Moreover, in order to illustrate the time consumption of the CNOT and JPM parity measurement scheme more clearly, we add Fig. R1-1(d) here. We mark the demonstration schemes used in the experiment, i.e., the CZ-based CNOT scheme and iSWAP-based JPM scheme, with red dashed circles, while indicating the corresponding time acceleration with a black dotted arrow. We also make detailed modifications to the captions of this figure and the corresponding description in the main text. We believe that these modifications will enable readers to quickly grasp the essential information conveyed in Fig. R1-1.

FIG. R1-1. (a) Joint parity measurement protocol for two qubits (Q_1, Q_2), where the syndrome qubit (Q_0) is initialized in state $|0\rangle$ and the data qubits are assumed to be $|\psi^k\rangle$ without loss of generality. Here, $|\psi^k\rangle$ denotes the collective two-qubit Dicke basis for the data qubits, with k being the number of $|1\rangle$ among data qubits. A three-qubit JPM gate U_s with time duration $\pi/(\sqrt{2}g)$ is turned on between two Hadamard gates on Q_0 . Then a second JPM gate is applied to undo the possible extra phase (refer to the last line of Table(c)). Even (odd) parity is mapped onto the measurement outcome 1 (0) of Q_0 . (b) When the JPM gate is turned on, depending on the syndrome qubit being $|0\rangle$ ($|1\rangle$), $|\psi^k\rangle$ undergoes two different oscillations, i.e., $|0\rangle|\psi^k\rangle \leftrightarrow |1\rangle|\psi^{k-1}\rangle$ and $|1\rangle|\psi^k\rangle \leftrightarrow |0\rangle|\psi^{k+1}\rangle$ with corresponding frequency gg_-^k and gg_+^k . (c) The table lists the oscillation frequency (third and fourth line) and the intermediate state (last line) after stage iii in (a) for different data qubits input state $|\psi^k\rangle$ (the second line). The blue (red) background denoting the even (odd) parity of the data qubits. (d) Circuit time comparison between the 2-qubit JPM scheme and CNOT scheme (the 4-qubit case has a similar result, except that all the circuit time \tilde{T} need to be doubled). Here, the circuit time calculated in the table mainly involves the entangling gate time for intuition with ignoring single-qubit gate time. The JPM scheme promotes $\sqrt{2}$ times speed up over the

CNOT scheme based on the homogeneous gate type, indicated as the black dotted arrow. The gate types circled by the red dashed line are utilized in the experiment.

Comment 4

Line 160: “map the even (odd) parity of the data qubits onto the state 0 (1)” , from the context, I think it should read “map the even (odd) parity of the data qubits onto the state 1 (0)”.

Response:

We thank the referee for his/her careful check. We have revised the typo as: “map the even (odd) parity of the data qubits onto the state $|1\rangle$ ($|0\rangle$)” in lines 176-177 in the revised manuscript.

Comment 5

Since the authors aim to apply the detection scheme into the quantum error correction codes, it is worth to have discussions on whether syndrome qubit errors will propagate to data qubits, which is crucial for the fault-tolerance of error syndrome detections.

Response:

We sincerely thank the referee for providing thoughtful feedback and insightful comments. We agree with the reviewer's comments that it is worthy to discuss on whether syndrome qubit errors will propagate to data qubits. Therefore, in the following, we would like to delve deeper into the analysis of the error propagation based on the two-qubit parity measurement case to demonstrate that although the syndrome qubit errors can propagate to data qubits, they doesn't lead to severe problem. Meanwhile, the JPM procedure may be made fault-tolerant for standard surface code with rough and smooth boundaries. The results for the four-qubit case can be derived with similar methods.

Recalling that the overall protocol for two-qubit parity measurement progress is described as (Eq.(5) in the main text)

$$U = U_s H_0 U_s H_0$$

with the JPM unitary gate $U_s = U_s(\pi/\sqrt{2}g)$ when the evolution time $t=\pi/(\sqrt{2}g)$. Ideally, we have

$$U = i\sigma_0^y(|00\rangle\langle 00| + |11\rangle\langle 11|) + I_0(|D^1\rangle\langle D^1| + |d^1\rangle\langle d^1|)$$

and

$$U_s = \sigma_0^z(|00\rangle\langle 00| - |11\rangle\langle 11|) + I_0(|d^1\rangle\langle d^1| - |D^1\rangle\langle D^1|)$$

where

$$|D^1\rangle = (|01\rangle + |10\rangle)/\sqrt{2}, \quad |d^1\rangle = (|01\rangle - |10\rangle)/\sqrt{2},$$

are the corresponding two-qubit Dicke bases introduced in the main text. However, the error can occur and propagates during the implementation of U_s . Following the reviewer's suggestion, we have initially examined the errors induced by interactions with the external environment (including the one causing leakage out of the computational space), commonly

known as decoherence. Besides, we also studied the intrinsic coherent errors, which is a consequence of imperfect control. When conducting error propagation analyses for syndrome measurement, it is crucial to ascertain those failures, which are highly probable in causing low weight errors, do not escalate into high weight errors affecting the data qubits. And this is critical to ensure the fault-tolerance.

(1) Decoherence errors caused by interactions with the environment

To analyze the impact of decoherence, which arises from interactions with the environment, on the quantum operation U_s involving three qubits, we adopt a simplified error model. Typically, one might consider weight-3 correlated Pauli noise, reflecting errors on all three qubits simultaneously. However, given that the underlying Hamiltonian primarily governs two-body interactions, it's more pragmatic to model the noise using weight-1 and weight-2 correlated Pauli errors (one on the ancilla qubits, one on the data qubit). These errors are akin to terms such as $X, Y, Z, XX, XY, XZ, \dots, ZZ$, occurring either before or after the U_s operation.

The operation U_s can be thought of as functionally analogous to a sequence of two CNOT gates, commonly used in standard syndrome measurement in quantum error correction. This analogy suggests that the pattern of error propagation in our system, influenced by U_s , should mirror that observed with consecutive CNOT operations.

For instance, consider the case when error ϵ_{C_1} (ϵ_{S_0}) acting on the data (syndrome) qubit, e.g, Q_1 (Q_0), before JPM gate U in our scheme. The error propagation of the bit-flip error, phase flip error, or both flip error through the JPM gate can be calculated as

① error on the first data qubit

$$U_s X_{C_1} U_s^\dagger = -Z_{C_1} Z_{S_0} X_{C_2}, \quad U_s Y_{C_1} U_s^\dagger = -Z_{C_1} Z_{S_0} Y_{C_2}, \quad U_s Z_{C_1} U_s^\dagger = I_{C_1} I_{S_0} Z_{C_2}$$

② error on syndrome qubit

$$U_s X_{S_0} U_s^\dagger = -Z_{C_1} X_{S_0} Z_{C_2}, \quad U_s Y_{S_0} U_s^\dagger = -Z_{C_1} Y_{S_0} Z_{C_2}, \quad U_s Z_{S_0} U_s^\dagger = I_{C_1} Z_{S_0} I_{C_2}$$

Then for the 2-qubit JPM measure-Z circuit, that is, $H_0 U_s H_0 U_s = U_z$, the error propagation can be recognized as

$$U_z X_{C_1} U_z^\dagger = Y_{C_1} Y_{S_0} Z_{C_2}, \quad U_z Y_{C_1} U_z^\dagger = -X_{C_1} Y_{S_0} Z_{C_2}, \quad U_z Z_{C_1} U_z^\dagger = Z_{C_1} I_{S_0} I_{C_2}$$

for the data qubit error, and

$$U_z X_{S_0} U_z^\dagger = -Z_{C_1} X_{S_0} Z_{C_2}, \quad U_z Y_{S_0} U_z^\dagger = I_{C_1} Y_{S_0} I_{C_2}, \quad U_z Z_{S_0} U_z^\dagger = -Z_{C_1} Z_{S_0} Z_{C_2}$$

for the syndrome qubit error, respectively. The error propagation for the 2-qubit measure-X circuit can be obtained in a similar way.

(2) Coherence errors caused by imperfect control

Taking into account failures of infinite small strength, denoted as ϵ , for both scenarios, the most critical situation to consider is when a weight 2 error occurs on both data qubits, occurs with a strength at the order of $o(\epsilon)$ — such faulty path can deteriorate the fault-tolerance and should be avoided by choosing a suitable error correction code. Therefore, our analysis should primarily focus on the error propagation resulting from failures with a strength approximating $o(\epsilon)$ in the process of parity check U . We further focus on the failure of U_s process, since the failures and their propagation on Hadamard gates are trivial and obviously benign.

(2-1) Imperfect JPM unitary $U_s' = U_s + \delta U_s$

We first consider the coherent errors caused by imperfect control and neglect the

decoherence (since the strength of both of them to occur is $o(\epsilon^2)$). Ideally the validity of the JPM parity measurement requires coupling strength between ancilla and both data qubits to be equal, i.e., $g_1 = g_2 = g$. In practical operation, situations that are not strictly equal may occasionally occur, e.g., $g_1 = g, g_2 = g + \delta g, \delta g \ll g$. The deviations in coupling strengths may lead to imperfect control of U_s . In this case, the Hamiltonian can be denoted as

$$H' = \sigma_0^{01}(g_1\sigma_1^{10} + g_2\sigma_2^{10}) + H.c..$$

Define $\Omega' = \sqrt{g_1^2 + g_2^2} = \Omega + \delta\Omega, \Omega = \sqrt{2}g$, we can effectively decompose the three qubits Hilbert space $\mathbb{C}^{\otimes 8}$ into $H_1 \oplus H_2$ where

$$H_1 = \text{span}\{|100\rangle, |0D_1^1\rangle, |1D_2^1\rangle, |011\rangle\}, \quad H_2 = \text{span}\{|000\rangle, |0d_1^1\rangle, |1d_2^1\rangle, |111\rangle\}$$

Here,

$$|D_1^1\rangle = \frac{g_1|10\rangle + g_2|01\rangle}{\Omega'} = \cos\left(\frac{\pi}{4} + \epsilon\right)|10\rangle + \sin\left(\frac{\pi}{4} + \epsilon\right)|01\rangle = |D^1\rangle + \epsilon|d^1\rangle + o(\epsilon^2),$$

$$|d_1^1\rangle = \frac{g_1|01\rangle - g_2|10\rangle}{\Omega'} = \cos\left(\frac{\pi}{4} + \epsilon\right)|01\rangle - \sin\left(\frac{\pi}{4} + \epsilon\right)|10\rangle = |d^1\rangle - \epsilon|D^1\rangle + o(\epsilon^2),$$

$$|D_2^1\rangle = \frac{g_2|10\rangle + g_1|01\rangle}{\Omega'} = \sin\left(\frac{\pi}{4} + \epsilon\right)|10\rangle + \cos\left(\frac{\pi}{4} + \epsilon\right)|01\rangle = |D^1\rangle - \epsilon|d^1\rangle + o(\epsilon^2),$$

$$|d_2^1\rangle = \frac{g_2|01\rangle - g_1|10\rangle}{\Omega'} = \sin\left(\frac{\pi}{4} + \epsilon\right)|01\rangle - \cos\left(\frac{\pi}{4} + \epsilon\right)|10\rangle = |d^1\rangle + \epsilon|D^1\rangle + o(\epsilon^2),$$

with $\epsilon \approx \frac{\delta g}{2g} = \frac{\delta\Omega}{\Omega}$ and $\delta g = \sqrt{2}\delta\Omega$. Then the dynamics of Hamiltonian H' follows can be calculated as

$$\begin{aligned} U'_s &= |000\rangle\langle 000| + |111\rangle\langle 111| + |0d_1^1\rangle\langle 0d_1^1| + |1d_2^1\rangle\langle 1d_2^1| \\ &\quad + \cos(\Omega't)I_1 - i\sin(\Omega't)\sigma_1^x + \cos(\Omega't)I_2 - i\sin(\Omega't)\sigma_2^x \\ U'_s &= |000\rangle\langle 000| + |111\rangle\langle 111| + |0d_1^1\rangle\langle 0d_1^1| + |1d_2^1\rangle\langle 1d_2^1| \\ &\quad - \left(1 - \frac{(\delta\Omega t)^2}{2}\right)(|100\rangle\langle 100| + |0D_1^1\rangle\langle 0D_1^1|) + i(\delta\Omega t)(|100\rangle\langle 0D_1^1| + |0D_1^1\rangle\langle 100|) \\ &\quad - \left(1 - \frac{(\delta\Omega t)^2}{2}\right)(|011\rangle\langle 011| + |1D_2^1\rangle\langle 1D_2^1|) + i(\delta\Omega t)(|011\rangle\langle 1D_2^1| + |1D_2^1\rangle\langle 011|) \end{aligned}$$

with the time condition $\cos(\Omega t) = \pi$, and I_1, σ_1^x (I_2, σ_2^x) defined on the two-level subspace spanned by $|100\rangle, |0D_1^1\rangle$ ($|011\rangle, |1D_2^1\rangle$) respectively. Note that the errors due to limited precision of time control is of the same strength order forms of $\delta\Omega t$, and hence would not be discussed here for simplicity. Expanding the above U'_s expression further and keeping only the first-order expansion, we have

$$U'_s = U_s + \delta U_s$$

and

$$\delta U_s = i\delta\Omega t \left[\begin{aligned} &\sigma_0^x \otimes \sigma_1^x (|01\rangle_{01}\langle 01| + |10\rangle_{01}\langle 10|) \otimes I_2 \\ &+ \sigma_0^x \otimes \sigma_2^x (|01\rangle_{02}\langle 01| + |10\rangle_{02}\langle 10|) \otimes I_1 \end{aligned} \right] - 2\epsilon\sigma_0^z$$

(2-2) Effect of imperfect gate on the parity measurement circuit U'

Next we will step further to consider the effect of imperfect gate on the parity measurement circuit. Taking the ZZ parity check as an example, we have

$$U' = U'_s H_0 U'_s H_0 = U_s H_0 U_s H_0 + U_s H_0 \delta U_s H_0 + \delta U_s H_0 U_s H_0$$

① The error at the end of parity check implementation U' caused by $\delta\Omega$ terms in δU_s . We first consider the error effect on U' caused by the first $\delta\Omega$ -correlated term, i.e.,

$$(|01\rangle_{01}\langle 01| + |10\rangle_{01}\langle 10|) \otimes I_2 = \frac{1}{2}(I_0 \otimes I_1 - \sigma_0^z \otimes \sigma_1^z) \otimes I_2.$$

Specifically, the error induced by the two different paths can be demonstrated as $U_s H_0 \delta U_s H_0 / (i\delta\Omega t)$

$$\begin{aligned} &= \left[\sigma_0^z (|00\rangle_{12}\langle 00| - |11\rangle_{12}\langle 11|) + I_0 (|d^1\rangle_{12}\langle d^1| - |D^1\rangle_{12}\langle D^1|) \right] H_0 \sigma_0^x \otimes \sigma_1^x (|01\rangle_{01}\langle 01| + |10\rangle_{01}\langle 10|) \otimes I_2 H_0 \\ &= \left[\sigma_0^z \otimes \sigma_1^z (|00\rangle_{12}\langle 00| + |11\rangle_{12}\langle 11|) - (|D^1\rangle_{12}\langle D^1| + |d^1\rangle_{12}\langle d^1|) \sigma_1^x \otimes \sigma_2^x \right] H_0 \sigma_0^x \otimes \sigma_1^x \frac{1}{2} (I_0 \otimes I_1 - \sigma_0^z \otimes \sigma_1^z) \otimes I_2 H_0 \\ &= \frac{i}{2} \sigma_1^y (|D^1\rangle_{12}\langle D^1| + |d^1\rangle_{12}\langle d^1|) - \frac{1}{2} \sigma_0^x \otimes \sigma_1^x (|D^1\rangle_{12}\langle D^1| + |d^1\rangle_{12}\langle d^1|) \\ &\quad - \frac{1}{2} \sigma_0^z \otimes \sigma_2^x (|00\rangle_{12}\langle 00| + |11\rangle_{12}\langle 11|) + \frac{i}{2} \sigma_0^y \otimes \sigma_1^z \otimes \sigma_2^x (|00\rangle_{12}\langle 00| + |11\rangle_{12}\langle 11|), \\ &\delta U_s H_0 U_s H_0 / (i\delta\Omega t) \end{aligned}$$

$$\begin{aligned} &= \sigma_0^x \sigma_1^x (|01\rangle_{01}\langle 01| + |10\rangle_{01}\langle 10|) \otimes I_2 \cdot H_0 \left[\sigma_0^z (|00\rangle_{12}\langle 00| - |11\rangle_{12}\langle 11|) + I_0 (|d^1\rangle_{12}\langle d^1| + |D^1\rangle_{12}\langle D^1|) \right] H_0 \\ &= \frac{1}{2} \sigma_0^x \sigma_1^x (I_0 \otimes I_1 - \sigma_0^z \otimes \sigma_1^z) \otimes I_2 \cdot H_0 \left[\sigma_0^z \otimes \sigma_1^z (|00\rangle_{12}\langle 00| + |11\rangle_{12}\langle 11|) - I_0 \otimes \sigma_1^x \otimes \sigma_2^x (|D^1\rangle_{12}\langle D^1| + |d^1\rangle_{12}\langle d^1|) \right] H_0 \\ &= -\frac{i}{2} \sigma_1^y (|00\rangle_{12}\langle 00| + |11\rangle_{12}\langle 11|) + \frac{1}{2} \sigma_0^z \otimes \sigma_1^x (|00\rangle_{12}\langle 00| + |11\rangle_{12}\langle 11|) \\ &\quad - \frac{1}{2} \sigma_0^x \otimes \sigma_2^x (|D^1\rangle_{12}\langle D^1| + |d^1\rangle_{12}\langle d^1|) + \frac{i}{2} \sigma_0^y \otimes \sigma_1^z \otimes \sigma_2^x (|D^1\rangle_{12}\langle D^1| + |d^1\rangle_{12}\langle d^1|), \end{aligned}$$

Note that all error terms occurring with a strength of $\delta\Omega$ can project to odd or even parity space followed by the weight 1 or 2 Pauli errors, and all these Pauli errors are not problematic for $U(2)$ parity check. The error induced by the second term $(|01\rangle_{02}\langle 01| + |10\rangle_{02}\langle 10|) \otimes I_1$ can also be calculated in a similar way and shown to be benign.

② The error at the end of parity check implementation U' caused by ϵ terms in δU_s

The term $2\epsilon\sigma_0^z \otimes \sigma_1^z (|D^1\rangle\langle D^1| + |d^1\rangle\langle d^1|)$ causes the parity check implementation U' to accumulate errors as

$$\begin{aligned} &U_s H_0 \delta U_s H_0 + \delta U_s H_0 U_s H_0 \\ &= 2\epsilon\sigma_0^x \otimes \sigma_1^z (|d^1\rangle\langle d^1| - |D^1\rangle\langle D^1|) - 2\epsilon\sigma_0^z \otimes \sigma_1^z (|d^1\rangle\langle d^1| - |D^1\rangle\langle D^1|) \end{aligned}$$

after the two faulty paths. It can be observed that in this situation, the term

$$\sigma_1^z (|d^1\rangle\langle d^1| - |D^1\rangle\langle D^1|) = \frac{1}{2} I_0 \otimes (-i\sigma_1^y \otimes \sigma_2^x + \sigma_1^x \otimes i\sigma_2^y)$$

actually introduces the weight 2 error on both data qubits. Fortunately, we can prevent the propagation of such Pauli errors by selecting proper encoding method, such as the standard surface code architecture. While error propagation can compromise the fault-tolerance of rotated surface codes, standard surface codes with rough and smooth boundaries maintain full tolerance as errors don't propagate along the logical X operator chain. The situation is similar for JPM procedure for X -style stabilizer. Therefore, JPM is more compatible with standard surface codes.

Comment 6

In lines 188-189, the authors claim that they can't find a time t to align all the even (odd) parity oscillation for the 4-qubit scheme. I am interested to see if you can find the time t for n -qubit scheme? Or does this only work for the 2-qubit scheme?

Response:

We appreciate the referee for his/her careful reading and rigorous consideration. We would like to point out that the current scheme, which sandwiches a three-qubit interaction U_s between two Hadamard gates H_0 (followed by the same U_s for phase elimination), is only applicable to the 2-qubit case. In the following, we will begin with the N -qubit scenario to introduce a unified computational approach, followed by a detailed explanation for the specific cases of the joint 2-qubit and 4-qubit schemes, i.e., $N = 2$ and $N = 4$.

(1) N -qubit case

We assume that N data qubits have identical frequency ω_q and coupling strength g with the syndrome qubit, then the collective operators for the qubits can be expressed as

$$J_z = \sum_{i=1}^N \sigma_i^z, J_{\pm} = \sum_{i=1}^N \sigma_i^{\pm},$$

where σ_i is the individual Pauli operator for qubit i . The collective operators obey the standard $SU(2)$ algebra commutation relations, i.e.,

$$[J_z, J_{\pm}] = \pm 2J_{\pm}, [J_+, J_-] = J_z,$$

For a system of N two-level atoms, the Hilbert space can be decomposed into a direct sum of subspaces, i.e., irreducible representation (irrep) of $SU(2)$ algebra, which may be indexed by the total spin of the irrep J . The dimension of the irrep is give by $d_J = 2J + 1$ with the allowed values of

$$J \in \left\{ \frac{N}{2}, \frac{N}{2} - 1, \frac{N}{2} - 2, \dots, J_{min} \right\}, \quad J_{min} \begin{cases} 0 & N = \text{even} \\ \frac{1}{2} & N = \text{odd} \end{cases}.$$

The multiplicity of a given irrep is

$$m_J = \frac{2J + 1}{N/2 + J + 1} \binom{N}{N/2 + J}$$

For a specific irrep with the total spin J , the eigenstates of this d_J -dimension subspace can be denoted as $|J, m_z\rangle$, where $m_z = -J, \dots, J$. Sometimes it is convenient to work in the excitation number basis $|s\rangle \equiv |J, s - J\rangle$ of the system, where the excitation number is defined as $s \equiv m_z + J$. Then an explicit representation for the collective spin operators is then given by

$$J_z = \sum_{s=0}^{2J} (s - J) |s\rangle \langle s|,$$
$$J_+ = \sum_{s=0}^{2J} \sqrt{(2J - s)(s + 1)} |s + 1\rangle \langle s|,$$

$$J_- = \sum_{s=0}^{2J} \sqrt{(2J-s+1)s} |s-1\rangle \langle s|.$$

(2) 4-qubit case

For $N = 4$, the subspaces decomposition can be written as $2 \otimes 1 \otimes 1 \otimes 1 \otimes 0 \otimes 0$. Here the number is maximum value of the angular momentum J for each subspace, and the repetition count indicates the multiplicity m_j of the corresponding irreducible representation. Specifically, the basis of all these representations can be written as follows:

① $J = 2$

One five-dimensional representation, with the basis denoted by the 4-qubit Dicke states $|D^k\rangle$, $k = 0, 1, \dots, 4$, i.e., an symmetric combination of all permutations of 4-qubit states with k excitations. The vacuum vector $|D^0\rangle = |0000\rangle$ and $J_-|D^0\rangle = 0$. The other states can be obtained by the collective operator J_+ , with the matrix element $g_+^k = \langle D^{k+1}|J_+|D^k\rangle = \sqrt{(4-k)(k+1)}$. For example,

$$|D^1\rangle = \frac{1}{2}(|0001\rangle + |0010\rangle + |0100\rangle + |1000\rangle).$$

② $J = 1$

Three three-dimensional representations, with the vacuum vectors:

$$|T^0\rangle = \frac{1}{2}(|1000\rangle + i|0100\rangle - |0010\rangle - i|0001\rangle),$$

$$|H^0\rangle = \frac{1}{2}(|1000\rangle - |0100\rangle + |0010\rangle - |0001\rangle),$$

$$|R^0\rangle = \frac{1}{2}(|1000\rangle - i|0100\rangle - |0010\rangle + i|0001\rangle).$$

The matrix element $g_+^k = \langle T^{k+1}|J_+|T^k\rangle = \sqrt{(2-k)(k+1)}$, $g_-^k = \langle T^{k+1}|J_-|T^k\rangle = \sqrt{(2-k+1)k}$, $k = 0, 1, 2$ and $J_+|T^2\rangle = 0$. The same holds for $|H^k\rangle, |R^k\rangle$.

③ $J = 0$

Two one-dimensional representations, with the vacuum vectors:

$$|W^0\rangle = \frac{1}{\sqrt{12}}(|1100\rangle + |0011\rangle + |1001\rangle + |0110\rangle - 2|0101\rangle - 2|1010\rangle),$$

$$|O^0\rangle = \frac{1}{2}(|1100\rangle + |0011\rangle - |1001\rangle - |0110\rangle),$$

The matrix element $g_+^k = 0$, $k = 0$ and $J_\pm|W^0\rangle = 0$. The same holds for $|O^0\rangle$.

Therefore, the coupling matrix element $g_+^k (g_-^k)$ of the transitions between $|\psi^k\rangle$ and $|\psi^{k+1}\rangle (|\psi^{k-1}\rangle)$ for 4-qubit scheme can be written as

	$ D^0\rangle$	$ D^1\rangle$	$ D^2\rangle$	$ D^3\rangle$	$ D^4\rangle$	$ T^0\rangle$	$ T^1\rangle$	$ T^2\rangle$	$ W^0\rangle$
g_-^k	0	2	$\sqrt{6}$	$\sqrt{6}$	2	0	$\sqrt{2}$	$\sqrt{2}$	0
g_+^k	2	$\sqrt{6}$	$\sqrt{6}$	2	0	$\sqrt{2}$	$\sqrt{2}$	0	0

Here, $|H^k\rangle, |R^k\rangle (|O^k\rangle)$ display the same behavior with $|T^k\rangle (|W^k\rangle)$, respectively. To implement even (odd) joint parity measurement in 4-qubit scheme, the time condition $\sin(gg_\pm t) = 0$ should be satisfied simultaneously for all subspaces. However, it's apparent

that we cannot find a time t at which $\sin(\sqrt{2}gt)$, $\sin(2gt)$ and $\sin(\sqrt{6}gt)$ reach 0 at the same time.

(3) 2-qubit case

In the case of $N = 2$, referring to the 2-qubit scheme shown in Fig.1(c) in the main text, the coupling matrix elements are represented by 0 and $\sqrt{2}$. As a result, the time condition can be expressed as $\sin(\sqrt{2}gt) = 0$, which can be satisfied by setting $\sqrt{2}gt = \pi$.

Building upon the aforementioned analysis, we conclude in the main text that the scheme is effective for the 2-qubit case, and we can't find a time t to align all the even (odd) parity oscillation for the 4-qubit scheme. While for the general N -qubit scheme, where the subspaces decomposition is expressed as a direct sum of J , with $J \in \{\frac{N}{2}, \frac{N}{2} - 1, \frac{N}{2} - 2, \dots, J_{min}\}$ and the multiplicity m_J , the coupling matrix element for a specific irreducible representation J can be written as $g_+^{s_j} = \sqrt{(2J - s_j)(s_j + 1)}$, $g_-^{s_j} = \sqrt{(2J - s_j + 1)s_j}$, $s_j = 0, \dots, 2J$. The parity measurement time t should be determined by finding $\sin(gg_{\pm}^{s_j}t) = 0$ for all J .

Comment 7

In line 201 and line 403, the author mentioned the $\sqrt{2}$ speed up over the traditional CNOT measurement scheme. I am confused with such a claim since no detailed condition is provided. Usually a CZ based CNOT just requires one CZ and two Hadamard gates, where the CZ operation time is on par with the JPM unitary mentioned here, I cannot see the speed up other than the constant time reduction in single-qubit operations.

Response:

We thank the referee for his/her thoughtful reading and valuable questions. We are not aware of and feel sorry about our inaccurate descriptions in the original manuscript and omissions in data presentation, which led to the misunderstandings and confusion from the referee about the time acceleration of our JPM scheme. In practice, the JPM scheme, taking advantage of the controllable collective behavior, promotes theoretically $\sqrt{2}$ times sped up over the CNOT scheme based on the homogeneous quantum gate, i.e., collective iSWAP (collective CZ) operation in the JPM scheme is usually $\sqrt{2}$ times faster than iSWAP (CZ) gate in the CNOT scheme. In the following, we elaborate the time consumption in both schemes and explain the applications in our experiment.

(1) Time consumption on unitary gate

In the THEORETICAL MODEL section, we derive that the system Hamiltonian of the JPM scheme in the interaction picture with the two-level approximation can be expressed as:

$$H_s = g\sigma_0^{n, n+1}(\sigma_1^{10} + \sigma_2^{10}) + H. c.,$$

where $\sigma_i^{n, m} = |n\rangle_i \langle m|$ is the $|m\rangle \rightarrow |n\rangle$ transition matrix element for qubit i . For instance, $n = 0$ for collective iSWAP gate and $n = 1$ for collective CZ gate. Therefore, the JPM unitary gate can then be acquired as $U_{siSWAP}(U_{sCZ})$ at $\sqrt{2}gt = \pi$ ($\sqrt{2}Gt = \pi, G = \sqrt{2}g$) under collective iSWAP (collective CZ) operation. Compared to the CNOT scheme, where iSWAP (CZ) gate is generated at specific gate time $gt = \pi/2$ ($Gt = \pi, G = \sqrt{2}g$), the JPM scheme is

typically $\sqrt{2}$ times sped up due to the collective behavior (i.e. $g \leftrightarrow \sqrt{2}g$, $G \leftrightarrow \sqrt{2}G$). We should notice that although the single iSWAP gate in the CNOT scheme only consumes half of the whole swap time, but when consider the overall parity measurement circuit time, we still find the acceleration of JPM scheme. For a more detailed comparison, please refer to Table. R1-1, which records the time consumption for the unitary gate based on the two schemes.

We also recommend the referee to check out our response in *Comment 5 for Reviewer #3*, where we verify that the collective iSWAP gate in the JPM scheme can be implemented faster after fully exploiting the leakage interference to approach the speed limit compared with the CZ gate, thereby further reducing the circuit time in experiment.

(2) Time consumption on parity measurement circuit

When it comes to the time acceleration of the JPM scheme in the parity measurement circuit compared to the CNOT scheme, we would like to discuss on several situations.

① Minus phase correction with another $U_s = U_{siSWAP}(U_{sCZ})$

As shown in Eq. (5) and Fig. 1(a) in the revised manuscript, we introduce another U_s (stage iv) to undone the minus phase in the Dicke states, e.g., $|D_1\rangle$ and $|D_2\rangle$ [last line in Fig1.(c)]. Then the circuit time \tilde{T} of parity measurement can be summarized in Table. R1-1. We find that with the homogeneous gate operation (i.e., collective iSWAP \leftrightarrow iSWAP or collective CZ \leftrightarrow CZ), the circuit time \tilde{T} in the JPM scheme are usually $\sqrt{2}$ times faster than the corresponding CNOT scheme, i.e., $\tilde{T}_{iSWAP-CNOT} = \sqrt{2}\tilde{T}_{siSWAP-JPM}$, $\tilde{T}_{CZ-CNOT} = \sqrt{2}\tilde{T}_{sCZ-JPM}$, indicated as the black dotted arrow. In our experiment, based on the surface code architecture and the consideration of not introducing additional energy levels (e.g., state $|2\rangle$), we choose the collective iSWAP-based JPM scheme to demonstrate the parity measurement, which is $\sqrt{2}$ times faster than the iSWAP-based CNOT scheme but seems to have the same \tilde{T} with CZ-based CNOT scheme, shown as the red dotted circle in the Table. R1-1. However, when the collective CZ-based JPM scheme is chosen, we will still get $\sqrt{2}$ times sped up over the CZ-based CNOT scheme.

Scheme	CNOT		JPM	
	iSWAP	CZ	Collective iSWAP	Collective CZ
Gate Type	iSWAP	CZ	Collective iSWAP	Collective CZ
Gate Time	$\frac{\pi}{2g}$	$\frac{\pi}{\sqrt{2}g}$	$\frac{\pi}{\sqrt{2}g}$	$\frac{\pi}{2g}$
Circuit Time \tilde{T} (2-qubit case)	$\frac{2\pi}{g}$	$\frac{2\pi}{\sqrt{2}g}$	$\frac{2\pi}{\sqrt{2}g}$	$\frac{\pi}{g}$
Circuit Time \tilde{T} (4-qubit case)	$\frac{4\pi}{g}$	$\frac{4\pi}{\sqrt{2}g}$	$\frac{4\pi}{\sqrt{2}g}$	$\frac{2\pi}{g}$

TABLE. R1-1. Circuit time comparison between the 2-qubit (4-qubit) JPM scheme and CNOT scheme. Here, the circuit time calculated in the table mainly involves the entangling gate time for intuition with ignorance of single-qubit gate time. The JPM scheme promotes $\sqrt{2}$ times sped up over the CNOT scheme based on the homogeneous gate type, indicated as the black dotted arrow. The gate type circled by the red dashed line is utilized in the experiment.

② Minus phase elimination with post-selection (for reference, not included in the paper)

FIG. R1-2. Joint parity measurement protocol with post-selection, where the syndrome qubit (Q_0) is initialized in state $|0\rangle$ and the data qubits (Q_1, Q_2) are assumed to be $|\psi^k\rangle$. A three-qubit JPM gate U_s with time duration $\pi/(\sqrt{2}g)$ is turned on between two Hadamard gates on Q_0 . Then, parity detection is executed on the syndrome qubit, followed by a digital σ_z operation on either of the two data qubits if the syndrome qubit is detected as flipped, or σ_x operations on both data qubits if the syndrome qubit remains unchanged.

In fact, we did not mention another potential time advantage for 2-qubit JPM scheme in the article. Instead of adding the second U_s , the minus phase can also be eliminated with post-selection, i.e., operating on the data qubits based on the outcome of the parity measurement on the syndrome qubit, as shown in Fig. R1-2. The overall protocol can then be expressed as:

$$H_0 U_s H_0 = \sigma_0^x (|00\rangle\langle 00| - |11\rangle\langle 11|) - I_0 (|D^1\rangle\langle D^1| - |d^1\rangle\langle d^1|).$$

It can be found that, when the syndrome qubit is detected as flipped, indicating that the input state of data qubits is in an even number, a digital σ_z operation on either of the two data qubits can be performed to cancel out the minus phase. Similarly, when the syndrome qubit remains unchanged, indicating that the input state of data qubits is in an odd number, σ_x operations can be executed on both data qubits. Under this circumstance, the JPM scheme gains further time advantage over the traditional CNOT scheme. Specifically, the circuit time \tilde{T} of the JPM scheme gets $2\sqrt{2}$ times faster than the corresponding CNOT scheme, as shown in Table. R1-2. Furthermore, even for the collective iSWAP-based JPM scheme, the acceleration remains twice that of the CZ-based CNOT scheme, which may allows for higher fidelity.

In practice, this method is promising for boundary parity detection in the surface code architecture. However, this protocol cannot be used in the 4-qubit JPM scheme owing to the lack of proper post-selection operators to eliminate the minus phase. Besides, this protocol needs more theoretical analyses when applied in the surface code. Therefore we did not include this part in the article to ensure the consistence of the content, but the corresponding idea can be utilized as reference for the referee.

Scheme	CNOT		JPM	
	iSWAP	CZ	Collective iSWAP	Collective CZ
Gate Type	iSWAP	CZ	Collective iSWAP	Collective CZ
Gate Time	$\frac{\pi}{2g}$	$\frac{\pi}{\sqrt{2}g}$	$\frac{\pi}{\sqrt{2}g}$	$\frac{\pi}{2g}$
Circuit Time \tilde{T} (post selection)	$\frac{2\pi}{g}$	$\frac{2\pi}{\sqrt{2}g}$	$\frac{\pi}{\sqrt{2}g}$	$\frac{\pi}{2g}$

$\xrightarrow{2\sqrt{2}}$ $\xrightarrow{2\sqrt{2}}$

TABLE. R1-2. Circuit time comparison between the 2-qubit JPM scheme and CNOT scheme

based on the post-selection protocol (not included in the article). The time advantage of the JPM scheme will be further enhanced to $2\sqrt{2}$ times over the CNOT scheme, denoted as the black dotted arrow.

Finally, in order to eliminate the potential confusion about the time acceleration and improve the presentation in the article, we have made the corresponding revisions to our main text as:

- In Fig. 1, we add Fig. 1(d) to clarify the time consumption for the JPM scheme and CNOT scheme. The corresponding caption for Fig. 1(d) is described as "(d) Circuit time comparison between the 2-qubit JPM scheme and CNOT scheme (the 4-qubit case has a similar result, except that all the circuit time \tilde{T} need to be doubled). Here, the circuit time calculated in the table mainly involves the entangling gate time for intuition while ignoring single-qubit gate time. The JPM scheme promotes $\sqrt{2}$ times sped up over the CNOT scheme based on the homogeneous gate type, indicated as the black dotted arrow. The gate types circled by the red dashed line are utilized in the experiment."
- In lines 86-90 in INTRODUCTION, we revise the sentence to "Meanwhile, this controllable collective behavior, which can be applied to both the iSWAP and CZ operations architecture, promotes $\sqrt{2}$ times sped up over the corresponding CNOT scheme based on the homogeneous gate operation type."
- In lines 219-229 in THEORETICAL MODEL, we revise the sentence to "..., the effective entangling gates in the JPM scheme are usually $\sqrt{2}$ times faster than that of the corresponding CNOT scheme with the homogeneous gate operation type, indicated as the black dotted arrow in Fig. 1(d), thus releasing the burden of iSWAP-based CNOT scheme with complex circuit composition. Besides, the iSWAP-based JPM scheme utilized in the following experimental analysis [marked as the red dotted circle in Fig. 1(d)] does not introduce any additional energy levels, which indicates its consistence with the surface code architecture."
- In lines 432-434 in DISCUSSIONS, we revise the sentence to "..., and thus is $\sqrt{2}$ times sped up over the commonly-used CNOT scheme for the homogeneous gate operation type."

Comment 8

Can authors elaborate more specifically for the near 2 percent fidelity improvement in the iSWAP based parity measurement than the CZ based CNOT scheme? Is it because the CZ fidelity is not high enough compared with the JPM unitary?

Response:

We thank the referee for his/her rigorous consideration and valuable questions. In practice, our intention is to propose a new parity measurement scheme with feasible experimental implementation, and thus the comparison with CNOT scheme is to help analyze the potential error sources in JPM unitary gate. As mention by the Reviewer #3 that "... further optimization could bring the performance of the CNOT-based scheme closer to that of the JPM scheme", we believe that both schemes can perform better with optimized gate operation and careful circuit optimization. Therefore, we feel sorry about our confusing expression in the original manuscript such as "The average parity detection fidelity of the JPM scheme [...] outperforms the CNOT scheme [...] in this experiment", and further focus on

the innovation of the scheme itself as suggested by the *Reviewer #2* "..., while the comparison is interesting, I don't see the emphasis on better performance as necessary for the message of the paper. Indeed, this new scheme's working principle is interesting no matter what, as long as fidelities is comparable, which seems to be the case here." Furthermore, regarding the phenomenon we observed in the experiment, where the parity measurement fidelity of the JPM scheme was slightly higher than that of the CNOT scheme, we carefully considered several reasons combined with the experimental analyses and simulations to clarify the referee's concerns.

Overall, we think the mechanism may be related to the fact that the CZ gate is more susceptible to the influence of global circuit quantum crosstalk (ZZ crosstalk) and leakage (leakage to coupler or leakage to qubit higher excited state). Besides, the impact of two-level systems (TLSs) on multi-qubit gate operations is an important consideration, as it can lead to errors in the gate operation and degradation of the overall scheme performance. As the operating frequencies of the qubits may differ during the JPM and CZ operations, the impact of TLSs on the two gate types may also differ. We believe that by optimizing the overall performance of the gate operation, such as reducing the frequency difference between idle positions of qubits within their anharmonicities, and improving the quality of the superconducting materials to reduce TLSs, as well as optimizing control waveforms, we can further improve the parity measurement fidelity in experiment.

(1) Theoretical analyses

We first conduct some theoretical calculations and analyses based on the experimental data. In order to ensure consistency in the calibration of gate fidelity for both the JPM unitary gate and CZ gate, we use the Quantum Process Tomography (QPT) with eliminating the SPAM error. The specific experimental parameters are as follows:

- JPM gate

We choose Q_0 , Q_2 and Q_4 to implement JPM unitary gate with Q_0 functioned as the syndrome qubit. In the experiment, Q_2 and Q_4 are tuned to the frequency of Q_0 , and both couplers C_{02} and C_{04} are adjusted to around 4.395 GHz. At this point, the gate operation time for the JPM gate is about 79 ns, and the average gate fidelity after eliminating the SPAM error is found to be 98.5% through QPT.

- CZ gate

We re-calibrate the CZ gates for $Q_0 - Q_2$ and $Q_0 - Q_4$ pairs separately. Keeping the frequency of Q_0 constant, we adjust the frequencies of the other qubit to make the $|11\rangle$ and $|02\rangle$ (or $|20\rangle$) states into resonate, thereby accumulating phase and implementing the CZ gate operation with C_{02} and C_{04} adjusted to around 4.395 GHz respectively. At this point, the gate operation time for both CZ gate pairs is also about 72 ns, and the average gate fidelity after eliminating the SPAM error is found to be 98.4% and 98.5% through QPT.

Ideally, the main source of circuit error comes from two-qubit gate errors when we ignore the qubits' decoherence error and coherent error, thereby the upper bound of parity measurement fidelity can be simply calculated as the product of the two-qubit gate fidelity, i.e. the JPM gate $98.5\% \times 98.5\% \approx 97.0\%$; the CZ gate $98.4\% \times 98.5\% \approx 96.9\%$. It can be observed that the upper limits are theoretically quite close, indicating that the difference in the upper limits between the two schemes would not be particularly large when the gate fidelity is comparable from an experimental perspective, and the fidelity errors reflected in the experiment would in principle be small.

(2) Simulation results

However, in our experiment, we find that under the current experimental environment and parameters, the parity fidelity of JPM scheme is slightly higher than that of CNOT scheme, as mentioned in our main text. We think that the reason may be related to the susceptibility of CZ gate to circuit crosstalk and leakage.

(2-1) Fidelity tolerance of ZZ crosstalk

Similar to the method utilized in *Phys. Rev. Lett.* 127, 060505(2021), we simulate the impact of spectator qubit on gate operations for both JPM gate and CZ gate in presence of ZZ crosstalk by QuTip, and further obtain the tolerance of gate operations to ZZ error. For instance, as shown in Fig. R1-3(a), the system Hamiltonian for implementing JPM unitary gate with one spectator qubit can be expressed as:

$$H = \sum_{i=0}^2 \frac{\alpha_i}{2} a_i^\dagger a_i^\dagger a_i a_i + \frac{\alpha_s}{2} a_s^\dagger a_s^\dagger a_s a_s + g(a_0 a_1^\dagger + a_1 a_0^\dagger) + g(a_0 a_2^\dagger + a_2 a_0^\dagger) + \xi_{ZZ} a_2^\dagger a_2 a_s^\dagger a_s,$$

where α_i ($i = 0 \sim 2$) is the anharmonicities of qubits Q_0, Q_1 and Q_2 , α_s is the anharmonicity of spectator qubit S_1 ; g is the effective coupling strength during JPM unitary gate and ξ_{ZZ} represents the ZZ interaction between spectator qubit S_1 and one data qubit Q_2 . We simulate the gate fidelity by QPT with varying the interaction strength of ξ_{ZZ} , and calculate the fidelity tolerance to ZZ crosstalk by subtracting the fidelity when $\xi_{ZZ} = 0$, i.e.

$$FT = F(\xi_{ZZ} = \xi) - F(\xi_{ZZ} = 0)$$

where FT represents fidelity tolerance. The simulation results can be found in Fig. R1-3(b) where we investigate two situations with spectator qubit in $|1\rangle$ state and $|0\rangle + |1\rangle$ state. Apparently, JPM gate with collective iSWAP gate is more tolerant to ZZ crosstalk compared with CZ gate. In practice, this may be reasonable since ZZ crosstalk can directly impact the phase accumulation which seems to be more sensitive to CZ gate procedure. In our experiment, the coupler offers the effective coupling strength between neighbouring qubits and hence may induce unwanted ZZ crosstalk if coupler is not efficiently closed. We believe through optimizing the parity circuit by fine tuning the coupler frequency may further improve the parity measurement procedure especially for CNOT scheme. Moreover, we also explore the quantum crosstalk in the surface code architecture for the JPM scheme and CNOT scheme in our response to *Comment 4 of Reviewer #3*.

FIG. R1-3. (a) The schematic diagram for simulating gate fidelity tolerance to ZZ crosstalk in presence of spectator qubit. The dotted box represents the gate procedure. In experiment, spectator qubits around the gate qubits may induce unwanted ZZ interaction which reducing the gate fidelity and further parity measurement fidelity. (b) Fidelity tolerance to ZZ crosstalk for both the JPM gate and the CZ gate. The spectator qubit is initialized in $|1\rangle$ state or $|0\rangle + |1\rangle$ state.

(2-2) Leakage channels

Error channels parasitic in the parity measurement circuit may further cause unwanted leakage like leakage to coupler state (details in our response to *Comment 5 of Reviewer #3*) and qubit higher excited state (details in our response to *Comment 4 of Reviewer #3*). We find that the leakage seems to be more serious for CZ gate (*Comment 5 of Reviewer #3*). This is because the coupler frequency is more close to the qubit frequency for CZ gate. To accurately compare the parity measurement fidelity of JPM gate and CZ gate in the experiment, we consistently maintain the operation frequency of the syndrome qubit Q_0 at around 3.988 GHz, while we uniformly adjust it to 3.988 GHz when executing the JPM gate and to around 4.243 GHz when executing the CZ gate for the data qubit. During the parity measurement circuit, this leakage effect may be further exacerbated, ultimately declining the parity fidelity. Similarly, CZ gate may be also easier to suffer from the leakage to qubit $|2\rangle$ states (*Comment 4 of Reviewer #3*) since diabatic CZ gate utilizes the swap procedure between $|11\rangle$ and $|02\rangle$ (or $|20\rangle$). In practice, these results reflect that the JPM scheme based on collective iSWAP will be more tolerant to such leakage. Nevertheless, we also believe that selecting a more appropriate parameter range to suppress leakage will further improve the fidelity of the CZ gate, thereby enhancing the parity measurement fidelity.

(2-3) Experimental limitations

In order to ensure a fair comparison between the JPM scheme and the CNOT schemes in the experiment, particularly considering the limitations of the four-qubit JPM scheme, we are only able to utilize $Q_0 - Q_4$ for the experimental analyses (see structure of sample in Fig. 2(a) in the main text). However, we find the performance of $Q_5 - Q_8$ may be better in the experiment since these qubits exhibit less Two-level systems (TLSs) and higher fidelities in implementing CZ gate with adjacent qubits, as can be found in Fig. S4 of the supplementary materials. Therefore, as mentioned by the referee, if there were no restrictions on comparing these two schemes, employing qubit pairs with better performance would further enhance parity detection fidelity.

In addition, when implementing collective-iSWAP based JPM unitary gate, all the qubits are operated at a single frequency, allowing for easier selection of a frequency with reduced TLSs and greater stability. Conversely, achieving similar characteristics for CZ gates is relatively challenging due to their reliance on multiple operating frequencies. Therefore, from this aspect, the JPM scheme can be beneficial to not only circumvent frequency crowding, but also facilitate multiqubit gate operations.

Finally, according to the above experimental analyses and simulations, combined with the referees' suggestions, we have made revisions to the relevant statements to improve the presentation and claim of our paper:

- In lines 17-18 in ABSTRACT, we revise the sentence to "... which shows comparable performance to the CNOT scheme."
- In line 92 in INTRODUCTION, we revise the sentence to "..., which are comparable to that of the CNOT scheme in this experiment."
- In line 339 in EXPERIMENTAL DEMONSTRATION, we revise the sentence to "... can be achieved comparable at around 95.2% (90.0%) for the iSWAP-based JPM scheme, ...".
- In the caption of Fig. 4, we revise the sentence to "... is on par with the CNOT scheme ...".

Comment 9

The term “maximally entangled state” mentioned in the manuscript is closely relevant with the entanglement measures and is not commonly acknowledged for general multiqubit states. I suggest the authors to replace the expression to “multi-qubit entangled state” for rigorousness.

Response:

We thank the referee for the valuable suggestion. We have revised our expression from "maximally entangled state" to "multiqubit entangled state" in lines 22, 383-384 and 413-414.

Comment 10

The authors claim the preparation of a four-qubit entangled state, it would be beneficial to state explicitly what the target entangled state is if an entanglement measure is unavailable.

Response:

We thank the referee for the careful reading and valuable suggestion. We have modified the corresponding statements in our revised manuscript as:

- In lines 420-424 in DISCUSSIONS, we add sentence "For instance, initializing \$|Q_2Q_1Q_0Q_4Q_3\rangle\$ to any one of the four initial states in Fig. 6(c) will eventually generate 4-qubit distant entangling state \$1/2(|1001\rangle+|1100\rangle+|0011\rangle+|0110\rangle\$ with specific phase correction for \$Q_2, Q_1, Q_4\$ and \$Q_3\$."
- In the caption of Fig. 6, we add sentence "At specific gate time (176 ns), 4-qubit distant entangling state for \$Q_2, Q_1, Q_4\$ and \$Q_3\$ [e.g. \$1/2(|1001\rangle+|1100\rangle+|0011\rangle+|0110\rangle)\$ ] can be generated with appropriate phase correction."

Response to Reviewer #2

Overall Comments

The authors present an interesting alternative scheme to measure joint parity of states in a setting very actively used in the superconducting circuit community, namely surface-code quantum error correction.

I found the concept very neat, using joint evolution (and the difference in rates of interaction coming from the "+1" in the stimulated emission of the bosonic operators) to distinguish odd from even parity states. In a way, this (and especially the population plots of Fig7(a)) reminded me of trapped ion MS gates, maybe there is a parallel to be made?

Response:

We thank the referee for his/her time and efforts in reviewing our manuscript. We are pleased to know that the referee recognizes our scheme as "*an interesting alternative scheme to measure joint parity of states in a setting very actively used in the superconducting circuit community.*" and "*found the concept very neat.*"

We also thank the referee for his/her pointing out the resemblance between trapped ion Mølmer-Sørensen (MS) gate and our protocol. The referee mention that the population evolution of the JEP scheme [redrawn in Fig. R2-1(f)] seems to be similar to the evolution of MS gate, and this is what we are not aware of actually. At first glance, they indeed look the same, and the similarity comes from that population of $|001\rangle$ and $|100\rangle$ in upper panel of Fig. R2-1(f) looks like the combined part of $|\downarrow\downarrow\rangle$ state and $|\uparrow\uparrow\rangle$ state [Fig. R2-1(b)]. However, we think the two processes are fundamentally different in theory. In practice, a recent interesting work in *PRX QUANTUM* 3, 040322 (2022) from Irfan Siddiqi's research group has already borrowed the idea of MS gate in trapped ion to experimentally realize the multiqubit sideband tone-assisted Rabi-driven gate based on superconducting qubit system, which is also quite different from our scheme and can be used for reference. In the following, we first briefly introduce the MS gate and our scheme and then we list major differences between these two processes.

(1) MS gate in trapped ion

The MS gate in the ion trap is implemented via exciting the collective vibration mode between ions. To active the gate process, the frequency, phase and intensity of lasers are precisely controlled to apply the bichromatic sideband interactions to each ion simultaneously, as shown in Fig. R2-1(a). Notice that during the gate operation, the phonons in the spatial model do not have an actual occupied population, and their major role is to offer interaction to ions, and eventually allow the second-order interaction between the two ions. For simplicity, the system Hamiltonian can be expressed as:

$$H = \sum_i v a_i^\dagger a_i + \frac{1}{2} \omega_0 \sum_i \sigma_z^i + \sum_i \frac{1}{2} \Omega_i (\sigma_+^i e^{i(\eta_i(a+a^\dagger)-f_i t)} + H.c.),$$

where v is trapping frequency, ω_0 is the difference in frequencies of the internal up and down states, Ω_i and f_i are the Rabi oscillation and frequency of the laser on the i th ion, η_i is

the Lamb-Dicke parameter. Here, the last term represents the interaction Hamiltonian between ions and lasers which can offer Jaynes-Cummings (JC) term and anti-JC term at the same time. After choosing appropriate system parameters, the unitary operator can be eventually utilized to generate two-qubit operation which is equivalent to the CNOT gate with extra single qubit operations. The typical population evolution of MS gate can be found in Fig. R2-1(b), finding that the population of $|\downarrow\downarrow\rangle$ state ($|\uparrow\uparrow\rangle$ state) is increased (decreased) from 0 (1) to 1 (0) at T_{gate} , passing through 0.5 where the two-qubit entangling state is generated.

FIG. R2-1. (a) Energy level diagram of MS gate from *New Journal of Physics* 10, 013002 (2008). A bichromatic laser field with frequencies ω_b , ω_r satisfying $2\omega_0 = \omega_b + \omega_r$ is tuned close to the upper and lower motional sideband of the qubit transition. The field couples the qubit states $|\downarrow\downarrow\rangle \leftrightarrow |\uparrow\uparrow\rangle$ via the four interfering paths shown in the figure. (b) Typical population evolution of MS gate in trapped ion from *Phys. Rev. Lett.* 121, 180502. Notice that here population of $|\downarrow\downarrow\rangle$ state ($|\uparrow\uparrow\rangle$ state) is increased (decreased) from 0 (1) to 1 (0) at T_{gate} , passing through 0.5 where the two-qubit entangling state is generated. (c) Energy level diagram of the Rabi-driven qubit states with n photons in the resonator and the sidebands at frequencies ω_r and ω_b from *PRX QUANTUM* 3, 040322 (2022). The red and blue sidebands drive two photon transitions that generate a population swap between the $|--, n\rangle$ and $|++, n\rangle$ levels. (d) Typical population evolution of Rabi-driven gate in superconducting qubit system from *PRX QUANTUM* 3, 040322 (2022). (e) Energy level diagram of the JEP scheme containing three qubits $Q_2 - Q_0 - Q_4$ in experiment modified

from Fig. 2(b) in the revised manuscript. (f) Typical population evolution in the 2-qubit JEP scheme with initial states $|Q_2Q_0Q_4\rangle$ prepared to be $|001\rangle$ (top panel) and $|100\rangle$ (bottom panel) modified from Fig. 6(a) in the revised manuscript. Notice that here population evolution of $|001\rangle$ and $|100\rangle$ is quite different since here we set $g_{02}/g_{04} = \tan(\pi/8)$.

(2) JPM scheme (or JEP scheme)

Our scheme is implemented via coupling control of multiqubit resonance evolution with the energy level diagram shown in Fig. R2-1(e). The system Hamiltonian containing three qubits ($Q_2 - Q_0 - Q_4$ in experiment) can be written as:

$$H = \frac{1}{2} \sum_i \omega_i \sigma_z^i + \sum_i g_i (\sigma_+^0 \sigma_-^i + \sigma_-^0 \sigma_+^i),$$

where ω_i is the qubit bare frequency, g_i is the coupling strength between Q_0 and Q_i . Here, the second term represents the interaction Hamiltonian between Q_0, Q_2 and Q_0, Q_4 which only supports JC term with anti-JC term suppressed by the rotating wave approximation (RWA). After choosing appropriate coupling strength, this scheme can not only support joint parity measurement but also support fast distant entangling state preparation as mentioned in DISCUSSIONS section. For convenience, we redraw it in Fig. R2-1(f) with the top panel initialized to be $|001\rangle$ state and the bottom panel initialized to be $|100\rangle$ state, finding that the population evolution of both situations is quite different due to the asymmetric coupling ratio for g_{02} and g_{04} .

(3) Comparisons between MS gate and JEP scheme

In summary, we think our scheme is different from MS gate as followed:

- The evolution for MS gate takes the form $\exp\{-i\sigma_x^0 \sum_i \sigma_x^i\}$, where the collective term $\sum \sigma_x^i$ already encodes the parity information (since after basis change, this is equivalent to $\sum \sigma_z^i$). Similar to this, in superconducting circuits, Nigg et. al. (*Phys. Rev. Lett.* **110**, 243604) have proposed to realize a unitary $\exp\{-ia^\dagger a \sum \sigma_z^i\}$. However, for our scheme, the ancilla undergoes oscillations based on parity $\sum \sigma_z^i$. Moreover, the interaction here is of the JC type, $a^\dagger \sigma^- + a \sigma^+$ [Eq. (1) in the manuscript], which conserves total excitation not the parity. Data qubits and ancilla qubit exchange one excitation to each other at a frequency correlated to how many qubits (both data and ancilla) are initially in state $|1\rangle$. Because of the Rabi oscillation occurred in JC type interaction, [Eq. (3) in the manuscript], data qubit starts with k qubits in $|1\rangle$, may end up in $k+1, k, k-1$ excitations (Please refer to our reply in *comment 6* of *Reviewer #1*). To detect the parity, we want to align the Rabi oscillation such that, the ancilla state $|1\rangle$ is correlated with even parity of data qubits, which is not that straightforward.
- Spatial mode $|n\rangle$ in MS gate offers second-order interaction to ions without directly participating into the evolution of qubit states, therefore the population of the spatial mode remains zero during the gate process. However, our scheme directly utilizes the qubit-qubit first-order interaction to generate multiqubit resonance evolution, therefore the middle (syndrome) qubit cannot be viewed as the spatial mode and the population of the middle qubit will be evolved over time as shown in Fig. R2-1(f).
- The interaction term for MS gate is generated via laser-ion interaction and the gate Hamiltonian is equivalent to XX interaction. However, the interaction term for our scheme comes from the parasitic qubit-qubit neighbouring interaction and the gate Hamiltonian

is in essence a three-qubit resonance interaction Hamiltonian.

- The population evolution is actually quite different between our scheme and MS gate as shown in Fig. R2-1(b) and (f). On the one hand, our scheme is a three-qubit joint evolution process and only at the specific time under appropriate system parameters, the three-qubit state can be viewed as a distant two-qubit entangling state regardless of the middle qubit state. On the other hand, the population evolution trend as the referee mentioned is also different since $|\uparrow\uparrow\rangle$ state for MS gate in Fig. R2-1(b) is changed from 1 to 0, while $|001\rangle$ state for our scheme in top panel of Fig. R2-1(f) is changed from 1 to 1 as an example, let alone the population evolution in the bottom panel. Nevertheless, at specific gate time, both schemes can similarly generate two-qubit entangling state.
- The middle qubit in JEP scheme can be excited or utilized to prepare other useful multiqubit state while spacial mode in MS gate only functions as a medium for providing interaction to ions.
- MS gate for superconducting qubit system has already been realized in *PRX QUANTUM 3, 040322 (2022)* from Irfan Siddiqi's research group. In their paper, a shared CPW resonator between each qubit is functioned as the spacial mode and the gate is implemented via using bichromatic sidebands as depicted in Fig. R2-1(c). It can be found that the realization of the energy level diagram in the superconducting system is slightly different from that in trapped ion due to the difference in frequency range, but the evolution process of the gate in Fig. R2-1(d) is similar to that in trapped ion.

Comment 1

The authors make a strong point in comparing their method to that of a CNOT-based scheme (decomposed into CZ gates). On the one hand, while the comparison is interesting, I don't see the emphasis on better performance as necessary for the message of the paper. Indeed, this new scheme's working principle is interesting no matter what, as long as fidelities are comparable, which seems to be the case here. On the other hand, I cannot follow the arguments being made. The CZ-based CNOT scheme has multiple possible implementations (especially when given a tunable coupler), and its standard interaction time is also $\pi/\sqrt{2}g$ from working in the second excitation manifold (as here, line 163). Where is the gain? Sure, one needs to apply two (four) CZ for a weight-2(4) parity check, but isn't that the same for the unitary U described here (line 171 and fig4)?

The duration of the total protocol in both cases is not mentioned explicitly (only for the JPM scheme, lines 314-316), but the CZ was made comparable in time to the unitary gate (fig4, caption). Where was the claimed speed gain lost? Are the authors able to stand behind the statements in line 84-87?

Response:

1. We thank the referee for his/her thoughtful reading and valuable questions. We also thank the referee for his/her profound insights and positive comments on the innovation and significance of our proposed JPM scheme as "..., while the comparison is interesting, I don't see the emphasis on better performance as necessary for the message of the paper. Indeed, this new scheme's working principle is interesting no matter what, as long as fidelities are comparable, which seems to be the case here". Based on these suggestions, we have made

the following revisions accordingly to the manuscript:

- In lines 17-18 in ABSTRACT, we revise the sentence to "... which shows comparable performance to the CNOT scheme."
- In line 92 in INTRODUCTION, we revise the sentence to "..., which are comparable to that of the CNOT scheme in this experiment."
- In line 339 in EXPERIMENTAL DEMONSTRATION, we revise the sentence to "... can be achieved comparable at around 95.2% (90.0%) for the iSWAP-based JPM scheme, ...".
- In the caption of Fig. 4, we revise the sentence to "... is on par with the CNOT scheme ...".

2. We are not aware of and feel sorry that our inaccurate description in the original manuscript and omissions in data presentation, which led to the misunderstandings and confusion from the referee (the four successive questions raised in the *Comment 1*) about the time acceleration of our JPM scheme. In practice, the JPM scheme, taking advantage of the controllable collective behavior, promotes theoretically $\sqrt{2}$ times sped up over the CNOT scheme based on the homogeneous quantum gate, i.e., collective iSWAP (collective CZ) gate in the JPM scheme is usually $\sqrt{2}$ times faster than iSWAP (CZ) gate in the CNOT scheme. In the following, we elaborate the time consumption in both scheme and explain the applications in our experiment to together clarify the four questions the referee raised.

(1) Time consumption on unitary gate

In the THEORETICAL MODEL section, we derive that the system Hamiltonian of the JPM scheme in the interaction picture with the two-level approximation can be expressed as:

$$H_s = g\sigma_0^{n, n+1}(\sigma_1^{10} + \sigma_2^{10}) + H. c.,$$

where $\sigma_i^{n, m} = |n\rangle_i \langle m|$ is the $|m\rangle \rightarrow |n\rangle$ transition matrix element for qubit i . For instance, $n = 0$ for collective iSWAP gate and $n = 1$ for collective CZ gate. Therefore, the JPM unitary gate can then be acquired as $U_{siSWAP}(U_{sCZ})$ at $\sqrt{2}gt = \pi$ ($\sqrt{2}Gt = \pi, G = \sqrt{2}g$) under collective iSWAP (collective CZ) operation. Compared to the CNOT scheme, where iSWAP (CZ) gate is generated at specific gate time $gt = \pi/2$ ($Gt = \pi, G = \sqrt{2}g$), the JPM scheme is typically $\sqrt{2}$ times sped up due to the collective behavior (i.e. $g \leftrightarrow \sqrt{2}g, G \leftrightarrow \sqrt{2}G$). We should notice that although the single iSWAP gate in the CNOT scheme only consumes half of the whole swap time, but when consider the overall parity measurement circuit time, we still find the acceleration of JPM scheme. For a more detailed comparison, please refer to Table. R2-1, which records the time consumption for the unitary gate based on the two schemes.

Now, for the first question the referee mentioned, it is right that the CZ gate time is theoretically equivalent to the collective iSWAP gate time with both $\sqrt{2}gt = \pi$, but the time gain of JPM scheme here refers to the utilization of homogeneous gate operation compared with the CNOT scheme, e.g., collective CZ gate ($2gt = \pi$) in JPM scheme and CZ gate ($\sqrt{2}gt = \pi$) in CNOT scheme. We also recommend the referee to check out our response in *Comment 5 for Reviewer #3*, where we verify that the collective iSWAP gate in the JPM scheme can be implemented faster after fully exploiting the leakage interference to approach the speed limit compared with the CZ gate, thereby further reducing the circuit time in experiment.

(2) Time consumption on parity measurement circuit

When it comes to the time acceleration of the JPM scheme in the parity measurement circuit compared to the CNOT scheme, we would like to discuss on several situations.

① Minus phase correction with another $U_s = U_{siSWAP}(U_{sCZ})$

As shown in Eq. (5) and Fig. 1(a) in the revised manuscript, we introduce another U_s (stage iv) to undone the minus phase in the Dicke states, e.g., $|D_1\rangle$ and $|D_2\rangle$ [last line in Fig1.(c)]. Then the circuit time \tilde{T} of parity measurement can be summarized in Table. R2-1. We find that with the homogeneous gate operation (i.e., collective iSWAP \leftrightarrow iSWAP or collective CZ \leftrightarrow CZ), the circuit time \tilde{T} in the JPM scheme are usually $\sqrt{2}$ times faster than the corresponding CNOT scheme, i.e., $\tilde{T}_{iSWAP-CNOT} = \sqrt{2}\tilde{T}_{siSWAP-JPM}$, $\tilde{T}_{CZ-CNOT} = \sqrt{2}\tilde{T}_{sCZ-JPM}$, indicated as the black dotted arrow, which may further clarify the first and second questions the referee raised.

As for the third and the fourth questions about the time consumption in the experiment, based on the surface code architecture and the consideration of not introducing additional energy levels (e.g., state $|2\rangle$), we choose the collective iSWAP-based JPM scheme to demonstrate the parity measurement, which is $\sqrt{2}$ times faster than the iSWAP-based CNOT scheme but seems to have the same \tilde{T} with CZ-based CNOT scheme, shown as the red dotted circle in the Table. R2-1. However, when the collective CZ-based JPM scheme is chosen, we will still get $\sqrt{2}$ times sped up over the CZ-based CNOT scheme. Meanwhile, we feel sorry about the omissions in time consumption for CZ-based CNOT scheme in the article and we have made revisions as listed later.

Scheme	CNOT		JPM	
Gate Type	iSWAP	CZ	Collective iSWAP	Collective CZ
Gate Time	$\frac{\pi}{2g}$	$\frac{\pi}{\sqrt{2}g}$	$\frac{\pi}{\sqrt{2}g}$	$\frac{\pi}{2g}$
Circuit Time \tilde{T} (2-qubit case)	$\frac{2\pi}{g}$	$\frac{2\pi}{\sqrt{2}g}$	$\frac{2\pi}{\sqrt{2}g}$	$\frac{\pi}{g}$
Circuit Time \tilde{T} (4-qubit case)	$\frac{4\pi}{g}$	$\frac{4\pi}{\sqrt{2}g}$	$\frac{4\pi}{\sqrt{2}g}$	$\frac{2\pi}{g}$

TABLE. R2-1. Circuit time comparison between the 2-qubit (4-qubit) JPM scheme and CNOT scheme. Here, the circuit time calculated in the table mainly involves the entangling gate time for intuition with ignoration of single-qubit gate time. The JPM scheme promotes $\sqrt{2}$ times sped up over the CNOT scheme based on the homogeneous gate type, indicated as the black dotted arrow. The gate type circled by the red dashed line is utilized in the experiment.

② Minus phase elimination with post-selection (for reference, not included in the paper)

FIG. R2-2. Joint parity measurement protocol with post-selection, where the syndrome qubit (Q_0) is initialized in state $|0\rangle$ and the data qubits (Q_1, Q_2) are assumed to be $|\psi^k\rangle$. A

three-qubit JPM gate U_s with time duration $\pi/(\sqrt{2}g)$ is turned on between two Hadamard gates on Q_0 . Then, parity detection is executed on the syndrome qubit, followed by a digital σ_z operation on either of the two data qubits if the syndrome qubit is detected as flipped, or σ_x operations on both data qubits if the syndrome qubit remains unchanged.

In fact, we did not mention another potential time advantage for 2-qubit JPM scheme in the article. Instead of adding the second U_s , the minus phase can also be eliminated with post-selection, i.e., operating on the data qubits based on the outcome of the parity measurement on the syndrome qubit, as shown in Fig. R2-2. The overall protocol can then be expressed as:

$$H_0 U_s H_0 = \sigma_0^x (|00\rangle\langle 00| - |11\rangle\langle 11|) - I_0 (|D^1\rangle\langle D^1| - |d^1\rangle\langle d^1|).$$

It can be found that, when the syndrome qubit is detected as flipped, indicating that the input state of data qubits is in an even number, a digital σ_z operation on either of the two data qubits can be performed to cancel out the minus phase. Similarly, when the syndrome qubit remains unchanged, indicating that the input state of data qubits is in an odd number, σ_x operations can be executed on both data qubits. Under this circumstance, the JPM scheme gains further time advantage over the traditional CNOT scheme. Specifically, the circuit time \tilde{T} of the JPM scheme gets $2\sqrt{2}$ times faster than the corresponding CNOT scheme, as shown in Table. R2-2. Furthermore, even for the collective iSWAP-based JPM scheme, the acceleration remains twice that of the CZ-based CNOT scheme, which may allow for higher fidelity.

In practice, this method is promising for boundary parity detection in rotated surface code architecture. However, this protocol cannot be used in the 4-qubit JPM scheme owing to the lack of proper post-selection operators to eliminate the minus phase. Besides, this protocol needs more theoretical analyses when applied in the surface code. Therefore we did not include this part in the article to ensure the consistency of the content, but the corresponding idea can be utilized as reference for the referee.

Scheme	CNOT		JPM	
	iSWAP	CZ	Collective iSWAP	Collective CZ
Gate Type	iSWAP	CZ	Collective iSWAP	Collective CZ
Gate Time	$\frac{\pi}{2g}$	$\frac{\pi}{\sqrt{2}g}$	$\frac{\pi}{\sqrt{2}g}$	$\frac{\pi}{2g}$
Circuit Time \tilde{T} (post selection)	$\frac{2\pi}{g}$	$\frac{2\pi}{\sqrt{2}g}$	$\frac{\pi}{\sqrt{2}g}$	$\frac{\pi}{2g}$

TABLE. R2-2. Circuit time comparison between the 2-qubit JPM scheme and CNOT scheme based on the post-selection protocol (not included in the article). The time advantage of the JPM scheme will be further enhanced to $2\sqrt{2}$ times over the CNOT scheme, denoted as the black dotted arrow.

Finally, in order to eliminate the potential confusion about the time acceleration and improve the presentation in the article, we have made the corresponding revisions to our main text as:

- In Fig. 1, we add Fig. 1(d) to clarify the time consumption for the JPM scheme and CNOT scheme. The corresponding caption for Fig. 1(d) is described as "(d) Circuit time comparison between the 2-qubit JPM scheme and CNOT scheme (the 4-qubit case has a

similar result, except that all the circuit time \tilde{T} need to be doubled). Here, the circuit time calculated in the table mainly involves the entangling gate time for intuition while ignoring single-qubit gate time. The JPM scheme promotes $\sqrt{2}$ times sped up over the CNOT scheme based on the homogeneous gate type, indicated as the black dotted arrow. The gate types circled by the red dashed line are utilized in the experiment."

- In lines 86-90 in INTRODUCTION, we revise the sentence to "Meanwhile, this controllable collective behavior, which can be applied to both the iSWAP and CZ operations architecture, promotes $\sqrt{2}$ times sped up over the corresponding CNOT scheme based on the homogeneous gate operation type."
- In lines 219-229 in THEORETICAL MODEL, we revise the sentence to "..., the effective entangling gates in the JPM scheme are usually $\sqrt{2}$ times faster than that of the corresponding CNOT scheme with the homogeneous gate operation type, indicated as the black dotted arrow in Fig. 1(d), thus releasing the burden of iSWAP-based CNOT scheme with complex circuit composition. Besides, the iSWAP-based JPM scheme utilized in the following experimental analysis [marked as the red dotted circle in Fig. 1(d)] does not introduce any additional energy levels, which indicates its consistence with the surface code architecture."
- In lines 334-336 in EXPERIMENTAL DEMONSTRATION, we revise the sentence to "We find that compared to the 92.9% (87.9%) for the CZ-based CNOT scheme with the entire circuit time of 220 ns (365 ns) in the measure-Z and measure-X procedures, ...".
- In lines 432-434 in DISCUSSIONS, we revise the sentence to "..., and thus is $\sqrt{2}$ times sped up over the commonly-used CNOT scheme for the homogeneous gate operation type."

Comment 2

I found the clarity in the presentation of the manuscript lacking:

- the theoretical model section is very heavy, and doesn't provide much insights into the physics happening (which I believe should be relatively easy to describe, as common hybridized modes of the system which undergo oscillations based on the total parity?)
- Many figures have texts that are too small to read (1a, 1b, 2b, 8, ...).
- It is unclear what one should look at in Table Fig.1c (is Ψ given at the interaction time? what do colors depict? ...).
- Fig 1d, 1e are lacking initial state descriptions, the color labeling is not obvious.
- Fig 7a, 7c don't have axis labels.
- Fig 8 needs proper stages temperature, instead of a vague gradient.
- is Fig 2c really a pulse sequence for the JPM scheme? Or only for the JPM unitary, without undoing the additional minus phase?
- line 295-296: "Figure 2(e) is the typical population oscillations in 4-qubit JPM scheme". See above point: Is it really? Or some population evolution given specific calibration input states for the JPM unitary gate.
- I really hope the authors don't "optimize the results" of the experimental chi matrix to quote a high fidelity, but rather compute the chi matrix that represents the process happening on the device, taking SPAM errors into account. (lines 272-274 and Fig 3 caption).
- lines 389-390: clearly populations plots can't convince that a Bell state is "generated no matter what states Q0 is in" (coherences are missing). While I can follow the expectation, the experimental evidence is lacking.
- Fig 5 is barely referenced, and unless a proper discussion of the infidelity is made, it should just be removed, or placed in the supplementary material files.

Response:

We thank the referee for the insightful comments and valuable suggestions. We reply to these comments separately.

Comment 2-1

the theoretical model section is very heavy, and doesn't provide much insights into the physics happening (which I believe should be relatively easy to describe, as common hybridized modes of the system which undergo oscillations based on the total parity?)

Response:

We agree with the referee's comment that the original theoretical description may be too heavy and difficult to follow, thus we have revised the manuscript to improve readability and clarify the parity-encoded oscillation. Besides, considering the fact that our protocol is actually not the same with the MS gate (as described in *Overall Comments*), we believe a detailed analysis of theoretical model and a clear description of the parity encoding process

is necessary for this new parity measurement scheme. Therefore, we redraw Fig. 1 and fully reconstruct the THEORETICAL MODEL section in our revised manuscript to be more accessible and reduce the heaviness of the section while still providing the necessary information.

Comment 2-2

Many figures have texts that are too small to read (1a, 1b, 2b, 8, ...).

Response:

We have revised all the figures to ensure the legibility by appropriately enlarging the texts, especially those are too small to read.

Comment 2-3

It is unclear what one should look at in Table Fig.1c (is Psi given at the interaction time? what do colors depict? ...).

Response:

We sincerely apologize for the confusion caused by our unclear presentation in the Fig. 1 in the original manuscript. Our intention was to convey the detailed transition rates g_{\pm}^k [Eq.(2)] for different initial states $|\psi^k\rangle$ and the resulting qubit states $|\phi\rangle$ after applying $H_0U_sH_0$ [Eq.(4)]. Here, $|\psi^k\rangle$ denotes the two-qubit Dicke basis for the data qubits, with k being the number of excitations among data qubits. The blue (red) background was used to indicate even (odd) parity of the data qubit. However, it is clear that we did not effectively convey this information effectively. Therefore, in order to better express our ideas, we revise the Fig. 1, including adding a parity labeling row in Fig. 1(c) and adjusting the order of Fig. 1(a) and (b). Additionally, we add labels in Fig. 1(a) to indicate the stages that are described in detail in Fig. 1(b) and (c). Moreover, we rewrite the THEORETICAL MODEL section correspondingly in the revised manuscript. Next, we will provide a detailed description of the modifications that have been made to Fig. 1 (redrawn in Fig. R2-3 here for convenience).

Firstly, we rearrange the order of Fig. R2-3(a) and (b). In Fig. R2-3(a), we present the whole circuit diagram of the JPM scheme for parity measurement to give readers an intuitive understanding, and then provide detailed explanations for some intermediate steps in (b) and (c). Specifically, we label each stage in the JPM circuit diagram of Fig. R2-3(a) with i, ii, iii, iv. Without loss of generality, we take $|0, \psi^k\rangle$ (in the order of $|Q_0, Q_1, Q_2\rangle$) as the initial state, and label the intermediate state $|\phi\rangle$ after state iii. Then, similar to the stimulated emission, in Fig. R2-3(b), we illustrate the evolution of the qubits at stage ii. Depending on the state of syndrome qubit ($|0\rangle$ or $|1\rangle$), the data qubits ($|\psi^k\rangle$) experience different oscillations, denoted by $|0\rangle|\psi^k\rangle \leftrightarrow |1\rangle|\psi^{k-1}\rangle$ and $|1\rangle|\psi^k\rangle \leftrightarrow |0\rangle|\psi^{k+1}\rangle$ with different transition rates g_-^k and g_+^k , respectively. In the table of Fig. R2-3(c), we detail the transition rates for different Dicke states $|\psi^k\rangle$ of the data qubits. For the situation where the excitation number of the data qubits is in even (odd) parity, we fill it with blue (red) background. At the bottom row of the table, we present the intermediate state $|\phi\rangle$ of the qubit after the third stage iii. And to

eliminate the accumulated π phase appeared in $|\phi\rangle$ for the input state $|0, D_1\rangle$ and $|0, D_2\rangle$, we introduce another U_s in state iv. Moreover, in order to illustrate the time consumption of the CNOT and JPM parity measurement scheme more clearly, we add Fig. R2-3(d) here. We mark the demonstration schemes used in the experiment, i.e., the CZ-based CNOT scheme and iSWAP-based JPM scheme, with red dashed circles, while indicating the corresponding time acceleration with a black dotted arrow. We also make detailed modifications to the captions of this figure and the corresponding description in the main text. We believe that these modifications will enable readers to quickly grasp the essential information conveyed in Fig. R2-3.

FIG. R2-3. (a) Joint parity measurement protocol for two qubits (Q_1, Q_2), where the syndrome qubit (Q_0) is initialized in state $|0\rangle$ and the data qubits are assumed to be $|\psi^k\rangle$ without loss of generality. Here, $|\psi^k\rangle$ denotes the collective two-qubit Dicke basis for the data qubits, with k being the number of $|1\rangle$ among data qubits. A three-qubit JPM gate U_s with time duration $\pi/(\sqrt{2}g)$ is turned on between two Hadamard gates on Q_0 . Then a second JPM gate is applied to undo the possible extra phase (refer to the last line of Table(c)). Even (odd) parity is mapped onto the measurement outcome 1 (0) of Q_0 . (b) When the JPM gate is turned on, depending on the syndrome qubit being $|0\rangle$ ($|1\rangle$), $|\psi^k\rangle$ undergoes two different oscillations, i.e., $|0\rangle|\psi^k\rangle \leftrightarrow |1\rangle|\psi^{k-1}\rangle$ and $|1\rangle|\psi^k\rangle \leftrightarrow |0\rangle|\psi^{k+1}\rangle$ with corresponding frequency gg_-^k and gg_+^k . (c) The table lists the oscillation frequency (third and fourth line) and the intermediate state (last line) after stage iii in (a) for different data qubits input state $|\psi^k\rangle$ (the second line). The blue (red) background denoting the even (odd) parity of the data qubits. (d) Circuit time comparison between the 2-qubit JPM scheme and CNOT scheme (the 4-qubit case has a similar result, except that all the circuit time \tilde{T} need to be doubled). Here, the circuit time calculated in the table mainly involves the entangling gate time for intuition with ignoring single-qubit gate time. The JPM scheme promotes $\sqrt{2}$ times speed up over the CNOT scheme based on the homogeneous gate type, indicated as the black dotted arrow. The gate types circled by the red dashed line are utilized in the experiment.

Comment 2-4

Fig 1d, 1e are lacking initial state descriptions, the color labeling is not obvious.

Response:

Here we think that the referee refers to Fig. 2(d) and (e). We have added initial state description as "*The quantum state is initialized to $|001\rangle$ (top panel), $|010\rangle$ (middle panel), $|100\rangle$ (bottom panel) in (d) and $|00010\rangle$ (top panel), $|00100\rangle$ (middle panel), $|00001\rangle$ (bottom panel) in (e) with the definition as $|Q_2Q_0Q_4\rangle$ in (d) and $|Q_2Q_1Q_0Q_4Q_3\rangle$ in (e).*" in the caption of Fig. 2. Meanwhile, in order to identify the initial state more clearly, we add the color labeling right below the Fig. 2(d) and (e) in the revised manuscript.

Comment 2-5

Fig 7a, 7c don't have axis labels.

Response:

Following the valuable suggestion in *Comment 2-11*, we have moved Fig. 5 from the original manuscript to the Supplementary Materials. As a result, Fig. 7 (Fig. 8) in the original version has been renumbered as Fig. 6 (Fig. 7) and we have added axis labels to Fig. 6(a) and (c) in the revised manuscript.

Comment 2-6

Fig 8 needs proper stages temperature, instead of a vague gradient.

Response:

We have added temperature labels to all the stages in Fig. 7 in the revised manuscript.

Comment 2-7

is Fig 2c really a pulse sequence for the JPM scheme? Or only for the JPM unitary, without undoing the additional minus phase?

Response:

The referee is right that Fig. 2(c) is the pulse sequence for the JPM unitary gate without undoing the additional minus phase. We have made the corresponding revisions as:

- We update Fig. 2(c) to be more general to describe the calibration and implementation of JPM unitary gate.
- In the caption of Fig. 2, we change the original description in (c) to "*Pulse sequence for calibrating and implementing the 4-qubit JPM unitary gate (2-qubit JPM unitary gate with ignoring one of the segment)*" and add sentence to (d)(e) as "*... with the pulse sequence shown in (c).*"

Comment 2-8

line 295-296: "Figure 2(e) is the typical population oscillations in 4-qubit JPM scheme". See above point: Is it really? Or some population evolution given specific calibration input states for the JPM unitary gate.

Response:

The referee is right that Fig. 2(d) and (e) both represent the population evolution after calibrating the JPM unitary gate given specific input states. We have made corresponding revisions as:

- In lines 317-320, we change the original sentence to "Figure 2(e) is the typical population oscillations after calibrating 4-qubit JPM unitary gate with three different initial states and the pulse sequence shown in Fig. 2(c)."
- In the caption of Fig. 2, we change the original sentence in (d)(e) to "Typical experimental calibration results for (d) 2-qubit and (e) 4-qubit iSWAP-based JPM unitary gate with the pulse sequence shown in (c)."

Comment 2-9

I really hope the authors don't "optimize the results" of the experimental chi matrix to quote a high fidelity, but rather compute the chi matrix that represents the process happening on the device, taking SPAM errors into account. (lines 272-274 and Fig3 caption).

Response:

We agree with the referee that on the one hand the raw experimental QPT fidelity is important, on the other hand, we think the optimized measured QPT gate fidelity with eliminating SPAM error is also beneficial especially for analyzing error sources and further improving gate performance. Therefore, we present both QPT fidelity data in the revised manuscript as:

- In the caption of Fig. 3, we modify the description as "Quantum process tomography (QPT) of the 2-qubit JPM unitary gate with characterizing the experimental χ_{exp} , reaching an average 95.0% raw gate fidelity in 79 ns. We further extract the optimized QPT data with 98.5% gate fidelity after eliminating the SPAM error."
- In lines 291-299, we modify the sentence as "To finally figure out the fidelity of the JPM unitary gate, we carry out the quantum process tomography (QPT) to extract the experimental χ_{exp} for verification, reaching an average 95.0% raw gate fidelity in 79 ns. We further optimize the results by eliminating the influence of the state preparation error and the measurement error (SPAM error), and acquire an optimized QPT fidelity of 98.5% as shown in Fig. 3(c)."

Comment 2-10

lines 389-390: clearly populations plots can't convince that a Bell state is "generated no matter what states Q0 is in" (coherences are missing). While I can follow the expectation, the experimental evidence is lacking.

Response:

We agree with the referee and make the corresponding revisions in the revised manuscript to clarify the statement:

- In line 411-417, we modify the description as "Clearly, at specific gate time $t_{JEP} \equiv \pi / \sqrt{(g_1^2 + g_2^2)}$ (faster than standard method by iSWAP gates or CZ gates), distant multiqubit entangled Bell state $|\psi^k\rangle \equiv \frac{1}{\sqrt{2}}(|00\rangle \pm |11\rangle)$ for Q_2 and Q_4 are directly generated as depicted in Fig. 6(b), with the extracted average 98.6% state fidelity by performing QST measurement."
- In the caption of Fig. 6, we modify the sentence as "(b) QST measurement of the distant Bell state of Q_2 and Q_4 at the gate time with Q_0 initialized to be in ground state."

Comment 2-11

Fig 5 is barely referenced, and unless a proper discussion of the infidelity is made, it should just be removed, or placed in the supplementary material files.

Response:

We agree with the referee and move the content of ERROR ANALYSES including Fig. 5 in the original main text to the supplementary materials in the revised manuscript. Moreover, we further carry out the error analyses for the JPM scheme (especially the collective-iSWAP based JPM scheme) in detail, exploring the error sources in the transversal and longitudinal parity measurement circuit based on the surface code architecture. For error sources in the transversal parity measurement circuit, we make a discussion about decoherence and calibration imperfection, while for error sources in the longitudinal parity measurement circuit, we make a discussion about crosstalk (especially ZZ crosstalk) and leakage (leakage to coupler state and leakage to qubit higher excited state).

To more clearly show the modifications and changes we have made, we list the corresponding modification points as follows:

- We remove ERROR ANALYSES in the main text and transfer it to the supplementary materials with new data to fully explore the error sources for the JPM scheme with comparison to the CNOT scheme.
- We move Fig. 5 in the original main text to the Fig. S6 in supplementary materials to illustrate error sources in the transversal parity measurement circuit.
- We add Fig. S7 in the supplementary materials to explore ZZ crosstalk in the surface code architecture.
- We add Fig. S8 in the supplementary materials to explore leakage interference methods for the JPM gate and the CZ gate.

Response to Reviewer #3

Overall Comments

The manuscript by Huai, Bu, Gu et al. focuses on the implementation of joint parity measurements (JPM), operations designed to map the parity information of multiple data qubits to the state of a syndrome qubit. The primary result is the experimental implementation of a fast, joint parity measurement scheme based on stimulated emission that requires a superconducting quantum processor architecture with tunable couplers. This interaction is achieved by controlling the collective behavior of multiple data qubits, ensuring resonance and consistent effective coupling strengths between all pairs involved, and benchmarked using quantum state tomography (QST) and quantum process tomography (QPT). Here, it is worth highlighting the strong emphasis placed on the use of this scheme for surface-code based quantum error correction applications, something that serves as motivation to the authors throughout the manuscript and generates multiple comparisons, despite the simultaneous demonstration of a joint entangling state preparation (JEP) scheme.

The main advantage of the scheme is stated to be a $\sqrt{2}$ speed up over traditional CNOT-gate based parity measurement schemes, which commonly rely on two-qubit control-Z (CZ) gates to build comparable parity extraction circuits, despite a significant increase in the complexity of calibration and quantum control required for the proposed implementation. The paper is fairly well written and makes the topic accessible. I find the work to be scientifically sound. However, the limited breath of benchmarking presented for the scheme, particularly in assessing the effect of correlated error sources, such as leakage, in a scheme where error propagation would be a source of particular concern, prevent my immediate recommendation for publication in Nature Communications.

Nevertheless, I would like to give the authors the opportunity to further clarify various statements made in the text, and further motivate the novelty and usefulness of the proposed scheme.

Response:

We thank the referee for his/her time and efforts in reviewing our manuscript. We are pleased to receive the positive evaluation of our work from the referee as "*The paper is fairly well written and makes the topic accessible. I find the work to be scientifically sound.*" We also appreciate the referee for giving us "*opportunity to further clarify various statements made in the text, and further motivate the novelty and usefulness of the proposed scheme.*" We follow the referee's insightful suggestions and make major revisions in the revised manuscript.

Comment 1

In the manuscript, it is stated that the proposed scheme “requires no additional device modifications or circuit complexity”. I believe this statement greatly oversimplifies the cost of such a scheme, given its requirement on tunable coupling elements, and particularly because of the additional complexity in calibration. Considering the stringent requirements on calibration, including the need to ensure resonance between all qubits involved (when all qubits are subject to dispersive shifts), and the additional complexity in quantum control sequences, required to maintain consistent effective coupling strengths in adjacent couplers, do the authors believe their original statement should stand? Moreover, could the authors comment on the benefits of the theoretically achievable $\sqrt{2}$ speed up over traditional CNOT-based measurement schemes, particularly considering the additional complexity introduced?

Response:

We thank the referee for the thoughtful reading and insightful comments. We reply to the two comments separately.

Comment 1-1

In the manuscript, it is stated that the proposed scheme “requires no additional device modifications or circuit complexity”. I believe this statement greatly oversimplifies the cost of such a scheme, given its requirement on tunable coupling elements, and particularly because of the additional complexity in calibration. Considering the stringent requirements on calibration, including the need to ensure resonance between all qubits involved (when all qubits are subject to dispersive shifts), and the additional complexity in quantum control sequences, required to maintain consistent effective coupling strengths in adjacent couplers, do the authors believe their original statement should stand?

Response:

We sincerely thank the referee for this valuable question. We feel sorry and are not aware of that our statement, especially "*circuit complexity*", in the original manuscript clearly misled the referee and resulted in a misunderstanding of the intended meaning. Therefore, here we will first explain the representation of "*device modifications or circuit complexity*" to eliminate potential misunderstandings by the referee. Second, we discuss the calibration procedure of our scheme in detail and make a comparison with the standard CNOT scheme to address the concerns about the calibration complexity. Finally, we return to our original statement and make a revision to improve the expression.

(1) Device structure: tunable transmon qubits with tunable couplers

In the main text, we mention that our joint parity measurement (JPM) scheme is compatible with the existing superconducting quantum processor architecture, requiring no additional device modification (additional extra circuit elements based on the commonly-used superconducting circuit design). In practical applications, our device structure, which utilizes superconducting transmon qubits with tunable couplers, is capable of providing effective coupling strength for qubit-qubit gate operations, while simultaneously suppressing static ZZ

crosstalk. This architecture has been recognized as one of the most promising architectures for surface code quantum error correction in recent years. Google has utilized this architecture to achieve quantum supremacy (*Nature* 574, 505–510 (2019)) and carry out quantum error correction research based on surface code (e.g. *Nature* 614, 676–681 (2023)). Therefore, our original intention behind the statement "requires no additional device modification" was to convey that, the JPM scheme is free of requiring the additional circuit elements compared to some previous work like *Science Advances* 4, eaau1695 (2018) where multiqubits are directly coupled to the nonlinear resonator and even requiring additional low-Q filters for the 4-qubit parity measurement, and the commonly-used device structure can naturally be utilized to realize our proposed scheme.

(2) Circuit complexity: parity measurement circuit

The circuit complexity described in the original statement is intended to demonstrate that the parity measurement circuit for JPM scheme is similar to that for CNOT scheme with the homogeneous gate operation type and simple circuit construction. However, we feel sorry that our incomplete and inaccurate description prompted the referee associate it with a broader meaning of circuit complexity like calibration complexity and control complexity, which is actually not our original intention, i.e., the parity measurement circuit complexity. To improve the presentation and clarify the statement, we have made the corresponding revisions in the main text, as listed in the end of the reply to this comment.

(3) Calibration of JPM scheme

We agree with the referee that the calibration procedure of the JPM scheme is more complex than the CNOT scheme since the JPM unitary gate is in principle equivalent to a three-qubit gate operation while CZ gate in CNOT scheme is a two-qubit gate operation, and thus additional complexity in control sequences for JPM scheme is inevitable. However, we believe that the calibration complexity of the JPM scheme is not a huge improvement compared with the CNOT scheme. Meanwhile, the JPM scheme may have certain advantages in some aspects:

① The JPM unitary gate requires accurate control of both frequencies and coupling strength as the referee mentioned, however, several previous work has investigated such resonance-based multiqubit gates for calibration and optimization (e.g. supplementary materials of *Nature Communications* 14, 1971 (2023) provides the calibration of Z Pulse Amplitude for N -qubit resonance). In practice, in CALIBRATION PROCEDURE FOR JOINT PARITY MEASUREMENT section of our supplementary materials, we have listed the calibration flow for JPM unitary gate, which utilized the Z-crosstalk matrix and numerical simulation results to help optimize the experimental results.

② In terms of the overall calibration complexity, we only need to calibrate one three-qubit JPM unitary gate each time for JPM scheme, while need to calibrate two pairs of two-qubit CZ gates for CNOT scheme. Additionally, all qubits in the JPM scheme share the same operation frequencies, while in the CNOT scheme, two pairs of CZ gates need at least three operation frequencies, which may increase the influence of TLSs on the selection of frequencies and gate operation.

③ The two pairs of CZ gates in CNOT scheme may require further global optimization (e.g., to reduce the impact of crosstalk) such as circuit benchmarking to improve the overall circuit fidelity.

Finally, we think our JPM scheme may not introduce too much additional complexity on the calibration process and we have changed the corresponding sentences in our revised

manuscript to improve the description followed by the referees' suggestions as:

- In lines 23-27 in ABSTRACT, we have revised the sentence as "*This strategy, combined with the superconducting qubit system with tunable couplers, reveals tremendous potential and applications in the surface code architecture without adding extra circuit elements.*"
- In lines 93-97 in INTRODUCTION, we have revised the sentence as "*Our joint parity measurement scheme, being compatible with the existing superconducting quantum processor architecture with tunable couplers, requires no additional device modification or circuit complexity of parity measurement.*"

Comment 1-2

Moreover, could the authors comment on the benefits of the theoretically achievable $\sqrt{2}$ speed up over traditional CNOT-based measurement schemes, particularly considering the additional complexity introduced?

Response:

We thank the referee for this valuable question and we interpret the referee's intentions to ask whether the JPM scheme with theoretical time acceleration would still have advantages after taking calibration complexity as mentioned in *Comment 1-1* into consideration. To clarify this concerns, we elaborate from several aspects.

First of all, theoretically achievable $\sqrt{2}$ times sped up over CNOT scheme is an inherent advantage of the JPM scheme, which cannot be overlooked or replaced by any experimental complexity. This capability may effectively reduce the time consumption of the parity detection circuit, thereby enhancing the depth of the quantum circuit and improving the fidelity of the parity measurement.

Next, the influence of the additional experimental calibration and control complexity can be gradually optimized with the development of the experimental technology, such as waveform optimization, instrument control optimization, and automatic calibration system etc. In practice, the fidelity of JPM scheme can be achieved comparable to CZ-based CNOT scheme under the current experimental conditions. We also believe that the JPM unitary gate deserves further exploration in the future to improve its calibration procedure and enhance its gate performance for parity measurement.

Furthermore, as we mentioned in *Comment 1-1*, we believe the calibration complexity of the JPM scheme is not a huge improvement compared with the CNOT scheme, instead, to a certain extent, the JPM scheme may be more convenient. For instance, In terms of the overall calibration complexity, we only need to calibrate one three-qubit JPM unitary gate each time for JPM scheme, while two pairs of two-qubit CZ gates for CNOT scheme. Also, the two pairs of CZ gates in CNOT scheme may require further global optimization such as circuit benchmarking to suppress the impact of crosstalk and improve the overall circuit fidelity.

Finally, if the referee is interested or still concerned about the time acceleration of the JPM scheme, we kindly suggest the referee to refer to our responses in *Comment 7 for Reviewer #1* or *Comment 1 for Reviewer #2*. In these comments, we provide further details on the time advantages of the JPM scheme and brainstorm another approach for time acceleration through post-selection to eliminate minus phase (not included in the paper). We think that

this approach may prove useful and have potential applications in parity measurement or even surface code architecture, with further enhanced capability for time acceleration.

Comment 2

In Figure 2 (d) and (e), the lack of a legend makes it hard to interpret the results. Could the various curves be more clearly labelled, perhaps in the style of Figure 7 (a) and (c)?

Response:

We thank the referee for careful reading and valuable suggestions. We have revised Fig. 2(d)-(e) in the main text with reference to the style of Fig. 6(a)(c) in the revised manuscript.

Comment 3

When assessing parity measurement schemes, it is crucial to quantify the effect of backaction on the data-qubit subspace, since this would be exacerbated in any real-world use case, given the cyclic nature of parity checks in stabilizer-based error correction codes. However, it could be hard to quantify such effects based purely on the results of QST and/or QPT presented. As such, can the authors comment on the repeatability and disturbance effects of the JPM scheme proposed?

Response:

We sincerely thank the referee for his/her valuable questions. We agree with the referee that "*..., it is crucial to quantify the effect of backaction on the data-qubit subspace, ...*" and "*..., it could be hard to quantify such effects based purely on the results of QST and/or QPT presented.*" Therefore, we re-measured our samples and added relevant experimental results in our revised manuscript to further verify the impact of parity measurement based on the JPM scheme on data-qubit subspace. In the following, we elaborate the experimental contents and our measurement method.

Inspired by *npj Quantum Information* 5,60669 (2019), we design two sets of experiments to investigate the disturbance effect induced by syndrome qubit measurement, as illustrated in Fig. R3-1(a). For instance, the left panel in Fig. R3-1(a) represents the reference group where QST measurements are simultaneously performed on both data qubits and syndrome qubit, while the right panel corresponds to the experimental group where QST measurements are only conducted on data qubits after the measurement of the syndrome qubit [red waveform in Fig. R3-1(a)]. It is worth noting that Dynamical Decoupling (DD) pulses with pulse optimization obtained by varying pulse intervals are employed here to mitigate dephasing influence on data qubits during syndrome qubit measurement. The only difference between these two groups lies in the relative order of measuring data qubits and syndrome qubit, enabling us to confirm the disturbance to data-qubit subspace through fidelity comparison. The results obtained from QST measurements are presented in Fig. R3-1(b), where the top panel illustrates the coherent state of the three qubits in the reference group, and the bottom panel characterizes the state of two data qubits conditioned on syndrome qubit measurement outcomes. By projecting the data qubits into the corresponding Bell states $|\Phi_+\rangle = (|01\rangle + |10\rangle)/\sqrt{2}$ and $|\Psi_+\rangle = (|00\rangle + |11\rangle)/\sqrt{2}$, respectively, for both syndrome

qubit measurement outcomes $|0\rangle$ and $|1\rangle$ in Fig. R3-1(c), we achieve fidelities (94.4% and 93.6%) that closely match those obtained by projecting the reconstructed three-qubit state onto its two-qubit subspace (95.8% and 96.4%). This level of agreement aligns well with the readout fidelity ($\sim 97.0\%$) of the syndrome qubit, indicating that parity measurement based on the proposed JPM scheme should have minimal impact on data qubits.

FIG. R3-1. (a) Circuit diagrams for verifying disturbance effects on data-qubit subspace based on the JPM scheme with reference group (experimental group) shown in left (right) panel. The Dynamical Decoupling (DD) pulses are utilized with optimized pulse intervals to mitigate dephasing effect during the measurement of syndrome qubit. (b) Measured expectation values of multiqubit Pauli operators for the reference group (top panel) and the experimental group conditioned on the syndrome qubit in $|0\rangle$ state and $|1\rangle$ state (bottom panel). Here, the numbers on the horizontal axis represent the order of the Pauli set in terms of I, X, Y, Z for each qubit. (c) The corresponding density matrix of Bell states $|\Phi_+\rangle = (|01\rangle + |10\rangle)/\sqrt{2}$ and $|\Psi_+\rangle = (|00\rangle + |11\rangle)/\sqrt{2}$ measured in the experimental group conditioned on the syndrome qubit in $|0\rangle$ state and $|1\rangle$ state.

In practice, the verification of the disturbance effect from syndrome qubit measurement to the data-qubit subspace can further ensure the repeatability of the JPM scheme, as demonstrated in *npj Quantum Information* 5,60669 (2019). In our experiment setup, due to limitations such as the absence of Purcell filters on the quantum chip which leading to slow readout time (generally exceeding $2 \mu\text{s}$), lack of a fast reset process (constrained by frequency arrangement of resonators and qubits, and functional limitations of the currently used self-developed electronic control systems), it is challenging to directly verify repeatability considering potential qubit decoherence effects. However, based on our theoretical analysis mentioned in the main text, and corroborating the disturbance verification experiments, we may reasonably deduce that our proposed JPM scheme exhibits robust repeatability.

Finally, following the referee's insightful suggestions, we make the corresponding revisions and supplements to our paper:

- We add subsection *Disturbance effects on the data-qubit subspace* in lines 343-378 in the EXPERIMENTAL DEMONSTRATIONS to experimentally explore the disturbance effects from syndrome qubit measurement.
- We reconstruct Fig. 4 with new figures shown in Fig. 4(d)-(f), to provide additional clarification on the disturbance effects.
- We add the corresponding descriptions of the disturbance experiments in the caption of Fig. 4.

Comment 4

When comparing JPM to CNOT-based parity measurement schemes, the authors understandably place a lot of emphasis on average parity detection fidelity. However, given the outsized influence of correlated errors in the overall performance of a stabilizer-based error correction code, I would argue that quantifying the effect of correlated errors, particularly of leakage, is just as important, since the overall performance might otherwise be limited despite the higher parity detection fidelity. Could the authors comment on the effects of JPM on correlated error sources such as leakage?

Response:

We sincerely thank the referee for his/her insightful questions. We fully concur with the referee's comment that, in addition to considering the average parity fidelity as a measure of performance when exploring parity, it is also crucial to address correlated errors during the error correction process, as they significantly impact the overall effectiveness of error correction codes. Before addressing the specific queries regarding correlated errors, we would like to clarify that our original article only briefly touched upon the application of JPM in surface code, as we mentioned in line 386 in the original manuscript that "*Meanwhile, we should also mention that more researches such as leakage and error propagation in the JPM scheme are deserved to be explored for supporting the experimental feasibility in the surface code architecture.*" This indicates that our intention in this work focus on introducing a novel parity scheme, which has also been pointed out by *Reviewer #2*, "*Indeed, this new scheme's working principle is interesting no matter what, as long as fidelities as comparable, which seems to be the case here.*"

Besides, in our current experiment, it is difficult to provide a quantitative description of the effect of correlated errors due to several limitations. These include the size and connectivity of the chip, the absence of Purcell filters (leading to slow readout times with typically exceeding 2 μ s), the lack of a fast reset process due to the frequency arrangement of resonators and qubits, and functional limitations of the currently used self-developed electronic control systems. However, we believe that the JPM scheme holds enormous potential and deserves detailed investigation in future works on more suitable chips. We hope that this clarifies our original intention for writing the article, and we are also keen to share our thought on correlated errors.

In the surface code architecture based on the superconducting transmon qubits with tunable couplers, correlated errors may originate from leakage, crosstalk, control system and even cosmic-ray. The fundamental operations, such as single-qubit gates, entangling gates, and measurement are known to populate non-computational levels. Specifically, the leakage, concerning the entangling gates, primarily comes from the leakage from qubit states to the higher excited state (e.g. $|2\rangle$ state) or qubit states to the coupler states.

(1) Leakage

(1-2) Leakage to qubit higher $|2\rangle$ state

Inspired by the research in *Nature Communications* 12, 1761(2021) and recognizing the difficulty of directly quantifying leakage during normal operation, we can use the error detection during the stabilizer code's operation to identify and visualize undesired correlated errors in JPM scheme.

To visualize the pattern of errors that leakage produces, we can intentionally inject leakage (by a full $|1\rangle - |2\rangle$ rotation) after the first Hadamard gate in the JPM scheme on specific locations (data/syndrome qubit) and rounds. By analyzing the fraction of error detection events, we may observe a tail of correlated detection events over the lifetime of the leakage state, which is a consequence of the temporal correlation of the errors. It should be noted that the impact of leakage injection on adjacent qubits may differ depending on the type of qubit where the leakage is injected, i.e., data or syndrome qubit, due to the multi-qubit entangled gate.

To further quantify how the leakage that naturally arise during the stabilizer codes lead to correlated errors, we can also analyze the correlations between detection events using the error graph. A Pauli error affecting any operation in the repetition code based on JPM scheme should produce two detections (except at the spatial boundaries of the code). By computing the pairwise correlation probabilities p_{ij} between arbitrary pairs of detection nodes. We may observe the spacelike error, timelike error, spacetime error in the planar graph of p_{ij} matrix. First, an error on a data qubit usually produces a detection on the two neighboring syndrome qubits in the same round - a spacelike error. The exception is an error during the JPM gates, which may cause detection events offset in time and space - a spacetime error. Finally, an error on a syndrome qubit which does not propagate to a data qubit will produce detections in two subsequent rounds - a timelike error.

However, in a study published in *Nature Communications* 12, 1761(2021), it was confirmed that the reset protocol can effectively eliminate significant amounts of leakage in syndrome qubits, thereby reducing leakage in data qubits and preventing its further spread, leading to a substantial reduction in time-correlated tails of detection events, i.e., the long-range correlations are mostly removed. This reset protocol should be applicable to JPM as well, as the key mechanism - resetting syndrome qubits to block leakage propagation - is not dependent on the specific method used for implementing parity checks.

(1-2) Leakage to coupler state

In order to achieve faster entangling gate, the couplers should be biased to a lower frequency positions during multiqubit gate operations, potentially leading to energy exchange between qubits and couplers and resulting in leakage. We thoroughly make a discussion about the leakage to coupler states for the JPM scheme and the CNOT scheme in *Comment 5* and propose an innovative leakage interference scheme based on JPM unitary gate. Through simulations, we observe that the leakage to the coupler states during collective iSWAP-based JPM gate is relatively weak compared with that in CZ gate, suggesting that theoretically the gate operation speed of JPM gate can be further increased.

In addition, to further suppress the leakage to couplers, we can optimize the chip design parameters for implementing entangling gates. Specifically, we can design the coupler to support the large positive coupling instead of negative coupling utilized in our paper by moving the couplers in the positive direction during gate operations, so as to reduce leakage.

(2) Crosstalk

(2-1) Quantum crosstalk

The prominent quantum crosstalk is ZZ crosstalk, a result of the weak anharmonicity of transmons. Certainly, there are other forms of quantum crosstalk as well, such as next-neighbouring parasitic coupling crosstalk. However, these crosstalk typically exerts influence on qubits in the idle positions (where single-qubit gate operations are executed) with no distinction for both the JPM and CNOT schemes. Therefore, we focus on considering

ZZ crosstalk in the following three types based on surface code architecture:

① Static ZZ crosstalk

The first type of ZZ crosstalk occurs primarily during the execution of single-qubit gates, commonly recognized as static ZZ crosstalk. For instance, as illustrated in Fig. R3-2(a) where all measure qubits simultaneously execute Hadamard gates, there may exist static ZZ crosstalk. However, under these circumstances, the JPM and CNOT schemes are the same, as all qubits are biased at idle frequencies and the tunable couplers are biased to the appropriate frequencies to minimize ZZ crosstalk between adjacent qubits.

FIG. R3-2. ZZ crosstalk in the surface code architecture. (a) Static ZZ crosstalk during single-qubit gate operations. The red dotted line represents the potential static ZZ crosstalk may occur when the frequency of coupler is inappropriate. (b) ZZ crosstalk from spectator qubits may cause correlated error when CZ gates are simultaneously implemented during the parity measurement of surface code. Here up to six adjacent spectator qubits may generate six pairs of ZZ error to gate qubits circled by purple shade. (c) ZZ crosstalk from spectator qubits may cause correlated error when JPM unitary gates are simultaneously implemented during the parity measurement of surface code. Here only up to four adjacent spectator qubits may generate four pairs of ZZ error to gate qubits circled by purple shade (notice that although gate qubits are surrounded by seven spectator qubits, three of them are free and can be biased far away in principle). (d) Fidelity vs. ZZ crosstalk for both the JPM gate and the CZ gate. The effective spectator qubits in (b) and (c) are all initialized in $|1\rangle$ states or $|0\rangle + |1\rangle$ states.

② ZZ crosstalk from spectator qubits

The second type of ZZ crosstalk happens during the execution of multiqubit entangling gates. In the CNOT scheme, global parallel CZ gates appear in a specific order and direction, while in the JPM scheme, global parallel multiqubit JPM unitary gates (collective iSWAP-based or collective CZ-based) are also required. We now investigate the impact of ZZ crosstalk from spectator qubits surrounding these multiqubit entangling gates. In the case of one CZ gate in the CNOT scheme as shown in Fig. R3-2(b), six pairs of ZZ interactions may affect the two gate qubits; whereas for one JPM unitary gate in the JPM scheme as shown in Fig. R3-2(c), only four pairs of ZZ interactions are influential (with some spectator qubits remaining free and not participating in multiqubit operations, thus their influence can be disregarded). However, due to limitations imposed by frequency crowding during gate operations, effectively eliminating ZZ crosstalk between gate qubits and spectator qubits becomes challenging. Here we further validate the effect of ZZ crosstalk during gate operations through simulation. As depicted in Fig. R3-2(d), it can be observed that an increase in number of spectator qubits (resulting in an increased ZZ crosstalk) exacerbates infidelity of the gates.

Moreover, we should further verify the fidelity tolerance of both the JPM gate and the CZ

gate towards single ZZ interaction, which we define as FT here

$$FT = F(\xi_{ZZ} = \xi) - F(\xi_{ZZ} = 0).$$

This is because during gate operations, some gate qubits and spectator qubits exhibit a frequency difference out of the range where couplers are capable of suppressing ZZ crosstalk. Under such circumstance, investigating the fidelity tolerance of single ZZ interaction with respect to gate operations will showcase the robustness of the gate operation itself against correlated errors. We have conducted simulations to validate this aspect with the results presented in our response to *Reviewer 1* in *Comment 8*. Notably, in the CNOT scheme, when performing CZ gates, two qubits are distributed at two operation frequencies (owing to anharmonicity), potentially leading to increased frequency crowding globally; whereas in collective iSWAP-based JPM scheme, three qubits are distributed resonantly and share a common frequency position. Consequently, during parallel gates execution, there will be less frequency crowding issues compared to the CNOT scheme which facilitates easier elimination of ZZ errors indicated by red dotted line in Fig. R3-2(c). It should be acknowledged that if the JPM unitary gates are implemented based on collective CZ gate, the frequency crowding issues would align with that observed in the CNOT scheme.

In summary, considering the impact of ZZ crosstalk on the surface code parity measurement process, the JPM scheme could be potentially beneficial. However, it is crucial that the parallel CZ gates with global optimization has been explored in previous work (*Nature 574, 505-510 (2019)*), thereby the JPM gate deserves further exploration to enhance the global optimization capabilities.

③ Dynamic ZZ crosstalk

The third type of ZZ crosstalk is parasitic ZZ crosstalk during gate procedure (or defined as dynamic ZZ crosstalk). CZ gate and collective CZ-based JPM gate do not have parasitic ZZ crosstalk, but may exist leakage between two gate qubits $|01\rangle$ state to $|10\rangle$ state (*Phys. Rev. Lett. 123, 210501(2019)*), while iSWAP-based JPM gate may encounter parasitic ZZ crosstalk. However, the ZZ-free iSWAP gate scheme proposed in *Phys. Rev. X 11, 021058(2021)* may further provide a new idea to optimize JPM gate free of parasitic ZZ crosstalk.

(2-2) Classical crosstalk

① X crosstalk

The microwave X crosstalk may cause leakage from the qubit $|1\rangle$ state to the higher excited $|2\rangle$ state, resulting in correlated error. As mentioned in the supplementary materials of *Nature volume 614, pages676-681 (2023)*, this situation mainly occurs during single-qubit gate operations, where all the qubits are positioned at idle frequencies. When the driving frequency of a source qubit is close to the f_{21} of a receiver qubit, it will cause a parasitic $|1\rangle \rightarrow |2\rangle$ leakage error on the receiver qubit. Meanwhile, the solution and optimization methods are given in *Nature volume 614, pages676-681 (2023)*, which are consistent and applicable for both the CNOT scheme and the JPM scheme with no difference.

② Z crosstalk

The Z crosstalk mainly arises from the uneven potential distribution on the chip with no distinction for both the CNOT scheme and the JPM scheme. We have implemented several methods to minimize the impact of Z crosstalk in our chip. Firstly, we have incorporated numerous airbridges along the control line to equalize the potentials at different positions as much as possible. Additionally, we have employed a Z crosstalk matrix method (*Nature 519, 66–69 (2015)*) for precise evaluation and mitigation of the crosstalk as shown in Fig. S5 in the supplementary materials. Furthermore, in subsequent chip manufacturing, we can further

utilize the full-capped airbridges especially in the flip-chip architecture, which has been proved to effectively suppress Z crosstalk to an extremely low level (*arXiv:2401.03537*).

(3) Control system

The noise of the control waveform or signal may bring correlated errors such as leakage. For the JPM unitary gate and the CZ gate, the influence of this part primarily comes from the Z waveform when implementing the entangling gate operations, where the JPM gate requires at least four Z waveform modulations (including modulations of 2 qubits and 2 couplers), while the CZ gate only involves at least two Z waveform modulations (including modulations of 1 qubit and 1 coupler). Therefore the CZ gate may be easier in controlling the Z waveform compared with the JPM unitary gate. In practice, as mentioned by the referee in *Comment 1* that the calibration procedure of the JPM scheme may be more complex than the CNOT scheme, and thus additional complexity in control sequences for JPM scheme is inevitable. However, we believe that the calibration complexity of the JPM scheme is not a huge increase compared with the CNOT scheme, instead, to a certain extent, the JPM scheme may be more convenient (refer to our responses in *Comment 1*). Moreover, this type of correlated error may be further effectively suppressed through pulse optimization protocol, such as the calibration and correction of Z distortion (e.g. *Appl. Phys. Lett. 116, 054001 (2020)*), the selection of optimized pulse shape (e.g. *Phys. Rev. X 11, 021058 (2021)*), and even the usage of machine learning methods.

Finally, we thank the referee for this valuable question and make the corresponding revisions to our paper:

- We remove ERROR ANALYSES in the main text and transfer it to the supplementary materials with new data to fully explore the error sources for the JPM scheme with comparison to the CNOT scheme.
- We move Fig. S6 from the main text to the supplementary materials to illustrate error sources in the transversal parity measurement circuit.
- We add Fig. S7 in the supplementary materials to explore ZZ crosstalk in the surface code architecture.
- We add Fig. S8 in the supplementary materials to explore leakage interference methods for the JPM gate and the CZ gate.

Comment 5

Moreover, when comparing JPM to CNOT-based parity measurement schemes, the authors state that “the CZ gate time is set to be close to the gate time of the 2-qubit JPM unitary gate, for accurate comparison”. However, based on the description of the JPM unitary gate and the device parameters presented, it appears that a 2-qubit non-adiabatic CZ gate, fully exploiting leakage interference to approach the speed limit, could potentially perform faster than the JPM scheme (owing to the need to undo the minus phase in the Dicke states for the latter). Could the authors comment on the durations used in this manuscript for 2-qubit CZ gates and their respective speed limits?

Response:

We sincerely thank the referee for his/her thoughtful reading and insightful comments. We interpret the referee's intention as the CZ gate, fully exploiting the leakage interference to

approach the speed limit based on the previous theoretical and experimental work, may further be implemented faster than the proposed JPM unitary gate (by adjusting the coupler frequencies). In the following, we elaborate from three aspects to clarify the concerns from the referee.

(1) Brief review of previous leakage interference protocols

We first briefly review some leakage interference protocols based on the superconducting transmon qubits with tunable couplers. Two previous work has developed such protocol to find optimized two-qubit CZ gate control pulse (*Phys. Rev. X* 11, 021058(2021)) or speed limit and threshold of iSWAP gate operation (*Phys. Rev. Applied* 14, 024070(2020)), to achieve higher fidelity gate performance with less gate time. As depicted in Fig. 4 of *Phys. Rev. X* 11, 021058(2021), the leakage effect from different waveform pulses can be amplified and measured in the experiment via monitoring the leakage population after applying repeated CZ gate, thereby ensuring the optimized control pulse (slowly varying Slepian-based optimal control waveform) can effectively suppress the leakage to coupler states and improve the gate performance. Similarly, as shown in Fig. 10 in the supplementary material of *Phys. Rev. Applied* 14, 024070(2021), the authors identify the threshold for implementing iSWAP gate by means of monitoring leakage interference: the two qubits are initialized in $|11\rangle$, then brought into resonance by varying the coupler frequency, and finally the population of $|11\rangle$ state is measured. In practice, these findings and methods are of significant importance for the leakage interference protocols we presented below.

(2) Leakage interference protocols for CZ gate and JPM unitary gate

As the referee mentioned that the leakage interference can be exploited to approach the speed limit of CZ gate operation, here, we demonstrate that this method and idea are also applicable to JPM unitary gate.

① Protocol for JPM unitary gate

As an example, we consider the implementation of the collective iSWAP-based JPM unitary gate utilized in our experiment, where the most common leakage arises from qubits to couplers. Specifically, two data qubits Q_1 and Q_2 , along with one syndrome qubit Q_0 , are brought into resonance, and the corresponding two couplers $C_1 (Q_0 - Q_1)$ and $C_2 (Q_0 - Q_2)$ are adjusted to ensure the consistency of the effective coupling strength. If no leakage occurs during the gate operation, then the population of quantum state $|10101\rangle$ (ordered as $|Q_1 C_1 Q_0 C_2 Q_2\rangle$) will not change over time. Once the leakage occurs from the qubits to couplers, the $|10101\rangle$ state may transform to other potential leakage channels like $|01101\rangle$, $|00111\rangle$ and $|01110\rangle$, resulting in the variation of population evolution. Therefore, we can obtain the leakage interference image by measuring the population evolution of the $|10101\rangle$ state during the JPM unitary gate procedure with varied coupler frequencies. To verify our proposed leakage interference protocol for JPM unitary gate, we simulate this procedure through QuTip package in Python based on the experimental gate parameters with pulse level, as shown in Fig. R3-3(a). It is obvious that when the coupler frequencies are lower than the threshold, marked as the red dotted line, the population of $|10101\rangle$ manifests relatively large oscillating ripples, revealing the appearance of non-negligible leakage to other potential leakage channels.

② Protocol for CZ gate

Similarly, we also propose a leakage interference protocol for CZ gate: two qubits Q_1 , Q_2 , together with the coupler C , are initialized to $|001\rangle$ state (ordered as $|Q_1 C Q_2\rangle$), followed by implementing CZ gate procedure with varied coupler frequency. Assuming that no leakage is

happened during the gate operation, then the population of quantum state $|001\rangle$ will not change over time. Once the leakage occurs from the qubit to coupler, the $|001\rangle$ state may appear oscillating ripples, as the simulation results shown in Fig. R3-3(b). We can easily extract the lower bound for coupler frequency, marked as the red dotted line, to be higher than that in JPM unitary gate. In practice, this is not surprising since the operation frequency of one qubit when performing CZ gate is tuned higher (around 3.988 GHz for one qubit and 4.243 GHz for the other qubit in the experiment) to achieve resonance between two qubits $|11\rangle$ and $|02\rangle$ (or $|20\rangle$), while qubits in the JPM unitary gate are all positioned with the same frequencies (around 3.988 GHz in the experiment). Therefore, leakage is more likely to occur in the CZ gate than in the JPM unitary gate, leading to a lower threshold of coupler frequencies for the collective iSWAP-based JPM unitary gate.

FIG. R3-3. Simulation results of leakage interference for (a) collective iSWAP-based JPM unitary gate and (b) CZ gate based on the QuTip in Python. The system quantum states are initialized as (a) $|10101\rangle$ (ordered as $|Q_1C_1Q_0C_2Q_2\rangle$) and (b) $|001\rangle$ (ordered as $|Q_1CQ_2\rangle$). Notice that the range of the vertical axis in the two images is different because the coupler cannot be tuned lower than qubits' frequencies when implementing gate operations. The purple dotted line represents the experimental frequencies for couplers while the red dotted line marked in the figure represents the potential lower bound (here we choose around 10% leakage as the threshold considering the balance between the time and fidelity of the quantum gate) for implementing gate operation. It can be easily found that the threshold in (a) is lower than that in (b), thereby the gate operation time for collective iSWAP-based JPM unitary gate can be further reduced.

(3) Comparison of gate speed between collective iSWAP-based JPM unitary gate and CZ gate

Based on our simulation results of the leakage interference shown in Fig. R3-3, we find the collective iSWAP-based JPM unitary gate allows for faster gate speed compared with the CZ gate after fully exploiting the leakage interference. Therefore, the JPM scheme may further reduce the circuit time in experiment.

Finally, we thank the referee for this valuable question and add the corresponding analyses on leakage interference to our paper:

- We remove ERROR ANALYSES in the main text and transfer it to the supplementary materials with new data to fully explore the error sources for the JPM scheme with comparison to the CNOT scheme.
- We add Fig. S8 in the supplementary materials to explore leakage interference methods for the JPM gate and the CZ gate.

Comment 6

Considering that the proposed JPM scheme effectively corresponds to a three-qubit entangling gate, it should be expected that error propagation, particularly within the parity-measurement cycle, be more problematic than in the CNOT-based scheme. Could the authors comment on the effect of their proposed scheme on error propagation? Can parity check measurements implemented using this scheme be made fully fault tolerant?

Response:

We sincerely thank the referee for his/her thoughtful feedback and insightful comments. We agree with the reviewer's comments that the JPM scheme is effectively equivalent to a three-qubit gate, and therefore, it is worth of discussion about the error propagation during the parity-measurement cycle. The fault tolerance of a parity check protocol is influenced by both the protocol itself and the specific quantum code chosen. In the following, we would like to delve deeper into the analysis of the error propagation based on the two-qubit parity measurement case to demonstrate that the JPM procedure may be made fault-tolerant when applied to the standard surface code, characterized by rough and smooth boundaries. The findings can be extended to the four-qubit case with similar methods.

Recalling that the overall protocol for two-qubit parity measurement progress is described as (Eq.(5) in the main text)

$$U = U_s H_0 U_s H_0$$

with the JPM unitary gate $U_s = U_s(\pi/\sqrt{2}g)$ when the evolution time $t=\pi/(\sqrt{2}g)$. Ideally, we have

$$U = i\sigma_0^y(|00\rangle\langle 00| + |11\rangle\langle 11|) + I_0(|D^1\rangle\langle D^1| + |d^1\rangle\langle d^1|)$$

and

$$U_s = \sigma_0^z(|00\rangle\langle 00| - |11\rangle\langle 11|) + I_0(|d^1\rangle\langle d^1| - |D^1\rangle\langle D^1|)$$

where

$$|D^1\rangle = (|01\rangle + |10\rangle)/\sqrt{2}, \quad |d^1\rangle = (|01\rangle - |10\rangle)/\sqrt{2},$$

are the corresponding two-qubit Dicke bases introduced in the main text. However, the error can occur and propagates during the implementation of U_s . Generally speaking, errors can be categorized into two types: intrinsic coherent errors, which is a consequence of imperfect control, and those induced by interactions with the external environment (including the one causing leakage out of the computational space), commonly referred to as decoherence. In conducting error propagation analyses for syndrome measurement, it is crucial to ascertain those failures, which are highly probable in causing low weight errors, do not escalate into high weight errors affecting the data qubits. And this is critical to ensure the fault-tolerance.

Taking into account failures of infinite small strength, denoted as ϵ , for both scenarios, the most critical situation to consider is when a weight 2 error occurs on both data qubits, occurs with a strength at the order of $o(\epsilon)$ — such faulty path can deteriorate the fault-tolerance and should be avoided by choosing a suitable error correction code. Therefore, our analysis should primarily focus on the error propagation resulting from failures with a strength approximating $o(\epsilon)$ in the process of parity check U . We further focus on the failure of U_s process, since the failures and their propagation on Hadamard gates are trivial and obviously

benign.

(1) Coherent errors caused by imperfect control

(1-1) Imperfect JPM unitary $U'_s = U_s + \delta U_s$

We first consider the coherent errors caused by imperfect control and neglect the decoherence (since the strength of both of them to occur is $o(\epsilon^2)$). Ideally the validity of the JPM parity measurement requires coupling strength between ancilla and both data qubits to be equal, i.e., $g_1 = g_2 = g$. In practical operation, situations that are not strictly equal may occasionally occur, e.g., $g_1 = g, g_2 = g + \delta g, \delta g \ll g$. The deviations in coupling strengths may lead to imperfect control of U_s . In this case, the Hamiltonian can be denoted as

$$H' = \sigma_0^{01}(g_1\sigma_1^{10} + g_2\sigma_2^{10}) + H. c..$$

Define $\Omega' = \sqrt{g_1^2 + g_2^2} = \Omega + \delta\Omega, \Omega = \sqrt{2}g$, we can effectively decompose the three qubits Hilbert space $\mathbb{C}^{\otimes 8}$ into $H_1 \oplus H_2$ where

$$H_1 = \text{span}\{|100\rangle, |0D_1^1\rangle, |1D_2^1\rangle, |011\rangle\}, \quad H_2 = \text{span}\{|000\rangle, |0d_1^1\rangle, |1d_2^1\rangle, |111\rangle\}$$

Here,

$$|D_1^1\rangle = \frac{g_1|10\rangle + g_2|01\rangle}{\Omega'} = \cos\left(\frac{\pi}{4} + \epsilon\right)|10\rangle + \sin\left(\frac{\pi}{4} + \epsilon\right)|01\rangle = |D^1\rangle + \epsilon|d^1\rangle + o(\epsilon^2),$$

$$|d_1^1\rangle = \frac{g_1|01\rangle - g_2|10\rangle}{\Omega'} = \cos\left(\frac{\pi}{4} + \epsilon\right)|01\rangle - \sin\left(\frac{\pi}{4} + \epsilon\right)|10\rangle = |d^1\rangle - \epsilon|D^1\rangle + o(\epsilon^2),$$

$$|D_2^1\rangle = \frac{g_2|10\rangle + g_1|01\rangle}{\Omega'} = \sin\left(\frac{\pi}{4} + \epsilon\right)|10\rangle + \cos\left(\frac{\pi}{4} + \epsilon\right)|01\rangle = |D^1\rangle - \epsilon|d^1\rangle + o(\epsilon^2),$$

$$|d_2^1\rangle = \frac{g_2|01\rangle - g_1|10\rangle}{\Omega'} = \sin\left(\frac{\pi}{4} + \epsilon\right)|01\rangle - \cos\left(\frac{\pi}{4} + \epsilon\right)|10\rangle = |d^1\rangle + \epsilon|D^1\rangle + o(\epsilon^2),$$

with $\epsilon \approx \frac{\delta g}{2g} = \frac{\delta\Omega}{\Omega}$ and $\delta g = \sqrt{2}\delta\Omega$. Then the dynamics of Hamiltonian H' follows can be calculated as

$$\begin{aligned} U'_s &= |000\rangle\langle 000| + |111\rangle\langle 111| + |0d_1^1\rangle\langle 0d_1^1| + |1d_2^1\rangle\langle 1d_2^1| \\ &\quad + \cos(\Omega't)I_1 - i\sin(\Omega't)\sigma_1^x + \cos(\Omega't)I_2 - i\sin(\Omega't)\sigma_2^x \\ U'_s &= |000\rangle\langle 000| + |111\rangle\langle 111| + |0d_1^1\rangle\langle 0d_1^1| + |1d_2^1\rangle\langle 1d_2^1| \\ &\quad - \left(1 - \frac{(\delta\Omega t)^2}{2}\right)(|100\rangle\langle 100| + |0D_1^1\rangle\langle 0D_1^1|) + i(\delta\Omega t)(|100\rangle\langle 0D_1^1| + |0D_1^1\rangle\langle 100|) \\ &\quad - \left(1 - \frac{(\delta\Omega t)^2}{2}\right)(|011\rangle\langle 011| + |1D_2^1\rangle\langle 1D_2^1|) + i(\delta\Omega t)(|011\rangle\langle 1D_2^1| + |1D_2^1\rangle\langle 011|) \end{aligned}$$

with the time condition $\cos(\Omega t) = \pi$, and I_1, σ_1^x (I_2, σ_2^x) defined on the two-level subspace spanned by $|100\rangle, |0D_1^1\rangle$ ($|011\rangle, |1D_2^1\rangle$) respectively. Note that the errors due to limited precision of time control is of the same strength order forms of $\delta\Omega t$, and hence would not be discussed here for simplicity. Expanding the above U'_s expression further and keeping only the first-order expansion, we have

$$U'_s = U_s + \delta U_s$$

and

$$\delta U_s = i\delta\Omega t \left[\begin{array}{l} \sigma_0^x \otimes \sigma_1^x (|01\rangle_{01}\langle 01| + |10\rangle_{01}\langle 10|) \otimes I_2 \\ + \sigma_0^x \otimes \sigma_2^x (|01\rangle_{02}\langle 01| + |10\rangle_{02}\langle 10|) \otimes I_1 \end{array} \right] - 2\epsilon\sigma_0^z$$

(1-2) Effect of imperfect gate on the parity measurement circuit U'

Next we will step further to consider the effect of imperfect gate on the parity measurement circuit. Taking the ZZ parity check as an example, we have

$$U' = U'_s H_0 U'_s H_0 = U_s H_0 U_s H_0 + U_s H_0 \delta U_s H_0 + \delta U_s H_0 U_s H_0$$

① The error at the end of parity check implementation U' caused by $\delta\Omega$ terms in δU_s

We first consider the error effect on U' caused by the first $\delta\Omega$ -correlated term, i.e.,

$$(|01\rangle_{01}\langle 01| + |10\rangle_{01}\langle 10|) \otimes I_2 = \frac{1}{2}(I_0 \otimes I_1 - \sigma_0^z \otimes \sigma_1^z) \otimes I_2.$$

Specifically, the error induced by the two different paths can be demonstrated as

$$U_s H_0 \delta U_s H_0 / (i\delta\Omega t)$$

$$= \left[\begin{array}{l} \sigma_0^z (|00\rangle_{12}\langle 00| - |11\rangle_{12}\langle 11|) \\ + I_0 (|d^1\rangle_{12}\langle d^1| - |D^1\rangle_{12}\langle D^1|) \end{array} \right] H_0 \sigma_0^x \otimes \sigma_1^x (|01\rangle_{01}\langle 01| + |10\rangle_{01}\langle 10|) \otimes I_2 H_0$$

$$= \left[\begin{array}{l} \sigma_0^z \otimes \sigma_1^z (|00\rangle_{12}\langle 00| + |11\rangle_{12}\langle 11|) \\ - (|D^1\rangle_{12}\langle D^1| + |d^1\rangle_{12}\langle d^1|) \sigma_1^x \otimes \sigma_2^x \end{array} \right] H_0 \sigma_0^x \otimes \sigma_1^x \frac{1}{2} (I_0 \otimes I_1 - \sigma_0^z \otimes \sigma_1^z) \otimes I_2 H_0$$

$$= \frac{i}{2} \sigma_1^y (|D^1\rangle_{12}\langle D^1| + |d^1\rangle_{12}\langle d^1|) - \frac{1}{2} \sigma_0^x \otimes \sigma_1^x (|D^1\rangle_{12}\langle D^1| + |d^1\rangle_{12}\langle d^1|)$$

$$- \frac{1}{2} \sigma_0^z \otimes \sigma_2^x (|00\rangle_{12}\langle 00| + |11\rangle_{12}\langle 11|) + \frac{i}{2} \sigma_0^y \otimes \sigma_1^z \otimes \sigma_2^x (|00\rangle_{12}\langle 00| + |11\rangle_{12}\langle 11|),$$

$$\delta U_s H_0 U_s H_0 / (i\delta\Omega t)$$

$$= \sigma_0^x \otimes \sigma_1^x (|01\rangle_{01}\langle 01| + |10\rangle_{01}\langle 10|) \otimes I_2 \cdot H_0 \left[\begin{array}{l} \sigma_0^z (|00\rangle_{12}\langle 00| - |11\rangle_{12}\langle 11|) \\ + I_0 (|d^1\rangle_{12}\langle d^1| + |D^1\rangle_{12}\langle D^1|) \end{array} \right] H_0$$

$$= \frac{1}{2} \sigma_0^x \otimes \sigma_1^x (I_0 \otimes I_1 - \sigma_0^z \otimes \sigma_1^z) \otimes I_2 \cdot H_0 \left[\begin{array}{l} \sigma_0^z \otimes \sigma_1^z (|00\rangle_{12}\langle 00| + |11\rangle_{12}\langle 11|) \\ - I_0 \otimes \sigma_1^x \otimes \sigma_2^x (|D^1\rangle_{12}\langle D^1| + |d^1\rangle_{12}\langle d^1|) \end{array} \right] H_0$$

$$= -\frac{i}{2} \sigma_1^y (|00\rangle_{12}\langle 00| + |11\rangle_{12}\langle 11|) + \frac{1}{2} \sigma_0^z \otimes \sigma_1^x (|00\rangle_{12}\langle 00| + |11\rangle_{12}\langle 11|)$$

$$- \frac{1}{2} \sigma_0^x \otimes \sigma_2^x (|D^1\rangle_{12}\langle D^1| + |d^1\rangle_{12}\langle d^1|) + \frac{i}{2} \sigma_0^y \otimes \sigma_1^z \otimes \sigma_2^x (|D^1\rangle_{12}\langle D^1| + |d^1\rangle_{12}\langle d^1|),$$

Note that all error terms occurring with a strength of $\delta\Omega$ can project to odd or even parity space followed by the weight 1 or 2 Pauli errors, and all these Pauli errors are not problematic for $U(2)$ parity check. The error induced by the second term $(|01\rangle_{02}\langle 01| + |10\rangle_{02}\langle 10|) \otimes I_1$ can also be calculated in a similar way and shown to be benign.

② The error at the end of parity check implementation U' caused by ϵ terms in δU_s

The term $2\epsilon\sigma_0^z \otimes \sigma_1^z (|D^1\rangle\langle D^1| + |d^1\rangle\langle d^1|)$ causes the parity check implementation U' to accumulate errors as

$$\begin{aligned} & U_s H_0 \delta U_s H_0 + \delta U_s H_0 U_s H_0 \\ & = 2\epsilon\sigma_0^x \otimes \sigma_1^z (|d^1\rangle\langle d^1| - |D^1\rangle\langle D^1|) - 2\epsilon\sigma_0^z \otimes \sigma_1^z (|d^1\rangle\langle d^1| - |D^1\rangle\langle D^1|) \end{aligned}$$

after the two faulty paths. It can be observed that in this situation, the term

$$\sigma_1^z (|d^1\rangle\langle d^1| - |D^1\rangle\langle D^1|) = \frac{1}{2} I_0 \otimes (-i\sigma_1^y \otimes \sigma_2^x + \sigma_1^x \otimes i\sigma_2^y)$$

actually introduces the weight 2 error on both data qubits. Fortunately, we can prevent the propagation of such Pauli errors by selecting proper encoding method, such as the standard

surface code architecture. While error propagation can compromise the fault-tolerance of rotated surface codes, standard surface codes with rough and smooth boundaries maintain full tolerance as errors don't propagate along the logical X operator chain. The situation is similar for JPM procedure for X -style stabilizer. Therefore, JPM is more compatible with standard surface codes.

(2) Decoherence errors caused by interactions with the environment

To analyze the impact of decoherence, which arises from interactions with the environment, on the quantum operation U_s involving three qubits, we adopt a simplified error model. Typically, one might consider weight-3 correlated Pauli noise, reflecting errors on all three qubits simultaneously. However, given that the underlying Hamiltonian primarily governs two-body interactions, it's more pragmatic to model the noise using weight-1 and weight-2 correlated Pauli errors (one on the ancilla qubits, one on the data qubit). These errors are akin to terms such as $X, Y, Z, XX, XY, XZ, \dots, ZZ$, occurring either before or after the U_s operation.

The operation U_s can be thought of as functionally analogous to a sequence of two CNOT gates, commonly used in standard syndrome measurement in quantum error correction. This analogy suggests that the pattern of error propagation in our system, influenced by U_s , should mirror that observed with consecutive CNOT operations.

For instance, consider the case when error ϵ_{C_1} (ϵ_{S_0}) acting on the data (syndrome) qubit, e.g, Q_1 (Q_0), before JPM gate U in our scheme. The error propagation of the bit-flip error, phase flip error, or both flip error through the JPM gate can be calculated as

① error on the first data qubit

$$U_s X_{C_1} U_s^\dagger = -Z_{C_1} Z_{S_0} X_{C_2}, \quad U_s Y_{C_1} U_s^\dagger = -Z_{C_1} Z_{S_0} Y_{C_2}, \quad U_s Z_{C_1} U_s^\dagger = I_{C_1} I_{S_0} Z_{C_2}$$

② error on syndrome qubit

$$U_s X_{S_0} U_s^\dagger = -Z_{C_1} X_{S_0} Z_{C_2}, \quad U_s Y_{S_0} U_s^\dagger = -Z_{C_1} Y_{S_0} Z_{C_2}, \quad U_s Z_{S_0} U_s^\dagger = I_{C_1} Z_{S_0} I_{C_2}$$

Then for the 2-qubit JPM measure-Z circuit, that is, $H_0 U_s H_0 U_s = U_z$, the error propagation can be recognized as

$$U_z X_{C_1} U_z^\dagger = Y_{C_1} Y_{S_0} Z_{C_2}, \quad U_z Y_{C_1} U_z^\dagger = -X_{C_1} Y_{S_0} Z_{C_2}, \quad U_z Z_{C_1} U_z^\dagger = Z_{C_1} I_{S_0} I_{C_2}$$

for the data qubit error, and

$$U_z X_{S_0} U_z^\dagger = -Z_{C_1} X_{S_0} Z_{C_2}, \quad U_z Y_{S_0} U_z^\dagger = I_{C_1} Y_{S_0} I_{C_2}, \quad U_z Z_{S_0} U_z^\dagger = -Z_{C_1} Z_{S_0} Z_{C_2}$$

for the syndrome qubit error, respectively. The error propagation for the 2-qubit measure-X circuit can be obtained in a similar way.

Therefore, by focusing on weight-1 and weight-2 Pauli errors, we align our error model with the practical constraints of the system and the typical error dynamics seen in standard quantum error correction procedures. Hence, based on the above analyses and as far as we considered, the JPM should be fully fault-tolerant.

Comment 7

In motivating the merit of the proposed scheme, a comparison with a CNOT-based parity measurement scheme features prominently. In the manuscript, the authors attribute the main source of error in the JPM scheme to qubit decoherence but, at the same time, state that the CZ gate time is set to be close to the gate time of the 2-qubit JPM unitary gate. As such, decoherence alone does not seem to capture the difference in performance between the two gates. Given the important role of this comparison in motivating the JPM scheme, it would be important for the authors to more thoroughly study the error channels currently limiting the performance of their CNOT scheme. Do the authors anticipate that further optimization could bring the performance of the CNOT-based scheme closer to that of the JPM scheme? Could the authors comment on the reason why error channels limiting the performance of the CNOT-based scheme do not equally affect the performance of the JPM scheme?

Response:

We thank the referee for his/her rigorous consideration and valuable questions. We agree with the referee that *"As such, decoherence alone dose not seem to capture the difference in performance between the two gates"* and our intention is to propose a new parity measurement scheme with feasible experimental implementation, and thus the comparison with CNOT scheme is to help analyze the potential error sources in JPM unitary gate. We also agree with the referee that *"... further optimization could bring the performance of the CNOT-based scheme closer to that of the JPM scheme"* and believe that both schemes can perform better with optimized gate operation and careful circuit optimization. Therefore, we feel sorry about our confusing expression in the original manuscript such as *"The average parity detection fidelity of the JPM scheme [...] outperforms the CNOT scheme [...] in this experiment"*, and further focus on the innovation of the scheme itself as suggested by the *Reviewer #2* *"..., while the comparison is interesting, I don't see the emphasis on better performance as necessary for the message of the paper. Indeed, this new scheme's working principle is interesting no matter what, as long as fidelities as comparable, which seems to be the case here."* Furthermore, regarding the phenomenon we observed in the experiment, where the parity measurement fidelity of the JPM scheme was slightly higher than that of the CNOT scheme, we carefully considered several reasons combined with the experimental analyses and simulations to clarify the referee's concerns.

Overall, we think the mechanism may be related to the fact that the CZ gate is more susceptible to the influence of global circuit quantum crosstalk (ZZ crosstalk) and leakage (leakage to coupler or leakage to qubit higher excited state). Besides, the impact of two-level systems (TLSs) on multi-qubit gate operations is an important consideration, as it can lead to errors in the gate operation and degradation of the overall scheme performance. As the operating frequencies of the qubits may differ during the JPM and CZ operations, the impact of TLSs on the two gate types may also differ. We believe that by optimizing the overall performance of the gate operation, such as reducing the frequency difference between idle positions of qubits within their anharmonicities, and improving the quality of the superconducting materials to reduce TLSs, as well as optimizing control waveforms, we can further improve the parity measurement fidelity in experiment.

(1) Theoretical analyses

We first conduct some theoretical calculations and analyses based on the experimental data. In order to ensure consistency in the calibration of gate fidelity for both the JPM unitary gate and CZ gate, we use the Quantum Process Tomography (QPT) with eliminating the SPAM error. The specific experimental parameters are as follows:

- JPM gate

We choose Q_0, Q_2 and Q_4 to implement JPM unitary gate with Q_0 functioned as the syndrome qubit. In the experiment, Q_2 and Q_4 are tuned to the frequency of Q_0 , and both couplers C_{02} and C_{04} are adjusted to around 4.395 GHz. At this point, the gate operation time for the JPM gate is about 79 ns, and the average gate fidelity after eliminating the SPAM error is found to be 98.5% through QPT.

- CZ gate

We re-calibrate the CZ gates for $Q_0 - Q_2$ and $Q_0 - Q_4$ pairs separately. Keeping the frequency of Q_0 constant, we adjust the frequencies of the other qubit to make the $|11\rangle$ and $|02\rangle$ (or $|20\rangle$) states into resonate, thereby accumulating phase and implementing the CZ gate operation with C_{02} and C_{04} adjusted to around 4.395 GHz respectively. At this point, the gate operation time for both CZ gate pairs is also about 72 ns, and the average gate fidelity after eliminating the SPAM error is found to be 98.4% and 98.5% through QPT.

Ideally, the main source of circuit error comes from two-qubit gate errors when we ignore the qubits' decoherence error and coherent error, thereby the upper bound of parity measurement fidelity can be simply calculated as the product of the two-qubit gate fidelity, i.e. the JPM gate $98.5\% \times 98.5\% \approx 97.0\%$; the CZ gate $98.4\% \times 98.5\% \approx 96.9\%$. It can be observed that the upper limits are theoretically quite close, indicating that the difference in the upper limits between the two schemes would not be particularly large when the gate fidelity is comparable from an experimental perspective, and the fidelity errors reflected in the experiment would in principle be small.

(2) Simulation results

However, in our experiment, we find that under the current experimental environment and parameters, the parity fidelity of JPM scheme is slightly higher than that of CNOT scheme, as mentioned in our main text. We think that the reason may be related to the susceptibility of CZ gate to circuit crosstalk and leakage.

(2-1) Fidelity tolerance of ZZ crosstalk

Similar to the method utilized in *Phys. Rev. Lett.* 127, 060505(2021), we simulate the impact of spectator qubit on gate operations for both JPM gate and CZ gate in presence of ZZ crosstalk by QuTip, and further obtain the tolerance of gate operations to ZZ error. For instance, as shown in Fig. R3-4(a), the system Hamiltonian for implementing JPM unitary gate with one spectator qubit can be expressed as:

$$H = \sum_{i=0}^2 \frac{\alpha_i}{2} a_i^\dagger a_i^\dagger a_i a_i + \frac{\alpha_s}{2} a_s^\dagger a_s^\dagger a_s a_s + g(a_0 a_1^\dagger + a_1 a_0^\dagger) + g(a_0 a_2^\dagger + a_2 a_0^\dagger) + \xi_{ZZ} a_2^\dagger a_2 a_s^\dagger a_s,$$

where α_i ($i = 0 \sim 2$) is the anharmonicities of qubits Q_0, Q_1 and Q_2 , α_s is the anharmonicity of spectator qubit S_1 ; g is the effective coupling strength during JPM unitary gate and ξ_{ZZ} represents the ZZ interaction between spectator qubit S_1 and one data qubit Q_2 . We simulate the gate fidelity by QPT with varying the interaction strength of ξ_{ZZ} , and calculate the fidelity tolerance to ZZ crosstalk by subtracting the fidelity when $\xi_{ZZ} = 0$, i.e.

$$FT = F(\xi_{ZZ} = \xi) - F(\xi_{ZZ} = 0)$$

where FT represents fidelity tolerance. The simulation results can be found in Fig. R3-4(b) where we investigate two situations with spectator qubit in $|1\rangle$ state and $|0\rangle + |1\rangle$ state. Apparently, JPM gate with collective iSWAP gate is more tolerant to ZZ crosstalk compared with CZ gate. In practice, this may be reasonable since ZZ crosstalk can directly impact the phase accumulation which seems to be more sensitive to CZ gate procedure. In our experiment, the coupler offers the effective coupling strength between neighbouring qubits and hence may induce unwanted ZZ crosstalk if coupler is not efficiently closed. We believe through optimizing the parity circuit by fine tuning the coupler frequency may further improve the parity measurement procedure especially for CNOT scheme. Moreover, we also explore the quantum crosstalk in the surface code architecture for the JPM scheme and CNOT scheme in our response to *Comment 4*.

FIG. R3-4. (a) The schematic diagram for simulating gate fidelity tolerance to ZZ crosstalk in presence of spectator qubit. The dotted box represents the gate procedure. In experiment, spectator qubits around the gate qubits may induce unwanted ZZ interaction which reducing the gate fidelity and further parity measurement fidelity. (b) Fidelity tolerance to ZZ crosstalk for both the JPM gate and the CZ gate. The spectator qubit is initialized in $|1\rangle$ state or $|0\rangle + |1\rangle$ state.

(2-2) Leakage channels

Error channels parasitic in the parity measurement circuit may further cause unwanted leakage like leakage to coupler state (details in our response to *Comment 5*) and qubit higher excited state (details in our response to *Comment 4*). It can be observed in Fig. R3-3 in *Comment 5*, the leakage seems to be more serious for CZ gate as marked by the purple dotted line (experimental point). This is because the coupler frequency is more close to the qubit frequency for CZ gate. To accurately compare the parity measurement fidelity of JPM gate and CZ gate in the experiment, we consistently maintain the operation frequency of the syndrome qubit Q_0 at around 3.988 GHz, while we uniformly adjust it to 3.988 GHz when executing the JPM gate and to around 4.243 GHz when executing the CZ gate for the data qubit. During the parity measurement circuit, this leakage effect may be further exacerbated, ultimately declining the parity fidelity. Similarly, as we mentioned in *Comment 4*, CZ gate may be also easier to suffer from the leakage to qubit $|2\rangle$ states since diabatic CZ gate utilizes the swap procedure between $|11\rangle$ and $|02\rangle$ (or $|20\rangle$). In practice, these results reflect that the JPM scheme based on collective iSWAP will be more tolerant to such leakage. Nevertheless, we also believe that selecting a more appropriate parameter range to suppress leakage will further improve the fidelity of the CZ gate, thereby enhancing the parity measurement fidelity.

(2-3) Experimental limitations

In order to ensure a fair comparison between the JPM scheme and the CNOT schemes in the experiment, particularly considering the limitations of the four-qubit JPM scheme, we are only able to utilize $Q_0 - Q_4$ for the experimental analyses (see structure of sample in Fig. 2(a) in the main text). However, we find the performance of $Q_5 - Q_8$ may be better in the experiment since these qubits exhibit less Two-level systems (TLSs) and higher fidelities in implementing CZ gate with adjacent qubits, as can be found in Fig. S4 of the supplementary materials. Therefore, as mentioned by the referee, if there were no restrictions on comparing these two schemes, employing qubit pairs with better performance would further enhance parity detection fidelity.

In addition, when implementing collective-iSWAP based JPM unitary gate, all the qubits are operated at a single frequency, allowing for easier selection of a frequency with reduced TLSs and greater stability. Conversely, achieving similar characteristics for CZ gates is relatively challenging due to their reliance on multiple operating frequencies. Therefore, from this aspect, the JPM scheme can be beneficial to not only circumvent frequency crowding, but also facilitate multiqubit gate operations.

Finally, according to the above experimental analyses and simulations, combined with the referees' suggestions, we have made revisions to the relevant statements to improve the presentation and claim of our paper:

- In lines 17-18 in ABSTRACT, we revise the sentence to "... which shows comparable performance to the CNOT scheme."
- In line 92 in INTRODUCTION, we revise the sentence to "..., which are comparable to that of the CNOT scheme in this experiment."
- In line 339 in EXPERIMENTAL DEMONSTRATION, we revise the sentence to "... can be achieved comparable at around 95.2% (90.0%) for the iSWAP-based JPM scheme, ...".
- In the caption of Fig. 4, we revise the sentence to "... is on par with the CNOT scheme ...".

Reviewer #1 (Remarks to the Author):

I have read the authors' responses and I am glad to see that the manuscript is now more accessible. Except for comment 10, I find the responses to my previous questions are clear and sufficient. I thank the authors for their great effort.

Further modifications are required regarding comment 10. As stated in the revised main text, the target entangled state in 4-qubit case is $1/2(|1001\rangle+|1100\rangle+|0011\rangle+|0110\rangle)$ can be rewritten as $1/2(|01\rangle+|10\rangle)\otimes(|01\rangle+|10\rangle)$. Actually, this is a separable tensor product state of two two-qubit subsystems and is not a genuine 4-qubit entanglement by definition. Additionally, the current data in the 4-qubit JEP case only involves excitation populations and is insufficient for entanglement verification. In my assessment, the discussion on the entanglement generation based on the 4-qubit JEP scheme is a bit misleading and should be removed.

Reviewer #2 (Remarks to the Author):

The authors have taken appropriate care of the comments made by myself in a previous round. While I still find the clarity to have margin for improvement, the manuscript has substantially improved by the updates made. I would suggest to accept the manuscript for publication.

Reviewer #3 (Remarks to the Author):

Having read through all three reviews and respective rebuttal, we greatly appreciate the breadth and depth of the authors' response. We believe the authors have sufficiently addressed our comments in the revised manuscript and with the additional Supplemental Materials. As such, we feel confident in recommending publication of this manuscript in Nature Communications.

Authors' Response to the Review Comments

Manuscript ID: NCOMMS-23-25497

Title of Paper: Fast joint parity measurement via collective interactions induced by stimulated emission

Date Sent: March, 2024

Response to Reviewer #1

Overall comments

I have read the authors' responses and I am glad to see that the manuscript is now more accessible. Except for comment 10, I find the responses to my previous questions are clear and sufficient. I thank the authors for their great effort.

Response:

We sincerely thank the referee for his/her time and efforts in reviewing our manuscript for the second time. We are pleased to know that our revised manuscript has clarified the referee's comments and suggestions as "... the manuscript is now more accessible." and "Except for comment 10, I find the responses to my previous questions are clear and sufficient." We are also glad to hear that our manuscript has been accepted for publication with the suggestion of the referee. In response to the suggestions raised in comment 10 by the referee, we have made further changes in the final manuscript based on his/her feedback.

Comment 1

Further modifications are required regarding comment 10. As stated in the revised main text, the target entangled state in 4-qubit case is $1/2(|1001\rangle + |1100\rangle + |0011\rangle + |0110\rangle)$ can be rewritten as $1/2(|01\rangle + |10\rangle) \otimes (|01\rangle + |10\rangle)$. Actually, this is a separable tensor product state of two two-qubit subsystems and is not a genuine 4-qubit entanglement by definition. Additionally, the current data in the 4-qubit JEP case only involves excitation populations and is insufficient for entanglement verification. In my assessment, the discussion on the entanglement generation based on the 4-qubit JEP scheme is a bit misleading and should be removed.

Response:

We thank the referee for his/her careful reading and valuable suggestions. The referee is right that here the target 4-qubit state stated in the previous version of our main text is not a genuine 4-qubit entanglement by definition. Therefore, to avoid potential misunderstanding, we have made the corresponding revisions to our manuscript regarding the entanglement

verification for 4-qubit JEP scheme based on the referee's valuable suggestions. Specifically, we have removed Fig.6(c) and the relevant descriptions of entanglement verification for 4-qubit JEP scheme in the final manuscript.

Response to Reviewer #2

Overall Comments

The authors have taken appropriate care of the comments made by myself in a previous round. While I still find the clarity to have margin for improvement, the manuscript has substantially improved by the updates made. I would suggest to accept the manuscript for publication.

Response:

We sincerely thank the referee for his/her time and efforts in reviewing our manuscript for the second time. We are pleased to know that our revised manuscript has clarified the referee's comments and suggestions as "... *the manuscript has substantially improved by the updates made.*" We are also glad to hear that our manuscript has been accepted for publication with the suggestion of the referee.

Response to Reviewer #3

Overall Comments

Having read through all three reviews and respective rebuttal, we greatly appreciate the breadth and depth of the authors' response. We believe the authors have sufficiently addressed our comments in the revised manuscript and with the additional Supplemental Materials. As such, we feel confident in recommending publication of this manuscript in Nature Communications.

Response:

We sincerely thank the referee for his/her time and efforts in reviewing our manuscript for the second time. We are pleased to know that the referee recognizes our work and our response as "..., *we greatly appreciate the breadth and depth of the authors' response.*" and "*We believe the authors have sufficiently addressed our comments in the revised manuscript and with the additional Supplemental Materials.*" We are also glad to hear that our manuscript has been accepted for publication with the suggestion of the referee.